# Clipped Stochastic Methods
# for Variational Inequalities with Heavy-Tailed Noise

**Eduard Gorbunov**[*][†]
MIPT, Russia
Mila & UdeM, Canada
MBZUAI, UAE

**Marina Danilova**[*]
MIPT, Russia

**David Dobre**[*]
Mila & UdeM, Canada

**Pavel Dvurechensky**
WIAS, Germany

**Alexander Gasnikov**
MIPT, Russia
HSE University, Russia
IITP RAS, Russia

**Gauthier Gidel**
Mila & UdeM, Canada
Canada CIFAR AI Chair

## Abstract

Stochastic first-order methods such as Stochastic Extragradient (SEG) or Stochastic Gradient Descent-Ascent (SGDA) for solving smooth minimax problems and, more generally, variational inequality problems (VIP) have been gaining a lot of attention in recent years due to the growing popularity of adversarial formulations in machine learning. However, while high-probability convergence bounds are known to reflect the actual behavior of stochastic methods more accurately, most convergence results are provided in expectation. Moreover, the only known high-probability complexity results have been derived under restrictive sub-Gaussian (light-tailed) noise and bounded domain assumption [Juditsky et al., 2011a]. In this work, we prove the first high-probability complexity results with logarithmic dependence on the confidence level for stochastic methods for solving monotone and structured non-monotone VIPs with non-sub-Gaussian (heavy-tailed) noise and unbounded domains. In the monotone case, our results match the best-known ones in the light-tails case [Juditsky et al., 2011a], and are novel for structured non-monotone problems such as negative comonotone, quasi-strongly monotone, and/or star-cocoercive ones. We achieve these results by studying SEG and SGDA with clipping. In addition, we numerically validate that the gradient noise of many practical GAN formulations is heavy-tailed and show that clipping improves the performance of SEG/SGDA.

## 1 Introduction

Recently, game formulations have been receiving a lot of interest from the optimization and machine learning communities. In such problems, different models/players competitively minimize their loss functions, e.g., see adversarial example games [Bose et al., 2020], hierarchical reinforcement learning [Wayne and Abbott, 2014, Vezhnevets et al., 2017], and generative adversarial networks (GANs) [Goodfellow et al., 2014]. Very often, such problems are studied through the lens of solving a *variational inequality problem* (VIP) [Harker and Pang, 1990, Ryu and Yin, 2021, Gidel et al.,

---

[*]Equal contribution.
[†]Corresponding author: `eduard.gorbunov@mbzuai.ac.ae`

36th Conference on Neural Information Processing Systems (NeurIPS 2022).

2019a]. In the unconstrained case, for an operator[3] $F : \mathbb{R}^d \to \mathbb{R}^d$ VIP can be written as follows:

$$\text{find } x^* \in \mathcal{X}^* \qquad \text{where} \qquad \mathcal{X}^* := \{x^* \in \mathbb{R}^d \text{ such that } F(x^*) = 0\}. \qquad \text{(VIP)}$$

In machine learning applications, operator $F$ usually has an expectation form, i.e., $F(x) = \mathbb{E}_\xi[F_\xi(x)]$, where $x$ corresponds to the parameters of the model, $\xi$ is a sample from some (possibly unknown) distribution $\mathcal{D}$, and $F_\xi(x)$ is the operator corresponding to the sample $\xi$.

Such problems are typically solved via first-order stochastic methods such as Stochastic Extragradient (SEG), also known as Mirror-Prox algorithm [Juditsky et al., 2011a], or Stochastic Gradient Descent-Ascent (SGDA) [Dem'yanov and Pevnyi, 1972, Nemirovski et al., 2009] due to their practical efficiency. However, despite the significant attention to these methods and their modifications, their convergence is usually analyzed in expectation only, e.g., see [Gidel et al., 2019a, Hsieh et al., 2019, 2020, Mishchenko et al., 2020, Loizou et al., 2021]. In contrast, while high-probability analysis more accurately reflects the behavior of the stochastic methods [Gorbunov et al., 2020], a little is known about it in the context of solving VIP. To the best of our knowledge, there is only one work addressing this question for monotone variational inequalities [Juditsky et al., 2011a]. However, [Juditsky et al., 2011a] derive their results under the assumption that the problem has "light-tailed" (sub-Gaussian) noise and bounded domain, which is restrictive even for simple classes of minimization problems [Zhang et al., 2020b]. This leads us to the following open question.

> **Q1:** *Is it possible to achieve the same high-probability results as in [Juditsky et al., 2011a] without assuming that the noise is sub-Gaussian and the domain is bounded?*

Next, in the context of GANs' training, empirical investigation [Jelassi et al., 2022] and practical use [Gulrajani et al., 2017, Miyato et al., 2018, Tran et al., 2019, Brock et al., 2019, Sauer et al., 2022] indicate the practical superiority of Adam-based methods, e.g., alternating SGDA with stochastic estimators from Adam [Kingma and Ba, 2014], over classical methods like SEG or (alternating) SGDA. This interesting phenomenon has no rigorous theoretical explanation yet. In contrast, there exists a partial understanding of why Adam-like methods perform well in different tasks such as training attention models [Zhang et al., 2020b]. In particular, Zhang et al. [2020b] empirically observe that stochastic gradient noise is heavy-tailed in several NLP tasks and theoretically shows that vanilla SGD [Robbins and Monro, 1951] can diverge in such cases and its version with gradient clipping (clipped-SGD) [Pascanu et al., 2013] converges. Moreover, the state-of-the-art high-probability convergence results for heavy-tailed stochastic minimization problems are also obtained for the methods with gradient clipping [Nazin et al., 2019, Gorbunov et al., 2020, 2021, Cutkosky and Mehta, 2021]. Since Adam can be seen as a version of adaptive clipped-SGD with momentum [Zhang et al., 2020b], these advances establish the connection between good performance of Adam, heavy-tailed gradient noise, and gradient clipping for *minimization problems*. Motivated by these recent advances in understanding the superiority of Adam for minimization problems, we formulate the following research question.

> **Q2:** *In the training of popular GANs, does the gradient noise have heavy-tailed distribution and does gradient clipping improve the performance of the classical SEG/SGDA?*

In this paper, we give positive answers to **Q1** and **Q2**. In particular, we derive high-probability results for clipped-SEG and clipped-SDGA for monotone and structured non-monotone VIPs with non-sub-Gaussian noise, validate that the gradient noise in several GANs formulations is indeed heavy-tailed, and show that gradient clipping does significantly improve the results of SEG/SGDA in these tasks. That is, our work closes a noticeable gap in theory of stochastic methods for solving VIPs.

## 1.1 Technical Preliminaries

Before we summarize our main contributions, we introduce some notations and technical assumptions.

**Notation.** Throughout the text $\langle x, y \rangle$ is the standard inner-product, $\|x\| = \sqrt{\langle x, x \rangle}$ denotes $\ell_2$-norm, $B_r(x) = \{u \in \mathbb{R}^d \mid \|u - x\| \leq r\}$. $\mathbb{E}[X]$ and $\mathbb{E}_\xi[X]$ are full and conditional expectations w.r.t. the randomness coming from $\xi$ of random variable $X$. $\mathbb{P}\{E\}$ denotes the probability of event $E$. $R_0 = \|x^0 - x^*\|$ denotes the distance between the starting point $x^0$ of a method and the solution $x^*$ of VIP.[4]

---

[3]For example, when we deal with a differentiable game/minimax problem, $F$ can be chosen as the concatenation of the gradients of the players' objective functions, e.g., see [Gidel et al., 2019a] for the details.

[4]If not specified, we assume that $x^*$ is the projection of $x^0$ to the solution set of VIP.

**High-probability convergence for VIP.** For deterministic VIPs there exist several convergence metrics $\mathcal{P}(x)$. These metrics include restricted gap-function [Nesterov, 2007] $\text{Gap}_R(x) := \max_{y \in B_R(x^*)} \langle F(y), x - y \rangle$, (averaged) squared norm of the operator $\|F(x)\|^2$, and squared distance to the solution $\|x - x^*\|^2$. Depending on the assumptions on the problem, one or another criterion is preferable. For example, for monotone and strongly monotone problems $\text{Gap}_R(x)$ and $\|x - x^*\|^2$ are valid metrics of convergence, while in the non-monotone case, one typically has to use $\|F(x)\|^2$.

In the stochastic case, one needs either to upper bound $\mathbb{E}[\mathcal{P}(x)]$ or to derive a bound on $\mathcal{P}(x)$ that holds with some probability. In this work, we focus on the second type of bounds. That is, for a given confidence level $\beta \in (0, 1]$, we aim at deriving bounds on $\mathcal{P}(x)$ that hold with probability at least $1 - \beta$, where $x$ is produced by clipped-SEG/clipped-SGDA. However, to achieve this goal one has to introduce some assumptions on the stochastic noise such as *bounded variance* assumption, which is standard in the stochastic optimization literature [Ghadimi and Lan, 2012, 2013, Juditsky et al., 2011b, Nemirovski et al., 2009]. We rely on a weaker version of this assumption.

**Assumption 1.1** (Bounded Variance). *There exists a bounded set $Q \subseteq \mathbb{R}^d$ and $\sigma \geq 0$ such that*

$$\mathbb{E}\left[\|F_\xi(x) - F(x)\|^2\right] \leq \sigma^2, \quad \forall\, x \in Q. \tag{1}$$

In contrast to most of the existing works assuming (1) on the whole space/domain, we need (1) to hold only on some *bounded* set $Q$. More precisely, in the analysis, we rely on (1) only on some ball $B_r(x^*)$ around the solution $x^*$ and radius $r \sim R_0$. Although we consider an *unconstrained* VIP, we manage to show that the iterates of clipped-SEG/clipped-SGDA stay inside this ball with high probability. Therefore, for achieving our purposes, it is sufficient to make all the assumptions on the problem only in some ball around the solution.

We also notice that the majority of existing works providing high-probability analysis with logarithmic dependence[5] on $1/\beta$ rely on the so-called *light tails* assumption: $\mathbb{E}[\exp(\|F_\xi(x) - F(x)\|^2/\sigma^2)] \leq \exp(1)$ meaning that the noise has a sub-Gaussian distribution. This assumption always implies (1) but not vice versa. However, even for minimization problems, there are only few works that do not rely on the light tails assumption [Nazin et al., 2019, Davis et al., 2021, Gorbunov et al., 2020, 2021, Cutkosky and Mehta, 2021]. In the context of solving VIPs, the existing high-probability guarantees in [Juditsky et al., 2011a] rely on the light tails assumption.

**Optimization properties.** We also need to introduce several assumptions about the operator $F$. We start with the standard Lipschitzness. As we write above, all assumptions are introduced only on some bounded set $Q$, which will be specified later.

**Assumption 1.2** (Lipschitzness). *Operator $F : \mathbb{R}^d \to \mathbb{R}^d$ is $L$-Lipschitz on $Q \subseteq \mathbb{R}^d$, i.e.,*

$$\|F(x) - F(y)\| \leq L\|x - y\|, \quad \forall\, x, y \in Q. \tag{Lip}$$

Next, we need to introduce some assumptions[6] on the monotonicity of $F$, since approximating local first-order optimal solutions is intractable in the general non-monotone case [Daskalakis et al., 2021, Diakonikolas et al., 2021]. We start with the standard monotonicity assumption and its relaxations.

**Assumption 1.3** (Monotonicity). *Operator $F : \mathbb{R}^d \to \mathbb{R}^d$ is monotone on $Q \subseteq \mathbb{R}^d$, i.e.,*

$$\langle F(x) - F(y), x - y \rangle \geq 0, \quad \forall\, x, y \in Q. \tag{Mon}$$

**Assumption 1.4** (Star-Negative Comonotonicity). *Operator $F : \mathbb{R}^d \to \mathbb{R}^d$ is $\rho$-star-negatively comonotone on $Q \subseteq \mathbb{R}^d$ for some $\rho \in [0, +\infty)$, i.e., for any $x^*$ we have*

$$\langle F(x), x - x^* \rangle \geq -\rho\|F(x)\|^2, \quad \forall\, x \in Q. \tag{SNC}$$

*When $\rho = 0$, the operator $F$ is called star-monotone (SM) on $Q$.*

(SNC) is also known as weak Minty condition [Diakonikolas et al., 2021], a relaxation of negative comonotonicity [Bauschke et al., 2021]. Another name of star-monotonicity (SM) is variational stability condition [Hsieh et al., 2020]. The following assumption is a relaxation of strong monotonicity.

---

[5]Using Markov inequality, one can easily derive high-probability bounds with non-desirable polynomial dependence on $1/\beta$, e.g., see the discussion in [Davis et al., 2021, Gorbunov et al., 2021].

[6]Each complexity result, which we derive, relies only on one or two of these assumptions simultaneously.

Table 1: Summary of known and new high-probability complexity results for solving variational inequalities. Column "Setup" indicates the additional assumptions in addition to Assumption 1.1. All assumptions are made only on some ball around the solution with radius $\sim R_0$ (unless the opposite is indicated). By the complexity we mean the number of stochastic oracle calls needed for a method to guarantee that $\mathbb{P}\{\text{Metric} \leq \varepsilon\} \geq 1 - \beta$ for some $\varepsilon > 0$, $\beta \in (0, 1]$ and "Metric" is taken from the corresponding column. For simplicity, we omit numerical and logarithmic factors in the complexity bounds. Column "HT?" indicates whether the result is derived in the heavy-tailed case (without assuming that the noise is sub-Gaussian) and column "UD?" shows whether the analysis works on unbounded domains. Notation: $\widetilde{x}_{\text{avg}}^K = \frac{1}{K+1} \sum_{k=0}^K \widetilde{x}^k$ (for clipped-SEG), $x_{\text{avg}}^K = \frac{1}{K+1} \sum_{k=0}^K x^k$ (for clipped-SGDA), $L$ = Lipschitz constant; $D$ = diameter of the domain (used in [Juditsky et al., 2011a]); $\text{Gap}_D(x) = \max_{y \in \mathcal{X}} \langle F(y), x - y \rangle$, where $\mathcal{X}$ is a bounded domain with diameter $D$ where the problem is defined (used in [Juditsky et al., 2011a]); $\sigma^2$ = bound on the variance (in the results from [Juditsky et al., 2011a] $\sigma^2$ is a sub-Gaussian variance); $R$ = any upper bound on $\|x^0 - x^*\|$; $\mu$ = quasi-strong monotonicity parameter; $\ell$ = star-cocoercivity parameter.

| Setup | Method | Citation | Metric | Complexity | HT? | UD? |
|---|---|---|---|---|---|---|
| (Mon)+(Lip) | Mirror-Prox | [Juditsky et al., 2011a][(1)] | $\text{Gap}_D(\widetilde{x}_{\text{avg}}^K)$ | $\max\left\{\frac{LD^2}{\varepsilon}, \frac{\sigma^2 D^2}{\varepsilon^2}\right\}$ | ✗ | ✗ |
| | clipped-SEG | Thm. C.1 & Cor. C.1 | $\text{Gap}_R(\widetilde{x}_{\text{avg}}^K)$ | $\max\left\{\frac{LR^2}{\varepsilon}, \frac{\sigma^2 R^2}{\varepsilon^2}\right\}$ | ✓ | ✓ |
| (SNC)+(Lip) | clipped-SEG | Thm. C.2 & Cor. C.2 [(2)] | $\frac{1}{K+1} \sum\limits_{k=0}^K \|F(x^k)\|^2$ | $L^2 \max\left\{\frac{R^2}{\varepsilon}, \frac{\sigma^2 R^2}{\varepsilon^2}\right\}$ | ✓ | ✓ |
| (QSM)+(Lip) | clipped-SEG | Thm. C.3 & Cor. C.3 | $\|x^K - x^*\|^2$ | $\max\left\{\frac{L}{\mu}, \frac{\sigma^2}{\mu\varepsilon}\right\}$ | ✓ | ✓ |
| (Mon)+(SC) | clipped-SGDA | Thm. D.1 & Cor. D.1 | $\text{Gap}_R(x_{\text{avg}}^K)$ | $\max\left\{\frac{\ell R^2}{\varepsilon}, \frac{\sigma^2 R^2}{\varepsilon^2}\right\}$ | ✓ | ✓ |
| (SC) | clipped-SGDA | Thm. D.2 & Cor. D.2 | $\frac{1}{K+1} \sum\limits_{k=0}^K \|F(x^k)\|^2$ | $\ell^2 \max\left\{\frac{R^2}{\varepsilon}, \frac{\sigma^2 R^2}{\varepsilon^2}\right\}$ | ✓ | ✓ |
| (QSM)+(SC) | clipped-SGDA | Thm. D.3 & Cor. D.3 | $\|x^K - x^*\|^2$ | $\max\left\{\frac{\ell}{\mu}, \frac{\sigma^2}{\mu\varepsilon}\right\}$ | ✓ | ✓ |

[(1)] Monotonicity and Lipschitzness of $F$ are assumed on the whole domain.
[(2)] The results holds for any $0 \leq \rho \leq 1/(640LA)$, where $A = \ln \frac{8(K+1)}{\beta}$, if parameters of the method are set properly. Moreover, batchsizes should be large enough (see Thm. 2.1 and C.2 for the details).

**Assumption 1.5** (Quasi-Strong Monotonicity). *Operator $F : \mathbb{R}^d \to \mathbb{R}^d$ is $\mu$-quasi strongly monotone on $Q \subseteq \mathbb{R}^d$ for some $\mu \geq 0$, i.e.,*

$$\langle F(x), x - x^* \rangle \geq \mu \|x - x^*\|^2, \quad \forall x \in Q, \quad \text{where } x^* \text{ is the unique solution of (VIP).} \quad \text{(QSM)}$$

Under this name, the above assumption is introduced by [Loizou et al., 2021], but it is also known as strong coherent Song et al. [2020] and strong stability [Mertikopoulos and Zhou, 2019] conditions. Moreover, unlike strong monotonicity that always implies monotonicity, (QSM) can hold for non-monotone operators [Loizou et al., 2021, Appendix A.6]. However, in contrast to (Mon), (QSM) allows to achieve linear convergence[7] of deterministic methods for solving VIP.

Finally, we consider a relaxation of standard cocoercivity: $\ell\langle F(x) - F(y), x - y \rangle \geq \|F(x) - F(y)\|^2$.

**Assumption 1.6** (Star-Cocoercivity). *Operator $F : \mathbb{R}^d \to \mathbb{R}^d$ is $\ell$-star-cocoercive on $Q \subseteq \mathbb{R}^d$ for some $\ell > 0$, i.e.,*

$$\|F(x)\|^2 \leq \ell\langle F(x), x - x^* \rangle, \quad \forall x \in Q, \quad \text{where } x^* \text{ is the projection of } x \text{ on } \mathcal{X}^*. \quad \text{(SC)}$$

This assumption is introduced by [Loizou et al., 2021]. One can construct an operator $F$ being star-cocoercive, but not cocoercive [Gorbunov et al., 2022b]. Moreover, although cocoercivity implies monotonicity and Lipschitzness, there exist operators satisfying (SC), but neither (Mon) nor (Lip) [Loizou et al., 2021, §A.6]. We summarize the relations between the assumptions in Fig. 1.

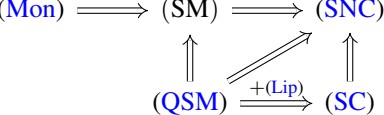

Figure 1: Relation between the assumptions on the structured non-monotonicity of the problem.

### 1.2 Our Contributions

◇ **New high-probability results for VIPs with heavy-tailed noise.** In our work, we circumvent the limitations of the existing high-probability analysis of stochastic methods for solving VIPs [Juditsky et al., 2011a] and derive the first high-probability results for the methods that solve monotone VIPs with heavy-tailed noise in the unconstrained case. The key algorithmic ingredient helping us to achieve these results is a proper modification of SEG and SGDA based on the *gradient clipping* and leading to our clipped-SEG and clipped-SGDA. Moreover, we derive several high-probability results for clipped-SEG/clipped-SGDA applied to solve structured non-monotone VIPs. To the best of our knowledge, these results do not have analogs even under the light tails assumption. We summarize the derived complexity results in Tbl. 1.

---

[7]Linear convergence can be also achieved under different relatively weak assumptions such as sufficient bilinearity [Abernethy et al., 2019, Loizou et al., 2020] or error bound condition [Hsieh et al., 2020].

$\diamond$ **Tight analysis.** The derived complexities satisfy a desirable for high-probability results property: they have logarithmic dependence on $1/\beta$, where $\beta$ is a confidence level. Next, up to the logarithmic factors, our results match known lower bounds[8] in monotone and strongly monotone cases [Beznosikov et al., 2020]. Moreover, we recover and even improve the results from [Juditsky et al., 2011a], which are obtained in the light-tailed case, since our bounds do not depend on the diameter of the domain, see Tbl. 1 for the details.

$\diamond$ **Weak assumptions.** One of the key features of our theoretical results is that it relies on assumptions restricted *to a ball around the solution*. We achieve this via showing that, with high probability, clipped-SEG/clipped-SGDA do not leave a ball (with a radius proportional to $R_0$) around the solution. In contrast, the existing works on stochastic methods for solving VIPs usually make assumptions such as boundedness of the variance and Lipschitzness on the whole domain of the considered problem. Since many practical tasks are naturally unconstrained, such assumptions become too unrealistic since, e.g., stochastic gradients and their variance in the training of neural networks with more than 2 layers grow polynomially fast when $x$ goes to infinity. However, for a large class of problems including the ones with polynomially growing operators, boundedness of the variance and Lipschitzness hold on *any compact set*. That is, our analysis covers a broad class of problems.

$\diamond$ **Numerical experiments.** We empirically observe that heavy-tailed gradient noise arises in the training of several practical GANs formulations including StyleGAN2 and WGAN-GP. Moreover, our experiments show that gradient clipping significantly improves the convergence of SEG/SGDA on such tasks. These results shed a light on why Adam-based methods are good at training GANs. Our codes are publicly available: https://github.com/busycalibrating/clipped-stochastic-methods.

### 1.3 Closely Related Work

In this section, we discuss the most closely related works. Further discussion is deferred to § A.

**High-probability convergence.** To the best of our knowledge, the only work deriving high-probability convergence in the context of solving VIPs is [Juditsky et al., 2011a]. In particular, Juditsky et al. [2011a] consider monotone and Lipschitz VIP defined[9] on a convex compact set $\mathcal{X}$ with the diameter $D := \max_{x,y \in \mathcal{X}} \|x - y\|$, and assume that the noise in $F_\xi(x)$ is light-tailed. In this setting, Juditsky et al. [2011a] propose a stochastic version of the celebrated Extragradient method (EG) [Korpelevich, 1976] with non-Euclidean proximal setup – Mirror-Prox. Moreover, Juditsky et al. [2011a] derive that after $K = \mathcal{O}(\max\{\frac{LD^2}{\varepsilon}, \frac{\sigma^2 D^2}{\varepsilon^2} \ln^2 \frac{1}{\beta}\})$ stochastic oracle calls for some $\varepsilon > 0$, $\beta \in (0,1]$, the averaged extrapolated iterate $\widetilde{x}_{\text{avg}}^K = \frac{1}{K+1} \sum_{k=0}^K \widetilde{x}^k$ (see also clipped-SEG) satisfies $\text{Gap}_D(\widetilde{x}_{\text{avg}}^K) \le \varepsilon$ with probability at least $1 - \beta$. Although this result has a desirable logarithmic dependence on $1/\beta$ and optimal dependence on $\varepsilon$ [Beznosikov et al., 2020], it is derived only for (i) light-tailed case and (ii) bounded domains. Our results do not have such limitations.

## 2 Clipped Stochastic Extragradient

In this section, we consider a version of SEG with gradient clipping. That is, we apply clipping operator $\text{clip}(y, \lambda) := \min\{1, \lambda/\|y\|\}y$, which is defined for any $y \in \mathbb{R}^d$ (when $y = 0$ we set $\text{clip}(0, \lambda) := 0$) and any clipping level $\lambda > 0$, to the mini-batched stochastic estimators used both at the extrapolation and update steps of SEG. This results in the following iterative algorithm:

$$x^{k+1} = x^k - \gamma_2 \widetilde{F}_{\boldsymbol{\xi}_2^k}(\widetilde{x}^k), \quad \text{where} \quad \widetilde{x}^k = x^k - \gamma_1 \widetilde{F}_{\boldsymbol{\xi}_1^k}(x^k), \quad \text{(clipped-SEG)}$$

$$\widetilde{F}_{\boldsymbol{\xi}_1^k}(x^k) = \text{clip}\left(\frac{1}{m_{1,k}} \sum_{i=1}^{m_{1,k}} F_{\xi_1^{i,k}}(x^k), \lambda_{1,k}\right), \quad \widetilde{F}_{\boldsymbol{\xi}_2^k}(\widetilde{x}^k) = \text{clip}\left(\frac{1}{m_{2,k}} \sum_{i=1}^{m_{2,k}} F_{\xi_2^{i,k}}(\widetilde{x}^k), \lambda_{2,k}\right),$$

where $\{\xi_1^{i,k}\}_{i=1}^{m_{1,k}}, \{\xi_2^{i,k}\}_{i=1}^{m_{2,k}}$ are independent samples from the distribution $\mathcal{D}$. In the considered algorithm, clipping bounds the effect of heavy-tailedness of the gradient noise, but also creates a bias

---

[8]These lower bounds are derived for the convergence in expectation. Deriving tight lower bounds for the convergence with high-probability for solving stochastic VIPs is an open question.

[9]In this case, the goal is to find $x^* \in \mathcal{X}$ such that inequality $\langle F(x^*), x - x^* \rangle \ge 0$ holds for all $x \in \mathcal{X}$.

that one has to properly control in the analysis. Moreover, we allow different stepsizes $\gamma_1, \gamma_2$ [Hsieh et al., 2020, Diakonikolas et al., 2021, Gorbunov et al., 2022a], different batchsizes $m_{1,k}, m_{2,k}$, and different clipping levels $\lambda_{1,k}, \lambda_{2,k}$. In particular, taking $\gamma_2 < \gamma_1$ is crucial for our analysis to handle VIP satisfying star-negative comonotonicity (SNC) with $\rho > 0$. Our convergence results for clipped-SEG are summarized in the following theorem. For simplicity, we omit here the numerical constants, which are explicitly given in § C.

**Theorem 2.1.** *Consider* clipped-SEG *run for $K \geq 0$ iterations. Let $R \geq R_0 = \|x^0 - x^*\|$.*
***Case 1.*** *Let Assump. 1.1, 1.2, 1.3 hold for $Q = B_{4R}(x^*)$, where $\gamma_1 = \gamma_2 = \gamma$ with $0 < \gamma = \mathcal{O}(1/(LA))$, $\lambda_{1,k} = \lambda_{2,k} \equiv \lambda = \Theta(R/(\gamma A))$, $m_{1,k} = m_{2,k} \equiv m = \Omega(\max\{1, (K+1)\gamma^2\sigma^2 A/R^2\})$, where $A = \ln\frac{6(K+1)}{\beta}$, $\beta \in (0, 1]$ are such that $A \geq 1$.*
***Case 2.*** *Let Assump. 1.1, 1.2, 1.4 hold for $Q = B_{3R}(x^*)$, where $\gamma_2 + 2\rho \leq \gamma_1 \leq \mathcal{O}(1/(LA))$, $\lambda_{1,k} \equiv \lambda_1 = \Theta(R/(\gamma_1 A))$, $\lambda_{2,k} \equiv \lambda_2 = \Theta(R/(\gamma_2 A))$, $m_{1,k} \equiv m_1 = \Omega(\max\{1, \max\{\gamma_1\gamma_2(K+1), \sqrt{\gamma_1^3\gamma_2(K+1)}\}\sigma^2 A/R^2\})$, $m_{2,k} \equiv m_2 = \Omega(\max\{1, (K+1)\gamma_2^2\sigma^2 A/R^2\})$, where $A = \ln\frac{8(K+1)}{\beta}$, $\beta \in (0, 1]$ are such that $A \geq 1$.*
***Case 3.*** *Let Assump. 1.1, 1.2, 1.5 hold for $Q = B_{3R}(x^*)$, where $\gamma_1 = \gamma_2 = \gamma$ with $0 < \gamma \leq \mathcal{O}(1/(LA))$, $\lambda_{1,k} = \lambda_{2,k} = \lambda_k = \Theta(\exp(-\gamma\mu(1+k/2))R/(\gamma A))$, $m_{1,k} = m_{2,k} = m_k = \Omega(\max\{1, (K+1)\gamma^2\sigma^2 A/(\exp(-\gamma\mu k)R^2)\})$, where $A = \ln\frac{6(K+1)}{\beta}$, $\beta \in (0, 1]$ are such that $A \geq 1$.*

*Then, to guarantee $\mathtt{Gap}_R(\widetilde{x}_{avg}^K) \leq \varepsilon$ in **Case 1** with $\widetilde{x}_{avg}^K = \frac{1}{K+1}\sum_{k=0}^{K}\widetilde{x}^k$, $\frac{1}{K+1}\sum_{k=0}^{K}\|F(x^k)\|^2 \leq L\varepsilon$ in **Case 2**, $\|x^K - x^*\|^2 \leq \varepsilon$ in **Case 3**, with probability $\geq 1 - \beta$ clipped-SEG requires*

$$\textbf{Case 1 and 2}: \quad \widetilde{\mathcal{O}}\left(\max\left\{\frac{LR^2}{\varepsilon}, \frac{\sigma^2 R^2}{\varepsilon^2}\right\}\right) \qquad and \qquad \textbf{Case 3}: \quad \widetilde{\mathcal{O}}\left(\max\left\{\frac{L}{\mu}, \frac{\sigma^2}{\mu\varepsilon}\right\}\right) \quad (2)$$

*oracle calls. The above guarantees hold in two different regimes: large step-sizes $\gamma \approx 1/LA$ (requiring large batch-sizes), and small step-sizes, allowing small batch-sizes $m = \mathcal{O}(1)$.*

*Proof sketch in **Case 1***. Modifying the analysis of EG, we first derive that for all $t \geq 0$ we have $\mathtt{Gap}_R(\widetilde{x}_{avg}^t)$ and $\|x^t - x^*\|^2$ are not greater than $\max_{u \in B_R(x^*)}\{\|x^0 - u\|^2 + 2\gamma\sum_{l=0}^{t-1}\langle x^l - u - \gamma F(\widetilde{x}^l), \theta_l\rangle + \gamma^2\sum_{l=0}^{t-1}(\|\theta_l\|^2 + 2\|\omega_l\|^2)\}$ if $x^l, \widetilde{x}^l$ lie in $B_{4R}(x^*)$ for all $l = 0, 1, \ldots, t - 1$, where $\theta_l = F(\widetilde{x}^l) - \widetilde{F}_{\boldsymbol{\xi}_2^l}(\widetilde{x}^l)$ and $\omega_l = F(x^l) - \widetilde{F}_{\boldsymbol{\xi}_1^l}(x^l)$. Next, using this recursion and the induction argument, we show that with high probability $x^t, \widetilde{x}^t \in B_{4R}(x^*)$ for all $t = 0, 1, \ldots, K + 1$. This gives us an upper bound for $\mathtt{Gap}_R(\widetilde{x}_{avg}^t)$. After that, we upper bound $\max_{u \in B_R(x^*)}\{2\gamma\sum_{l=0}^{t-1}\langle x^l - u - \gamma F(\widetilde{x}^l), \theta_l\rangle + \gamma^2\sum_{l=0}^{t-1}(\|\theta_l\|^2 + 2\|\omega_l\|^2)\}$ by $5R^2$ with high-probability. We achieve this via the proper choice of the clipping level $\lambda = \Theta(R/(\gamma A))$ implying that $\|F(\widetilde{x}^l)\| \leq \lambda/2$ and $\|F(x^l)\| \leq \lambda/2$ with high probability for all $l = 0, 1, \ldots, t - 1$. This clipping politics helps to properly bound the bias and the variance of the clipped estimators using Lem. B.2. After that, it remains to apply the Bernstein inequality for the martingale differences (Lem. B.1). See the detailed proof in § C. □

In addition to the discussion given in the introduction (see § 1.2, 1.3 and Tbl. 1), we provide here several important details about the derived results. First of all, we notice that up to the logarithmic factors depending on $\beta$ our high-probability convergence results recover the state-of-the-art in-expectation ones for SEG in the monotone [Beznosikov et al., 2020], star-negative comonotone [Diakonikolas et al., 2021], and quasi-strongly monotone [Gorbunov et al., 2022a] cases. Moreover, as we show in Corollaries C.1 and C.3, to achieve these results in monotone and quasi-strongly monotone cases, it is sufficient to choose constant batchsize $m = \mathcal{O}(1)$ and small enough stepsize $\gamma$. In contrast, when the operator is star-negatively comonotone, we do rely on the usage of large stepsize $\gamma_1$ and large batchsize $m_1 = \mathcal{O}(K)$ for the extrapolation step to obtain (2). However, known in-expectation results from [Diakonikolas et al., 2021, Lee and Kim, 2021] also rely on large $\mathcal{O}(K)$ batchsizes in this case. We leave the investigation of this limitation to the future work.

# 3 Clipped Stochastic Gradient Descent-Ascent

In this section, we extend the approach described above to the analysis of SGDA with clipping. That is, we consider the following algorithm:

$$x^{k+1} = x^k - \gamma \widetilde{F}_{\boldsymbol{\xi}^k}(x^k), \quad \text{where} \quad \widetilde{F}_{\boldsymbol{\xi}^k}(x^k) = \texttt{clip}\left(\frac{1}{m_k}\sum_{i=1}^{m_k} F_{\xi^{i,k}}(x^k), \lambda_k\right) \quad \text{(clipped-SGDA)}$$

where $\{\xi^{i,k}\}_{i=1}^m$ are independent samples from the distribution $\mathcal{D}$. In the above method, clipping plays a similar role as in clipped-SEG. Our convergence results for clipped-SGDA are summarized below. The general idea of the proof is similar to the one for clipped-SEG, see the details in Appendix D, where we also explicitly give the constants that are omitted here for simplicity.

**Theorem 3.1.** *Consider clipped-SGDA run for $K \geq 0$ iterations. Let $R \geq R_0 \geq \|x^0 - x^*\|$.*
*Case 1. Let Assump. 1.1, 1.3, 1.6 hold for $Q = B_{3R}(x^*)$, where $0 < \gamma = \mathcal{O}(1/(\ell A))$, $\lambda_k \equiv \lambda = \Theta(R/(\gamma A))$, $m_k \equiv m = \Omega(\max\{1, (K+1)\gamma^2\sigma^2 A/R^2\})$, where $\beta \in (0,1]$ and $A = \ln \frac{6(K+1)}{\beta} \geq 1$.*
*Case 2. Let Assump. 1.1, 1.6 hold for $Q = B_{2R}(x^*)$, where $0 < \gamma = \mathcal{O}(1/(\ell A))$, $\lambda_k \equiv \lambda = \Theta(R/(\gamma A))$, $m_k \equiv m = \Omega(\max\{1, (K+1)\gamma^2\sigma^2 A/R^2\})$, where $\beta \in (0,1]$ and $A = \ln \frac{4(K+1)}{\beta} \geq 1$.*
*Case 3. Let Assump. 1.1, 1.5, 1.6 hold for $Q = B_{2R}(x^*)$, where $0 < \gamma = \mathcal{O}(1/(\ell A))$, $\lambda_k = \Theta(\frac{e^{-\gamma\mu(1+k/2)}R}{(\gamma A)})$, $m_k = \Omega(\max\{1, \frac{(K+1)\gamma^2\sigma^2 A}{e^{\gamma\mu k}R^2}\})$, where $\beta \in (0,1]$ and $A = \ln \frac{4(K+1)}{\beta} \geq 1$.*

*Then, to guarantee $\texttt{Gap}_R(\widetilde{x}_{avg}^K) \leq \varepsilon$ in Case 1 with $\widetilde{x}_{avg}^K = \frac{1}{K+1}\sum_{k=0}^K \widetilde{x}^k$, $\frac{1}{K+1}\sum_{k=0}^K \|F(x^k)\|^2 \leq \ell\varepsilon$ in Case 2, $\|x^K - x^*\|^2 \leq \varepsilon$ in Case 3, with probability $\geq 1 - \beta$, clipped-SGDA requires*

$$\textbf{Case 1 and 2:} \quad \widetilde{\mathcal{O}}\left(\max\left\{\frac{\ell R^2}{\varepsilon}, \frac{\sigma^2 R^2}{\varepsilon^2}\right\}\right) \qquad and \qquad \textbf{Case 3:} \quad \widetilde{\mathcal{O}}\left(\max\left\{\frac{\ell}{\mu}, \frac{\sigma^2}{\mu\varepsilon}\right\}\right) \tag{3}$$

*oracle calls. The above guarantees hold in two different regimes: large step-sizes $\gamma \approx 1/\ell A$ (requiring large batch-sizes), and small step-sizes, allowing small batch-sizes $m = \mathcal{O}(1)$.*

As for the clipped-SEG, up to the logarithmic factors depending on $\beta$, our high-probability convergence results from the theorem above recover the state-of-the-art in-expectation ones for SGDA under monotonicity and star-cocoercivity [Beznosikov et al., 2022], star-cocoercivity [Gorbunov et al., 2022a][10], quasi-strong monotonicity and star-cocoercivity [Loizou et al., 2021] assumptions. Moreover, in *all the cases*, one can achieve the results from (3) using constant batchsize $m = \mathcal{O}(1)$ and small enough stepsize $\gamma$ (see Corollaries D.1, D.2, D.3 for the details). Finally, to the best of our knowledge, we derive the first high-probability convergence results for SGDA-type methods.

# 4 Experiments

To validate our theoretical results, we conduct experiments on heavy-tailed min-max problems to demonstrate the importance of clipping when using non-adaptive methods such as SGDA or SEG. We train a Wasserstein GAN with gradient penalty [Gulrajani et al., 2017] on CIFAR-10 [Krizhevsky et al., 2009] using SGDA, clipped-SGDA, and clipped-SEG, and show the evolution of the gradient noise histograms during training. We demonstrate that gradient clipping also stabilizes the training of more sophisticated GANs when using SGD by training a StyleGAN2 model [Karras et al., 2020] on FFHQ [Karras et al., 2019], downsampled to $128 \times 128$. Although the generated sample quality is not competitive with a model trained with Adam, we find that StyleGAN2 fails to generate anything meaningful when trained with regular SGDA whereas clipped methods learn meaningful features. We do not claim to outperform state-of-the-art adaptive methods (such as Adam); our focus is to validate our theoretical results and to demonstrate that clipping improves SGDA in this context.

**WGAN-GP.** In this section, we focus on the ResNet architecture proposed in Gulrajani et al. [2017], and we adapt our code from a publicly available WGAN-GP implementation.[11] We first compute the gradient noise distribution and validate if it is heavy-tailed. Taking the fixed randomly initialized

---

[10]Although Gorbunov et al. [2022a] do not consider SGDA explicitly, it does fit their framework implying that SGDA achieves $\frac{1}{K+1}\sum_{k=0}^K \mathbb{E}[\|F(x^k)\|^2] \leq \varepsilon$ after $K = \mathcal{O}(\max\{\frac{\ell R^2}{\varepsilon}, \frac{\sigma^2 R^2}{\varepsilon^2}\})$ iterations with $m = 1$.

[11]https://github.com/w86763777/pytorch-gan-collections

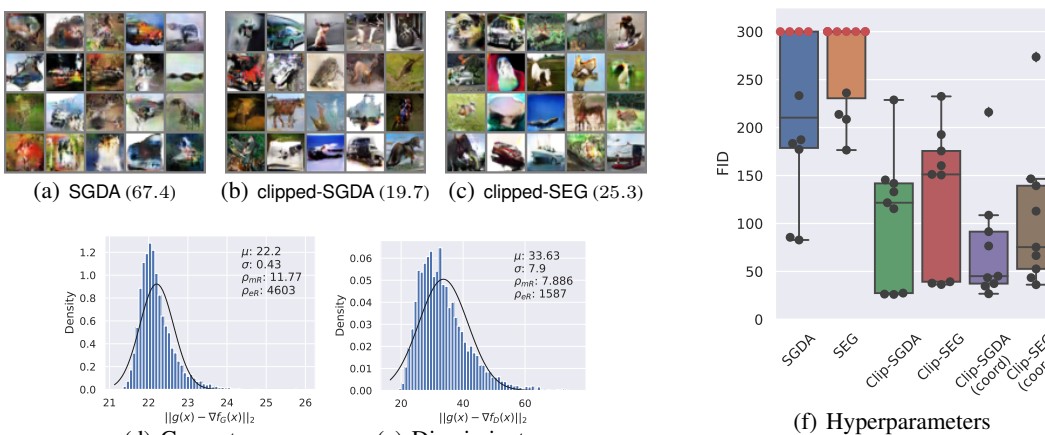

(a) SGDA (67.4)  (b) clipped-SGDA (19.7)  (c) clipped-SEG (25.3)

(d) Generator  (e) Discriminator

(f) Hyperparameters

Figure 2: (a, b, c) Random samples generated from the best CIFAR models trained (100k steps) by the specified optimizers, with their corresponding FIDs. (d, e) Gradient noise histograms for WGAN-GP upon initialization; a Gaussian was fit using maximum likelihood estimation to showcase the tail of the distribution. $p_{mR}$ is a value estimating the "heavy tailed-ness" of the distribution. (f) The best (lowest) FID scores obtained within the first 35k (out of 100k) training iterations for the same WGAN-GP model trained on CIFAR-10 when sweeping over different hyperparameters. Red points indicate that the run *diverged*, meaning that the loss becomes NaN. Note that we clip the FID for *diverged* runs to 300 to not skew the boxplot. Boxes are the quartiles of data.

weights, we iterate through 1000 steps (without parameter updates) to compute the noise norm for each minibatch (sample with replacement as in normal GAN training). We show the distributions for the generator and discriminator in Fig. 2 and also compute $p_{mR} = F_{1.5}(X)$ and $p_{eR} = F_3(X)$, where $F_\lambda(X) = P(Q_3 + \lambda(Q_3 - Q_1) < X)$, $X$ is the gradient noise distribution, and $Q_i$ is the $i^{th}$ quartile. This is a metric introduced by Jordanova and Petkova [2017] to measure how heavy-tailed a distribution is based on the distribution's quantiles, where $p_{mR}$ and $p_{eR}$ quantify "mild" and "extreme" (right side) heavy tails respectively. A normal distribution should have $p_{mR\mathcal{N}} \approx 0.0035$ and $p_{eR\mathcal{N}} \approx 1.2 \times 10^{-6}$. In all tests, we compute the ratios $\rho_{mR} = p_{mR}/p_{mR\mathcal{N}}$ and $\rho_{eR} = p_{eR}/p_{eR\mathcal{N}}$ and empirically find that we at least have mild heavy tails, and sometimes extremely heavy tails.

We train the ResNet generator on CIFAR-10 with SGDA/SEG, and clipped-SGDA/SEG. We use the default architectures and training parameters specified in Gulrajani et al. [2017] ($\lambda_{GP} = 10$, $n_{dis} = 5$, learning rate decayed linearly to 0 over 100k steps), with the exception of doubling the feature map of the generator. The clipped methods are implemented as standard gradient norm clipping, applied after computing the gradient penalty term for the critic. In addition to norm clipping, we also test coordinate-wise gradient clipping, which is more common in practice [Goodfellow et al., 2016b]. For all methods, we tune the learning rates and clipping threshold where applicable. We do not use momentum following a standard practice for GAN training [Gidel et al., 2019b].

We find that clipped methods outperform regular SGDA and SEG. In addition to helping prevent exploding gradients, clipped methods achieve a better *Fréchet inception distance* (FID) score [Heusel et al., 2017]; the best FID obtained for clipped methods is 19.65 in comparison to 67.37 for regular SGDA, both trained for 100k steps. A summary of FIDs obtained during hyperparameter tuning is shown Fig. 2, where the best FID score obtained in the first 35k iterations is drawn for each hyperparameter configuration and optimization method. At a high level, we do a log-space sweep over [2e−5, 0.2] for the learning rates, $[10^{-1}, 10]$ for the norm-clip parameter, and $[10^{-3}, 10^{-1}]$ for the coordinate clip parameter (with some exceptions) – please refer to § E for further details. We also show the evolution of the gradient noise histograms during training for SGD and clipped-SGDA in Fig. 4. Note that for regular SGD, the noise distribution remains heavy tailed and does not appear to change much throughout training. In contrast, the noise histograms for clipped-SGDA (in particular the generator) seem to develop lighter tails during training.

**StyleGAN2.** We extend our experiments to StyleGAN2, but limit our scope to clipped-SGDA with coordinate clipping as coordinate clipped-SGDA generally performs the best, and StyleGAN2 is expensive to train. We train on FFHQ downsampled to $128 \times 128$ pixels, and use the recommended StyleGAN2 hyperparameter configuration for this resolution (batch size $= 32$, $\gamma = 0.1024$, map depth $= 2$, channel multiplier $= 16384$), see further experimental details in § E. We obtain the gradient noise histograms at initialization and for the best trained clipped-SGDA from the hyperparameter sweep and once again observe heavy tails (especially in the discriminator, see Fig. 3). We find that *all* regular SGDA runs fail to learn anything meaningful, with FID scores fluctuating

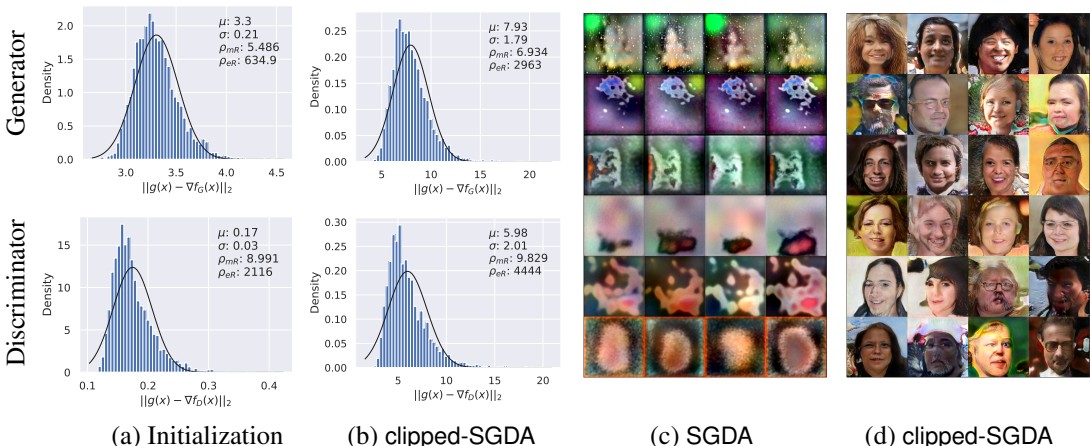

(a) Initialization      (b) clipped-SGDA      (c) SGDA      (d) clipped-SGDA

Figure 3: (a) Gradient noise histograms for StyleGAN2 at random initialization. (b) Gradient noise histograms for StyleGAN2 trained with clipped-SGDA. (c) Random samples generated from several models trained with SGDA with different learning rates (FID > 300). Each row corresponds to a different trained model, and all of our attempts to train StyleGAN2 with SGDA produced similar results. (d) Random samples generated from the best clipped-SGDA trained model (FID = 72.68).

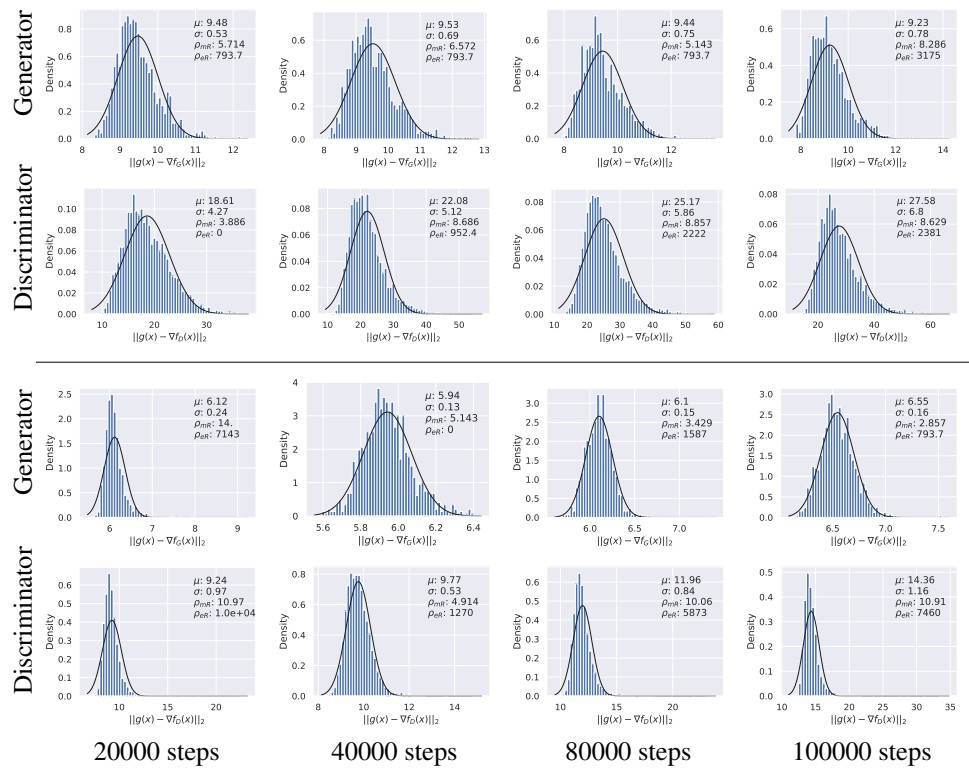

Figure 4: Evolution of gradient noise histograms for a WGAN-GP model. The top two rows are trained with SGDA, and the bottom two rows are trained with clipped-SGDA.

around 320 and only able to generate noise. In contrast, while there is a clear gap in quality when compared to what StyleGAN2 is capable of, a model trained with clipped-SGDA with appropriately set hyperparameters is able to produce images that distinctly resemble faces (see Fig. 3).

## Acknowledgments and Disclosure of Funding

This work was partially supported by a grant for research centers in the field of artificial intelligence, provided by the Analytical Center for the Government of the Russian Federation in accordance with the subsidy agreement (agreement identifier 000000D730321P5Q0002) and the agreement with the Moscow Institute of Physics and Technology dated November 1, 2021 No. 70-2021-00138. The work by P. Dvurechensky was funded by the Deutsche Forschungsgemeinschaft (DFG, German Research Foundation) under Germany's Excellence Strategy – The Berlin Mathematics Research Center MATH+ (EXC-2046/1, project ID: 390685689).

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
