# Contents

# A  Further Related Work

**Convergence in expectation.** Convergence in expectation of stochastic methods for solving VIPs is relatively well-studied in the literature. In particular, versions of SEG are studied under bounded variance [Beznosikov et al., 2020, Hsieh et al., 2020], smoothness of stochastic realizations [Mishchenko et al., 2020], and more refined assumptions unifying previously used ones [Gorbunov et al., 2022a]. Recent advances on the in-expectation convergence of SGDA are obtained in [Loizou et al., 2021, Beznosikov et al., 2022].

**Gradient clipping.** In the context of solving minimization problems, gradient clipping [Pascanu et al., 2013] and normalization [Hazan et al., 2015] are known to have a number of favorable properties such as practical robustness to the rapid changes of the loss function [Goodfellow et al., 2016a], provable convergence for structured non-smooth problems with polynomial growth Zhang et al. [2020a], Mai and Johansson [2021] and for the problems with heavy-tailed noise in convex [Nazin et al., 2019, Gorbunov et al., 2020, 2021] and non-convex cases [Zhang et al., 2020b, Cutkosky and Mehta, 2021]. Our work makes a further step towards a better understanding of gradient clipping and is the first to study the theoretical convergence of clipped first-order stochastic methods for VIPs.

**Structured non-monotonicity.** There is a noticeable growing interest of the community in studying the theoretical convergence guarantees of deterministic methods for solving VIP with non-monotone operators $F(x)$ having a certain structure, e.g., negative comonotonicty [Diakonikolas et al., 2021, Lee and Kim, 2021, Böhm, 2022], quasi-strong monotonicity [Song et al., 2020, Mertikopoulos and Zhou, 2019] and/or star-cocoercivity [Loizou et al., 2021, Gorbunov et al., 2022b,a, Beznosikov et al., 2022]. In the context of stochastic VIPs, SEG (with different extrapolation and update stepsizes) is analyzed under negative comonotonicity by Diakonikolas et al. [2021] and under quasi-strong monotonicity by Gorbunov et al. [2022a], while SGDA is studied under quasi-strong monotonicity and/or star-cocoercivity by [Loizou et al., 2021, Beznosikov et al., 2022]. These results establish in-expectation convergence rates. Our paper continues this line of works and provides the first high-probability analysis of stochastic methods for solving VIPs with structured non-monotonicity.

# B Auxiliary Results

**Useful inequalities.** For all $a, b \in \mathbb{R}^d$ and $\alpha > 0$ the following relations hold:

$$2\langle a, b \rangle = \|a\|^2 + \|b\|^2 - \|a - b\|^2, \tag{4}$$

$$\|a + b\|^2 \leq 2\|a\|^2 + 2\|b\|^2, \tag{5}$$

$$-\|a - b\|^2 \leq -\frac{1}{2}\|a\|^2 + \|b\|^2. \tag{6}$$

**Bernstein inequality.** In our proofs, we rely on the following lemma known as *Bernstein inequality for martingale differences* [Bennett, 1962, Dzhaparidze and Van Zanten, 2001, Freedman et al., 1975].

**Lemma B.1.** *Let the sequence of random variables $\{X_i\}_{i \geq 1}$ form a martingale difference sequence, i.e. $\mathbb{E}[X_i \mid X_{i-1}, \ldots, X_1] = 0$ for all $i \geq 1$. Assume that conditional variances $\sigma_i^2 \stackrel{def}{=} \mathbb{E}\left[X_i^2 \mid X_{i-1}, \ldots, X_1\right]$ exist and are bounded and assume also that there exists deterministic constant $c > 0$ such that $|X_i| \leq c$ almost surely for all $i \geq 1$. Then for all $b > 0$, $G > 0$ and $n \geq 1$*

$$\mathbb{P}\left\{\left|\sum_{i=1}^n X_i\right| > b \text{ and } \sum_{i=1}^n \sigma_i^2 \leq G\right\} \leq 2\exp\left(-\frac{b^2}{2G + 2cb/3}\right). \tag{7}$$

**Bias and variance of clipped stochastic vector.** We also use the following properties of clipped stochastic estimators from [Gorbunov et al., 2020].

**Lemma B.2** (Simplified version of Lemma F.5 from [Gorbunov et al., 2020]). *Let $X$ be a random vector in $\mathbb{R}^d$ and $\widetilde{X} = \texttt{clip}(X, \lambda)$. Then,*

$$\left\|\widetilde{X} - \mathbb{E}[\widetilde{X}]\right\| \leq 2\lambda. \tag{8}$$

*Moreover, if for some $\sigma \geq 0$*

$$\mathbb{E}[X] = x \in \mathbb{R}^d, \quad \mathbb{E}[\|X - x\|^2] \leq \sigma^2 \tag{9}$$

*and $x \leq \lambda/2$, then*

$$\left\|\mathbb{E}[\widetilde{X}] - x\right\| \leq \frac{4\sigma^2}{\lambda}, \tag{10}$$

$$\mathbb{E}\left[\left\|\widetilde{X} - x\right\|^2\right] \leq 18\sigma^2, \tag{11}$$

$$\mathbb{E}\left[\left\|\widetilde{X} - \mathbb{E}[\widetilde{X}]\right\|^2\right] \leq 18\sigma^2. \tag{12}$$

*Proof.* The proof of this lemma is identical to the original one, since Gorbunov et al. [2020] rely only on $\widetilde{X} = \texttt{clip}(X, \lambda)$ to derive (8), and to prove (10)-(12) they use only (9), $\widetilde{X} = \texttt{clip}(X, \lambda)$ and $x \leq \lambda/2$ ☐

## C  Clipped Stochastic Extragradient: Missing Proofs and Details

### C.1  Monotone Case

**Lemma C.1.** *Let Assumptions 1.1, 1.2, 1.3 hold for $Q = B_{4R}(x^*)$, where $R \geq R_0 \overset{def}{=} \|x^0 - x^*\|$, and $\gamma_1 = \gamma_2 = \gamma$, $0 < \gamma \leq 1/\sqrt{2}L$. If $x^k$ and $\widetilde{x}^k$ lie in $B_{4R}(x^*)$ for all $k = 0, 1, \ldots, K$ for some $K \geq 0$, then for all $u \in B_{4R}(x^*)$ the iterates produced by* clipped-SEG *satisfy*

$$\langle F(u), \widetilde{x}^K_{avg} - u \rangle \leq \frac{\|x^0 - u\|^2 - \|x^{K+1} - u\|^2}{2\gamma(K+1)} + \frac{\gamma}{2(K+1)} \sum_{k=0}^{K} \left( \|\theta_k\|^2 + 2\|\omega_k\|^2 \right)$$
$$+ \frac{1}{K+1} \sum_{k=0}^{K} \langle x^k - u - \gamma F(\widetilde{x}^k), \theta_k \rangle, \tag{13}$$

$$\widetilde{x}^K_{avg} \overset{def}{=} \frac{1}{K+1} \sum_{k=0}^{K} \widetilde{x}^k, \tag{14}$$

$$\theta_k \overset{def}{=} F(\widetilde{x}^k) - \widetilde{F}_{\boldsymbol{\xi}_2^k}(\widetilde{x}^k), \tag{15}$$

$$\omega_k \overset{def}{=} F(x^k) - \widetilde{F}_{\boldsymbol{\xi}_1^k}(x^k). \tag{16}$$

*Proof.* Using the update rule of clipped-SEG, for all $u \in B_{4R}(x^*)$ we obtain

$$\begin{aligned}
\|x^{k+1} - u\|^2 &= \|x^k - u\|^2 - 2\gamma\langle x^k - u, \widetilde{F}_{\boldsymbol{\xi}_2^k}(\widetilde{x}^k)\rangle + \gamma^2\|\widetilde{F}_{\boldsymbol{\xi}_2^k}(\widetilde{x}^k)\|^2 \\
&= \|x^k - u\|^2 - 2\gamma\langle x^k - u, F(\widetilde{x}^k)\rangle + 2\gamma\langle x^k - u, \theta_k\rangle \\
&\quad + \gamma^2\|F(\widetilde{x}^k)\|^2 - 2\gamma^2\langle F(\widetilde{x}^k), \theta_k\rangle + \gamma^2\|\theta_k\|^2 \\
&= \|x^k - u\|^2 - 2\gamma\langle \widetilde{x}^k - u, F(\widetilde{x}^k)\rangle - 2\gamma\langle x^k - \widetilde{x}^k, F(\widetilde{x}^k)\rangle \\
&\quad + 2\gamma\langle x^k - u - \gamma F(\widetilde{x}^k), \theta_k\rangle + \gamma^2\|F(\widetilde{x}^k)\|^2 + \gamma^2\|\theta_k\|^2 \\
&\overset{(Mon)}{\leq} \|x^k - u\|^2 - 2\gamma\langle \widetilde{x}^k - u, F(u)\rangle - 2\gamma^2\langle \widetilde{F}_{\boldsymbol{\xi}_1^k}(x^k), F(\widetilde{x}^k)\rangle \\
&\quad + 2\gamma\langle x^k - u - \gamma F(\widetilde{x}^k), \theta_k\rangle + \gamma^2\|F(\widetilde{x}^k)\|^2 + \gamma^2\|\theta_k\|^2 \\
&\overset{(4)}{=} \|x^k - u\|^2 - 2\gamma\langle \widetilde{x}^k - u, F(u)\rangle \\
&\quad + \gamma^2\|\widetilde{F}_{\boldsymbol{\xi}_1^k}(x^k) - F(\widetilde{x}^k)\|^2 - \gamma^2\|F(\widetilde{x}^k)\|^2 - \gamma^2\|\widetilde{F}_{\boldsymbol{\xi}_1^k}(x^k)\|^2 \\
&\quad + 2\gamma\langle x^k - u - \gamma F(\widetilde{x}^k), \theta_k\rangle + \gamma^2\|F(\widetilde{x}^k)\|^2 + \gamma^2\|\theta_k\|^2 \\
&\overset{(5)}{\leq} \|x^k - u\|^2 - 2\gamma\langle \widetilde{x}^k - u, F(u)\rangle \\
&\quad + 2\gamma^2\|\omega_k\|^2 + 2\gamma^2\|F(x^k) - F(\widetilde{x}^k)\|^2 - \gamma^2\|\widetilde{F}_{\boldsymbol{\xi}_1^k}(x^k)\|^2 \\
&\quad + 2\gamma\langle x^k - u - \gamma F(\widetilde{x}^k), \theta_k\rangle + \gamma^2\|\theta_k\|^2 \\
&\overset{(Lip)}{\leq} \|x^k - u\|^2 - 2\gamma\langle \widetilde{x}^k - u, F(u)\rangle - \gamma^2\left(1 - 2\gamma^2 L^2\right)\|\widetilde{F}_{\boldsymbol{\xi}_1^k}(x^k)\|^2 \\
&\quad + 2\gamma\langle x^k - u - \gamma F(\widetilde{x}^k), \theta_k\rangle + \gamma^2\|\theta_k\|^2 + 2\gamma^2\|\omega_k\|^2
\end{aligned}$$

where in the last step we additionally use $x^k - \widetilde{x}^k = \gamma\widetilde{F}_{\boldsymbol{\xi}_1^k}(x^k)$ after the application of Lipschitzness of $F$. Since $\gamma \leq 1/\sqrt{2}L$, we have $\gamma^2\left(1 - 2\gamma^2 L^2\right)\|\widetilde{F}_{\boldsymbol{\xi}_1^k}(x^k)\|^2 \geq 0$, implying

$$\begin{aligned}
2\gamma\langle F(u), \widetilde{x}^k - u\rangle &\leq \|x^k - u\|^2 - \|x^{k+1} - u\|^2 + \gamma^2\left(\|\theta_k\|^2 + 2\|\omega_k\|^2\right) \\
&\quad + 2\gamma\langle x^k - u - \gamma F(\widetilde{x}^k), \theta_k\rangle.
\end{aligned}$$

Finally, we sum up the above inequality for $k = 0, 1, \ldots, K$ and divide both sides of the result by $2\gamma(K+1)$:

$$
\begin{aligned}
\langle F(u), \widetilde{x}_{\mathtt{avg}}^K - u \rangle &= \frac{1}{K+1} \sum_{k=0}^{K} \langle F(u), \widetilde{x}^k - u \rangle \\
&\leq \frac{1}{2\gamma(K+1)} \sum_{k=0}^{K} \left( \|x^k - u\|^2 - \|x^{k+1} - u\|^2 \right) + \frac{\gamma}{2(K+1)} \sum_{k=0}^{K} \|\theta_k\|^2 \\
&\quad + \frac{1}{K+1} \sum_{k=0}^{K} \langle x^k - u - \gamma F(\widetilde{x}^k), \theta_k \rangle + \frac{\gamma}{K+1} \sum_{k=0}^{K} \|\omega_k\|^2 \\
&= \frac{\|x^0 - u\|^2 - \|x^{K+1} - u\|^2}{2\gamma(K+1)} + \frac{\gamma}{2(K+1)} \sum_{k=0}^{K} \left( \|\theta_k\|^2 + 2\|\omega_k\|^2 \right) \\
&\quad + \frac{1}{K+1} \sum_{k=0}^{K} \langle x^k - u - \gamma F(\widetilde{x}^k), \theta_k \rangle.
\end{aligned}
$$

This concludes the proof. $\qquad\square$

**Theorem C.1.** *Let Assumptions 1.1, 1.2, 1.3 hold for $Q = B_{4R}(x^*)$, where $R \geq R_0 \overset{def}{=} \|x^0 - x^*\|$, and*[12] $\gamma_1 = \gamma_2 = \gamma$,

$$
0 < \gamma \leq \frac{1}{160L \ln \frac{6(K+1)}{\beta}}, \tag{17}
$$

$$
\lambda_{1,k} = \lambda_{2,k} \equiv \lambda = \frac{R}{20\gamma \ln \frac{6(K+1)}{\beta}}, \tag{18}
$$

$$
m_{1,k} = m_{2,k} \equiv m \geq \max\left\{ 1, \frac{10800(K+1)\gamma^2\sigma^2 \ln \frac{6(K+1)}{\beta}}{R^2} \right\}, \tag{19}
$$

*for some $K \geq 0$ and $\beta \in (0, 1]$ such that $\ln \frac{6(K+1)}{\beta} \geq 1$. Then, after $K$ iterations the iterates produced by* clipped-SEG *with probability at least $1 - \beta$ satisfy*

$$
Gap_R(\widetilde{x}_{avg}^K) \leq \frac{9R^2}{2\gamma(K+1)}, \tag{20}
$$

*where $\widetilde{x}_{avg}^K$ is defined in (14).*

*Proof.* We introduce new notation: $R_k = \|x^k - x^*\|$ for all $k \geq 0$. The proof is based on deriving via induction that $R_k^2 \leq \tilde{C}R^2$ for some numerical constant $\tilde{C} > 0$. In particular, for each $k = 0, \ldots, K+1$ we define probability event $E_k$ as follows: inequalities

$$
\underbrace{\max_{u \in B_R(x^*)} \left\{ \|x^0 - u\|^2 + 2\gamma \sum_{l=0}^{t-1} \langle x^l - u - \gamma F(\widetilde{x}^l), \theta_l \rangle + \gamma^2 \sum_{l=0}^{t-1} \left( \|\theta_l\|^2 + 2\|\omega_l\|^2 \right) \right\}}_{A_t} \leq 9R^2, \tag{21}
$$

$$
\left\| \gamma \sum_{l=0}^{t-1} \theta_l \right\| \leq R \tag{22}
$$

hold for $t = 0, 1, \ldots, k$ simultaneously. Our goal is to prove that $\mathbb{P}\{E_k\} \geq 1 - \frac{k\beta}{(K+1)}$ for all $k = 0, 1, \ldots, K+1$. We use the induction to show this statement. For $k = 0$ the statement

---

[12]In this and further results, we have relatively large numerical constants in the conditions on step-sizes, batch-sizes, and clipping levels. However, our main goal is deriving results in terms of $\mathcal{O}(\cdot)$, where numerical constants are not taken into consideration. Although it is possible to significantly improve the dependence on numerical factors, we do not do it for the sake of proofs' simplicity.

is trivial since $\|x^0 - u\|^2 \leq 2\|x^0 - x^*\|^2 + 2\|x^* - u\|^2 \leq 4R^2 \leq 9R^2$ and $\|\gamma \sum_{l=0}^{k-1} \theta_l\| = 0$ for any $u \in B_R(x^*)$. Next, assume that the statement holds for $k = T - 1 \leq K$, i.e., we have $\mathbb{P}\{E_{T-1}\} \geq 1 - {(T-1)\beta}/{(K+1)}$. We need to prove that $\mathbb{P}\{E_T\} \geq 1 - {T\beta}/{(K+1)}$. First of all, we show that probability event $E_{T-1}$ implies $R_t \leq 3R$ for all $t = 0, 1, \ldots, T$. For $t = 0$ we already proved it. Next, assume that we have $R_t \leq 3R$ for all $t = 0, 1, \ldots, t'$, where $t' < T$. Then, for all $t = 0, 1, \ldots, t'$ we have

$$
\begin{aligned}
\|\widetilde{x}^t - x^*\| &= \|x^t - x^* - \gamma \widetilde{F}_{\boldsymbol{\xi}_1^t}(x^t)\| \leq \|x^t - x^*\| + \gamma \|\widetilde{F}_{\boldsymbol{\xi}_1^t}(x^t)\| \\
&\leq \|x^t - x^*\| + \gamma\lambda \overset{(18)}{\leq} 3R + \frac{R}{20 \ln \frac{6(K+1)}{\beta}} \leq 4R,
\end{aligned}
\tag{23}
$$

i.e., $\widetilde{x}^t \in B_{4R}(x^*)$. This means that the assumptions of Lemma C.1 hold and we have that probability event $E_{T-1}$ implies

$$
\begin{aligned}
\max_{u \in B_R(x^*)} &\left\{ 2\gamma(t'+1)\langle F(u), \widetilde{x}_{\mathtt{avg}}^{t'} - u\rangle + \|x^{t'+1} - u\|^2 \right\} \\
&\leq \max_{u \in B_R(x^*)} \left\{ \|x^0 - u\|^2 + 2\gamma \sum_{l=0}^{t'-1} \langle x^l - u - \gamma F(\widetilde{x}^l), \theta_l\rangle \right\} \\
&\qquad\qquad + \gamma^2 \sum_{l=0}^{t'-1} \left( \|\theta_l\|^2 + 2\|\omega_l\|^2 \right) \\
&\overset{(21)}{\leq} 9R^2,
\end{aligned}
$$

meaning that

$$
\begin{aligned}
\|x^{t'+1} - x^*\|^2 &\leq \max_{u \in B_R(x^*)} \left\{ 2\gamma(t'+1)\langle F(u), \widetilde{x}_{\mathtt{avg}}^{t'} - u\rangle + \|x^{t'+1} - u\|^2 \right\} \\
&\leq 9R^2,
\end{aligned}
$$

i.e., $R_{t'+1} \leq 3R$. That is, we proved that probability event $E_{T-1}$ implies $R_t \leq 3R$ and

$$
\max_{u \in B_R(x^*)} \left\{ 2\gamma(t+1)\langle F(u), \widetilde{x}_{\mathtt{avg}}^t - u\rangle + \|x^{t+1} - u\|^2 \right\} \leq 9R^2
\tag{24}
$$

for all $t = 0, 1, \ldots, T$. Moreover, in view of (23) $E_{T-1}$ also implies that $\|\widetilde{x}^t - x^*\| \leq 4R$ for all $t = 0, 1, \ldots, T$. Using this, we derive that $E_{T-1}$ implies

$$
\begin{aligned}
\|x^t - x^* - \gamma F(\widetilde{x}^t)\| &\leq \|x^t - x^*\| + \gamma\|F(\widetilde{x}^t)\| \overset{(\text{Lip})}{\leq} 3R + \gamma L \|\widetilde{x}^t - x^*\| \\
&\overset{(23)}{\leq} 3R + 4R\gamma L \overset{(17)}{\leq} 5R,
\end{aligned}
\tag{25}
$$

for all $t = 0, 1, \ldots, T$. Consider random vectors

$$
\eta_t = \begin{cases} x^t - x^* - \gamma F(\widetilde{x}^t), & \text{if } \|x^t - x^* - \gamma F(\widetilde{x}^t)\| \leq 5R, \\ 0, & \text{otherwise,} \end{cases}
$$

for all $t = 0, 1, \ldots, T$. We notice that $\eta_t$ is bounded with probability 1:

$$
\|\eta_t\| \leq 5R
\tag{26}
$$

for all $t = 0, 1, \ldots, T$. Moreover, in view of (25), probability event $E_{T-1}$ implies $\eta_t = x^t - x^* - \gamma F(\widetilde{x}^t)$ for all $t = 0, 1, \ldots, T$. Therefore, $E_{T-1}$ implies

$$
\begin{aligned}
A_T &= \max_{u \in B_R(x^*)} \left\{ \|x^0 - u\|^2 + 2\gamma \sum_{l=0}^{T-1} \langle x^* - u, \theta_l \rangle \right\} \\
&\quad + 2\gamma \sum_{l=0}^{T-1} \langle x^l - x^* - \gamma F(\widetilde{x}^l), \theta_l \rangle + \gamma^2 \sum_{l=0}^{T-1} \left( \|\theta_l\|^2 + 2\|\omega_l\|^2 \right) \\
&\leq 4R^2 + 2\gamma \max_{u \in B_R(x^*)} \left\{ \left\langle x^* - u, \sum_{l=0}^{T-1} \theta_l \right\rangle \right\} \\
&\quad + 2\gamma \sum_{l=0}^{T-1} \langle \eta_l, \theta_l \rangle + \gamma^2 \sum_{l=0}^{T-1} \left( \|\theta_l\|^2 + 2\|\omega_l\|^2 \right) \\
&= 4R^2 + 2\gamma R \left\| \sum_{l=0}^{T-1} \theta_l \right\| + 2\gamma \sum_{l=0}^{T-1} \langle \eta_l, \theta_l \rangle + \gamma^2 \sum_{l=0}^{T-1} \left( \|\theta_l\|^2 + 2\|\omega_l\|^2 \right),
\end{aligned}
$$

where $A_T$ is defined in (21). To continue our derivation we introduce new notation:

$$
\theta_l^u \stackrel{\text{def}}{=} \mathbb{E}_{\boldsymbol{\xi}_2^l} \left[ \widetilde{F}_{\boldsymbol{\xi}_2^l}(\widetilde{x}^l) \right] - \widetilde{F}_{\boldsymbol{\xi}_2^l}(\widetilde{x}^l), \quad \theta_l^b \stackrel{\text{def}}{=} F(\widetilde{x}^l) - \mathbb{E}_{\boldsymbol{\xi}_2^l} \left[ \widetilde{F}_{\boldsymbol{\xi}_2^l}(\widetilde{x}^l) \right], \tag{27}
$$

$$
\omega_l^u \stackrel{\text{def}}{=} \mathbb{E}_{\boldsymbol{\xi}_1^l} \left[ \widetilde{F}_{\boldsymbol{\xi}_1^l}(x^l) \right] - \widetilde{F}_{\boldsymbol{\xi}_1^l}(x^l), \quad \omega_l^b \stackrel{\text{def}}{=} F(x^l) - \mathbb{E}_{\boldsymbol{\xi}_1^l} \left[ \widetilde{F}_{\boldsymbol{\xi}_1^l}(x^l) \right], \tag{28}
$$

for all $l = 0, \ldots, T-1$. By definition we have $\theta_l = \theta_l^u + \theta_l^b$, $\omega_l = \omega_l^u + \omega_l^b$ for all $l = 0, \ldots, T-1$. Using the introduced notation, we continue our derivation as follows: $E_{T-1}$ implies

$$
\begin{aligned}
A_T &\stackrel{(5)}{\leq} 4R^2 + 2\gamma R \left\| \sum_{l=0}^{T-1} \theta_l \right\| + \underbrace{2\gamma \sum_{l=0}^{T-1} \langle \eta_l, \theta_l^u \rangle}_{\text{①}} + \underbrace{2\gamma \sum_{l=0}^{T-1} \langle \eta_l, \theta_l^b \rangle}_{\text{②}} \\
&\quad + \underbrace{2\gamma^2 \sum_{l=0}^{T-1} \left( \mathbb{E}_{\boldsymbol{\xi}_2^l} \left[ \|\theta_l^u\|^2 \right] + 2\mathbb{E}_{\boldsymbol{\xi}_1^l} \left[ \|\omega_l^u\|^2 \right] \right)}_{\text{③}} \\
&\quad + \underbrace{2\gamma^2 \sum_{l=0}^{T-1} \left( \|\theta_l^u\|^2 + 2\|\omega_l^u\|^2 - \mathbb{E}_{\boldsymbol{\xi}_2^l} \left[ \|\theta_l^u\|^2 \right] - 2\mathbb{E}_{\boldsymbol{\xi}_1^l} \left[ \|\omega_l^u\|^2 \right] \right)}_{\text{④}} \\
&\quad + \underbrace{2\gamma^2 \sum_{l=0}^{T-1} \left( \|\theta_l^b\|^2 + 2\|\omega_l^b\|^2 \right)}_{\text{⑤}} \tag{29}
\end{aligned}
$$

The rest of the proof is based on deriving good enough upper bounds for $2\gamma R \left\| \sum_{l=0}^{T-1} \theta_l \right\|$, ①, ②, ③, ④, ⑤, i.e., we want to prove that $2\gamma R \left\| \sum_{l=0}^{T-1} \theta_l \right\| + ① + ② + ③ + ④ + ⑤ \leq 5R^2$ with high probability.

Before we move on, we need to derive some useful inequalities for operating with $\theta_l^u, \theta_l^b, \omega_l^u, \omega_l^b$. First of all, Lemma B.2 implies that

$$
\|\theta_l^u\| \leq 2\lambda, \quad \|\omega_l^u\| \leq 2\lambda \tag{30}
$$

for all $l = 0, 1, \ldots, T - 1$. Next, since $\{\xi_1^{i,l}\}_{i=1}^m$, $\{\xi_2^{i,l}\}_{i=1}^m$ are independently sampled from $\mathcal{D}$, we have $\mathbb{E}_{\boldsymbol{\xi}_1^l}[F_{\boldsymbol{\xi}_1^l}(x^l)] = F(x^l)$, $\mathbb{E}_{\boldsymbol{\xi}_2^l}[F_{\boldsymbol{\xi}_2^l}(\widetilde{x}^l)] = F(\widetilde{x}^l)$, and

$$\mathbb{E}_{\boldsymbol{\xi}_1^l}\left[\|F_{\boldsymbol{\xi}_1^l}(x^l) - F(x^l)\|^2\right] = \frac{1}{m^2}\sum_{i=1}^m \mathbb{E}_{\xi_1^{i,l}}\left[\|F_{\xi_1^{i,l}}(x^l) - F(x^l)\|^2\right] \overset{(1)}{\leq} \frac{\sigma^2}{m},$$

$$\mathbb{E}_{\boldsymbol{\xi}_2^l}\left[\|F_{\boldsymbol{\xi}_2^l}(\widetilde{x}^l) - F(\widetilde{x}^l)\|^2\right] = \frac{1}{m^2}\sum_{i=1}^m \mathbb{E}_{\xi_2^{i,l}}\left[\|F_{\xi_2^{i,l}}(\widetilde{x}^l) - F(\widetilde{x}^l)\|^2\right] \overset{(1)}{\leq} \frac{\sigma^2}{m},$$

for all $l = 0, 1, \ldots, T - 1$. Moreover, probability event $E_{T-1}$ implies

$$\|F(x^l)\| \overset{(\text{Lip})}{\leq} L\|x^l - x^*\| \leq 3LR \overset{(17)}{\leq} \frac{R}{40\gamma \ln \frac{6(K+1)}{\beta}} \overset{(18)}{=} \frac{\lambda}{2},$$

$$\|F(\widetilde{x}^l)\| \overset{(\text{Lip})}{\leq} L\|\widetilde{x}^l - x^*\| \overset{(23)}{\leq} 4LR \overset{(17)}{\leq} \frac{R}{40\gamma \ln \frac{6(K+1)}{\beta}} \overset{(18)}{=} \frac{\lambda}{2}$$

for all $l = 0, 1, \ldots, T - 1$. Therefore, in view of Lemma B.2, $E_{T-1}$ implies that

$$\|\theta_l^b\| \leq \frac{4\sigma^2}{m\lambda}, \quad \|\omega_l^b\| \leq \frac{4\sigma^2}{m\lambda}, \tag{31}$$

$$\mathbb{E}_{\boldsymbol{\xi}_2^l}\left[\|\theta_l\|^2\right] \leq \frac{18\sigma^2}{m}, \quad \mathbb{E}_{\boldsymbol{\xi}_1^l}\left[\|\omega_l\|^2\right] \leq \frac{18\sigma^2}{m}, \tag{32}$$

$$\mathbb{E}_{\boldsymbol{\xi}_2^l}\left[\|\theta_l^u\|^2\right] \leq \frac{18\sigma^2}{m}, \quad \mathbb{E}_{\boldsymbol{\xi}_1^l}\left[\|\omega_l^u\|^2\right] \leq \frac{18\sigma^2}{m}, \tag{33}$$

for all $l = 0, 1, \ldots, T - 1$.

**Upper bound for ①.**  Since $\mathbb{E}_{\boldsymbol{\xi}_2^l}[\theta_l^u] = 0$, we have

$$\mathbb{E}_{\boldsymbol{\xi}_2^l}\left[2\gamma\langle\eta_l, \theta_l^u\rangle\right] = 0.$$

Next, the summands in ① are bounded with probability 1:

$$|2\gamma\langle\eta_l, \theta_l^u\rangle| \leq 2\gamma\|\eta_l\| \cdot \|\theta_l^u\| \overset{(26),(30)}{\leq} 20\gamma R\lambda \overset{(18)}{=} \frac{R^2}{\ln \frac{6(K+1)}{\beta}} \overset{\text{def}}{=} c. \tag{34}$$

Moreover, these summands have bounded conditional variances $\sigma_l^2 \overset{\text{def}}{=} \mathbb{E}_{\boldsymbol{\xi}_2^l}\left[4\gamma^2\langle\eta_l, \theta_l^u\rangle^2\right]$:

$$\sigma_l^2 \leq \mathbb{E}_{\boldsymbol{\xi}_2^l}\left[4\gamma^2\|\eta_l\|^2 \cdot \|\theta_l^u\|^2\right] \overset{(26)}{\leq} 100\gamma^2 R^2 \mathbb{E}_{\boldsymbol{\xi}_2^l}\left[\|\theta_l^u\|^2\right]. \tag{35}$$

That is, sequence $\{2\gamma\langle\eta_l, \theta_l^u\rangle\}_{l\geq 0}$ is a bounded martingale difference sequence having bounded conditional variances $\{\sigma_l^2\}_{l\geq 0}$. Applying the Bernstein's inequality (Lemma B.1) with $X_l = 2\gamma\langle\eta_l, \theta_l^u\rangle$, $c$ defined in (34), $b = R^2$, $G = \frac{R^4}{6\ln \frac{6(K+1)}{\beta}}$, we get that

$$\mathbb{P}\left\{|①| > R^2 \text{ and } \sum_{l=0}^{T-1}\sigma_l^2 \leq \frac{R^4}{6\ln \frac{6(K+1)}{\beta}}\right\} \leq 2\exp\left(-\frac{b^2}{2G + 2cb/3}\right) = \frac{\beta}{3(K+1)}.$$

In other words, $\mathbb{P}\{E_①\} \geq 1 - \frac{\beta}{3(K+1)}$, where probability event $E_①$ is defined as

$$E_① = \left\{\text{either} \quad \sum_{l=0}^{T-1}\sigma_l^2 > \frac{R^4}{6\ln \frac{6(K+1)}{\beta}} \quad \text{or} \quad |①| \leq R^2\right\}. \tag{36}$$

Moreover, we notice here that probability event $E_{T-1}$ implies that

$$\sum_{l=0}^{T-1}\sigma_l^2 \overset{(35)}{\leq} 100\gamma^2 R^2 \sum_{l=0}^{T-1}\mathbb{E}_{\boldsymbol{\xi}_2^l}\left[\|\theta_l^u\|^2\right] \overset{(33),T\leq K+1}{\leq} \frac{1800(K+1)\gamma^2 R^2\sigma^2}{m} \overset{(19)}{\leq} \frac{R^4}{6\ln \frac{6(K+1)}{\beta}}. \tag{37}$$

**Upper bound for ②.** Probability event $E_{T-1}$ implies

$$② \quad \leq \quad 2\gamma \sum_{l=0}^{T-1} \|\eta_l\| \cdot \|\theta_l^b\| \quad \overset{(26),(31),T \leq K+1}{\leq} \quad \frac{40(K+1)\gamma R\sigma^2}{m\lambda}$$

$$\overset{(18)}{=} \quad \frac{40(K+1)\gamma^2\sigma^2 \ln \frac{6(K+1)}{\beta}}{m} \quad \overset{(19)}{\leq} \quad R^2. \tag{38}$$

**Upper bound for ③.** Probability event $E_{T-1}$ implies

$$2\gamma^2 \sum_{l=0}^{T-1} \mathbb{E}_{\boldsymbol{\xi}_2^l}[\|\theta_l^u\|^2] \quad \overset{(32),T \leq K+1}{\leq} \quad \frac{36\gamma^2(K+1)\sigma^2}{m} \quad \overset{(19)}{\leq} \quad \frac{1}{12}R^2, \tag{39}$$

$$4\gamma^2 \sum_{l=0}^{T-1} \mathbb{E}_{\boldsymbol{\xi}_1^l}[\|\omega_l^u\|^2] \quad \overset{(32),T \leq K+1}{\leq} \quad \frac{72\gamma^2(K+1)\sigma^2}{m} \quad \overset{(19)}{\leq} \quad \frac{1}{12}R^2, \tag{40}$$

$$③ \quad \overset{(39),(40)}{\leq} \quad \frac{1}{6}R^2. \tag{41}$$

**Upper bound for ④.** First of all,

$$2\gamma^2 \mathbb{E}_{\boldsymbol{\xi}_1^l, \boldsymbol{\xi}_2^l} \left[ \|\theta_l^u\|^2 + 2\|\omega_l^u\|^2 - \mathbb{E}_{\boldsymbol{\xi}_2^l}\left[\|\theta_l^u\|^2\right] - 2\mathbb{E}_{\boldsymbol{\xi}_1^l}\left[\|\omega_l^u\|^2\right] \right] = 0.$$

Next, the summands in ④ are bounded with probability 1:

$$2\gamma^2 \left| \|\theta_l^u\|^2 + 2\|\omega_l^u\|^2 - \mathbb{E}_{\boldsymbol{\xi}_2^l}\left[\|\theta_l^u\|^2\right] - 2\mathbb{E}_{\boldsymbol{\xi}_1^l}\left[\|\omega_l^u\|^2\right] \right| \quad \leq \quad 2\gamma^2\|\theta_l^u\|^2 + 2\gamma^2\mathbb{E}_{\boldsymbol{\xi}_2^l}\left[\|\theta_l^u\|^2\right]$$

$$+ 4\gamma^2\|\omega_l^u\|^2 + 4\gamma^2\mathbb{E}_{\boldsymbol{\xi}_1^l}\left[\|\omega_l^u\|^2\right]$$

$$\overset{(30)}{\leq} \quad 48\gamma^2\lambda^2$$

$$\overset{(18)}{\leq} \quad \frac{R^2}{6\ln\frac{6(K+1)}{\beta}} \overset{\text{def}}{=} c. \tag{42}$$

Moreover, these summands have bounded conditional variances $\widetilde{\sigma}_l^2 \overset{\text{def}}{=}$ $4\gamma^4 \mathbb{E}_{\boldsymbol{\xi}_1^l, \boldsymbol{\xi}_2^l} \left[ \left| \|\theta_l^u\|^2 + 2\|\omega_l^u\|^2 - \mathbb{E}_{\boldsymbol{\xi}_2^l}\left[\|\theta_l^u\|^2\right] - 2\mathbb{E}_{\boldsymbol{\xi}_1^l}\left[\|\omega_l^u\|^2\right] \right|^2 \right]$:

$$\widetilde{\sigma}_l^2 \quad \overset{(42)}{\leq} \quad \frac{\gamma^2 R^2}{3\ln\frac{6(K+1)}{\beta}} \mathbb{E}_{\boldsymbol{\xi}_1^l, \boldsymbol{\xi}_2^l} \left[ \left| \|\theta_l^u\|^2 + 2\|\omega_l^u\|^2 - \mathbb{E}_{\boldsymbol{\xi}_2^l}\left[\|\theta_l^u\|^2\right] - 2\mathbb{E}_{\boldsymbol{\xi}_1^l}\left[\|\omega_l^u\|^2\right] \right| \right]$$

$$\leq \quad \frac{2\gamma^2 R^2}{3\ln\frac{6(K+1)}{\beta}} \mathbb{E}_{\boldsymbol{\xi}_1^l, \boldsymbol{\xi}_2^l} \left[ \|\theta_l^u\|^2 + 2\|\omega_l^u\|^2 \right]. \tag{43}$$

That is, sequence $\left\{ 2\gamma^2 \left( \|\theta_l^u\|^2 + 2\|\omega_l^u\|^2 - \mathbb{E}_{\boldsymbol{\xi}_2^l}\left[\|\theta_l^u\|^2\right] - 2\mathbb{E}_{\boldsymbol{\xi}_1^l}\left[\|\omega_l^u\|^2\right] \right) \right\}_{l \geq 0}$ is a bounded martingale difference sequence having bounded conditional variances $\{\widetilde{\sigma}_l^2\}_{l \geq 0}$. Applying the Bernstein's inequality (Lemma B.1) with $X_l = 2\gamma^2 \left( \|\theta_l^u\|^2 + 2\|\omega_l^u\|^2 - \mathbb{E}_{\boldsymbol{\xi}_2^l}\left[\|\theta_l^u\|^2\right] - 2\mathbb{E}_{\boldsymbol{\xi}_1^l}\left[\|\omega_l^u\|^2\right] \right)$, $c$ defined in (42), $b = \frac{1}{6}R^2$, $G = \frac{R^4}{216\ln\frac{6(K+1)}{\beta}}$, we get that

$$\mathbb{P}\left\{ |④| > \frac{1}{6}R^2 \text{ and } \sum_{l=0}^{T-1} \widetilde{\sigma}_l^2 \leq \frac{R^4}{216\ln\frac{6(K+1)}{\beta}} \right\} \leq 2\exp\left( -\frac{b^2}{2G + 2cb/3} \right) = \frac{\beta}{3(K+1)}.$$

In other words, $\mathbb{P}\{E_④\} \geq 1 - \frac{\beta}{3(K+1)}$, where probability event $E_④$ is defined as

$$E_④ = \left\{ \text{either} \quad \sum_{l=0}^{T-1} \widetilde{\sigma}_l^2 > \frac{R^4}{216\ln\frac{6(K+1)}{\beta}} \quad \text{or} \quad |④| \leq \frac{1}{6}R^2 \right\}. \tag{44}$$

Moreover, we notice here that probability event $E_{T-1}$ implies that

$$\sum_{l=0}^{T-1} \widetilde{\sigma}_l^2 \overset{(43)}{\leq} \frac{2\gamma^2 R^2}{3\ln\frac{6(K+1)}{\beta}} \sum_{l=0}^{T-1} \mathbb{E}_{\boldsymbol{\xi}_1^l, \boldsymbol{\xi}_2^l}\left[\|\theta_l^u\|^2 + 2\|\omega_l^u\|^2\right]$$
$$\overset{(33),T\leq K+1}{\leq} \frac{36(K+1)\gamma^2 R^2\sigma^2}{m} \overset{(19)}{\leq} \frac{R^4}{216\ln\frac{6(K+1)}{\beta}}. \tag{45}$$

**Upper bound for ⑤.**   Probability event $E_{T-1}$ implies

$$⑤ = 2\gamma^2\sum_{l=0}^{T-1}\left(\|\theta_l^b\|^2 + 2\|\omega_l^b\|^2\right) \overset{(31),T\leq K+1}{\leq} \frac{96\gamma^2\sigma^4(K+1)}{m^2\lambda^2}$$
$$\overset{(18)}{=} \frac{38400\gamma^4\sigma^4(K+1)\ln^2\frac{6(K+1)}{\beta}}{m^2 R^2} \overset{(19)}{\leq} \frac{1}{6}R^2. \tag{46}$$

**Upper bound for $2\gamma R\left\|\sum_{l=0}^{T-1}\theta_l\right\|$.**   To handle this term, we introduce new notation:

$$\zeta_l = \begin{cases} \gamma\sum_{r=0}^{l-1}\theta_r, & \text{if } \left\|\gamma\sum_{r=0}^{l-1}\theta_r\right\| \leq R, \\ 0, & \text{otherwise} \end{cases}$$

for $l = 1, 2, \ldots, T-1$. By definition, we have

$$\|\zeta_l\| \leq R. \tag{47}$$

Therefore, in view of (22), probability event $E_{T-1}$ implies

$$2\gamma R\left\|\sum_{l=0}^{T-1}\theta_l\right\| = 2R\sqrt{\gamma^2\left\|\sum_{l=0}^{T-1}\theta_l\right\|^2}$$
$$= 2R\sqrt{\gamma^2\sum_{l=0}^{T-1}\|\theta_l\|^2 + 2\gamma\sum_{l=0}^{T-1}\left\langle\gamma\sum_{r=0}^{l-1}\theta_r, \theta_l\right\rangle}$$
$$= 2R\sqrt{\gamma^2\sum_{l=0}^{T-1}\|\theta_l\|^2 + 2\gamma\sum_{l=0}^{T-1}\langle\zeta_l, \theta_l\rangle}$$
$$\overset{(27)}{\leq} 2R\sqrt{③ + ④ + ⑤ + 2\gamma\underbrace{\sum_{l=0}^{T-1}\langle\zeta_l, \theta_l^u\rangle}_{⑥} + 2\gamma\underbrace{\sum_{l=0}^{T-1}\langle\zeta_l, \theta_l^b\rangle}_{⑦}}. \tag{48}$$

Following similar steps as before, we bound ⑥ and ⑦.

**Upper bound for ⑥.**   Since $\mathbb{E}_{\boldsymbol{\xi}_2^l}[\theta_l^u] = 0$, we have

$$\mathbb{E}_{\boldsymbol{\xi}_2^l}\left[2\gamma\langle\zeta_l, \theta_l^u\rangle\right] = 0.$$

Next, the summands in ④ are bounded with probability 1:

$$|2\gamma\langle\zeta_l, \theta_l^u\rangle| \leq 2\gamma\|\eta_l\|\cdot\|\theta_l^u\| \overset{(47),(30)}{\leq} 4\gamma R\lambda \overset{(18)}{\leq} \frac{R^2}{4\ln\frac{6(K+1)}{\beta}} \overset{\text{def}}{=} c. \tag{49}$$

Moreover, these summands have bounded conditional variances $\hat{\sigma}_l^2 \overset{\text{def}}{=} \mathbb{E}_{\boldsymbol{\xi}_2^l}\left[4\gamma^2\langle\zeta_l, \theta_l^u\rangle^2\right]$:

$$\hat{\sigma}_l^2 \leq \mathbb{E}_{\boldsymbol{\xi}_2^l}\left[4\gamma^2\|\zeta_l\|^2\cdot\|\theta_l^u\|^2\right] \overset{(47)}{\leq} 4\gamma^2 R^2\mathbb{E}_{\boldsymbol{\xi}_2^l}\left[\|\theta_l^u\|^2\right]. \tag{50}$$

That is, sequence $\{2\gamma\langle\zeta_l, \theta_l^u\rangle\}_{l\geq 0}$ is a bounded martingale difference sequence having bounded conditional variances $\{\hat{\sigma}_l^2\}_{l\geq 0}$. Applying Bernstein's inequality (Lemma B.1) with $X_l = 2\gamma\langle\zeta_l, \theta_l^u\rangle$, $c$ defined in (34), $b = \frac{R^2}{4}$, $G = \frac{R^4}{96\ln\frac{6(K+1)}{\beta}}$, we get that

$$\mathbb{P}\left\{|⑤| > \frac{1}{4}R^2 \text{ and } \sum_{l=0}^{T-1}\hat{\sigma}_l^2 \leq \frac{R^4}{96\ln\frac{4(K+1)}{\beta}}\right\} \leq 2\exp\left(-\frac{b^2}{2G + \frac{2cb}{3}}\right) = \frac{\beta}{3(K+1)}.$$

In other words, $\mathbb{P}\{E_⑥\} \geq 1 - \frac{\beta}{3(K+1)}$, where probability event $E_⑥$ is defined as

$$E_⑥ = \left\{\text{either} \quad \sum_{l=0}^{T-1}\hat{\sigma}_l^2 > \frac{R^4}{96\ln\frac{6(K+1)}{\beta}} \quad \text{or} \quad |⑤| \leq \frac{1}{4}R^2\right\}. \tag{51}$$

Moreover, we notice here that probability event $E_{T-1}$ implies that

$$\sum_{l=0}^{T-1}\hat{\sigma}_l^2 \overset{(50)}{\leq} 4\gamma^2 R^2 \sum_{l=0}^{T-1}\mathbb{E}_{\boldsymbol{\xi}_2^l}\left[\|\theta_l^u\|^2\right] \overset{(33),T\leq K+1}{\leq} \frac{72(K+1)\gamma^2 R^2\sigma^2}{m} \overset{(19)}{\leq} \frac{R^4}{96\ln\frac{6(K+1)}{\beta}}. \tag{52}$$

**Upper bound for ⑦.**   Probability event $E_{T-1}$ implies

$$⑦ \quad \leq \quad 2\gamma\sum_{l=0}^{T-1}\|\zeta_l\|\cdot\|\theta_l^b\| \overset{(47),(31),T\leq K+1}{\leq} \frac{8(K+1)\gamma R\sigma^2}{m\lambda}$$

$$\overset{(18)}{=} \quad \frac{160(K+1)\gamma^2\sigma^2\ln\frac{6(K+1)}{\beta}}{m} \overset{(19)}{\leq} \frac{1}{4}R^2. \tag{53}$$

**Final derivation.**   Putting all bounds together, we get that $E_{T-1}$ implies

$$A_T \overset{(29)}{\leq} 4R^2 + 2\gamma R\left\|\sum_{l=0}^{T-1}\theta_l\right\| + ① + ② + ③ + ④ + ⑤,$$

$$2\gamma R\left\|\sum_{l=0}^{T-1}\theta_l\right\| \overset{(48)}{\leq} 2R\sqrt{③ + ④ + ⑤ + ⑥ + ⑦},$$

$$② \overset{(38)}{\leq} R^2, \quad ③ \overset{(41)}{\leq} \frac{1}{6}R^2, \quad ⑤ \overset{(46)}{\leq} \frac{1}{6}R^2, \quad ⑦ \overset{(53)}{\leq} \frac{1}{4}R^2,$$

$$\sum_{l=0}^{T-1}\sigma_l^2 \overset{(37)}{\leq} \frac{R^4}{6\ln\frac{6(K+1)}{\beta}}, \quad \sum_{l=0}^{T-1}\tilde{\sigma}_l^2 \overset{(45)}{\leq} \frac{R^4}{216\ln\frac{6(K+1)}{\beta}}, \quad \sum_{l=0}^{T-1}\hat{\sigma}_l^2 \overset{(52)}{\leq} \frac{R^4}{96\ln\frac{6(K+1)}{\beta}}.$$

Moreover, in view of (36), (44), (51), and our induction assumption, we have

$$\mathbb{P}\{E_{T-1}\} \geq 1 - \frac{(T-1)\beta}{K+1},$$

$$\mathbb{P}\{E_①\} \geq 1 - \frac{\beta}{3(K+1)}, \quad \mathbb{P}\{E_④\} \geq 1 - \frac{\beta}{3(K+1)}, \quad \mathbb{P}\{E_⑥\} \geq 1 - \frac{\beta}{3(K+1)},$$

where probability events $E_①$, $E_④$, and $E_⑥$ are defined as

$$E_① = \left\{\text{either} \quad \sum_{l=0}^{T-1}\sigma_l^2 > \frac{R^4}{6\ln\frac{6(K+1)}{\beta}} \quad \text{or} \quad |①| \leq R^2\right\},$$

$$E_④ = \left\{\text{either} \quad \sum_{l=0}^{T-1}\tilde{\sigma}_l^2 > \frac{R^4}{216\ln\frac{6(K+1)}{\beta}} \quad \text{or} \quad |④| \leq \frac{1}{6}R^2\right\},$$

$$E_⑥ = \left\{\text{either} \quad \sum_{l=0}^{T-1}\hat{\sigma}_l^2 > \frac{R^4}{96\ln\frac{6(K+1)}{\beta}} \quad \text{or} \quad |⑥| \leq \frac{1}{4}R^2\right\}.$$

Putting all of these inequalities together, we obtain that probability event $E_{T-1} \cap E_① \cap E_④ \cap E_⑥$ implies

$$\left\| \gamma \sum_{l=0}^{T-1} \theta_l \right\| \leq \sqrt{\frac{1}{6}R^2 + \frac{1}{6}R^2 + \frac{1}{6}R^2 + \frac{1}{4}R^2 + \frac{1}{4}R^2} = R, \tag{54}$$

$$\begin{aligned} A_T &\leq 4R^2 + 2R\sqrt{\frac{1}{6}R^2 + \frac{1}{6}R^2 + \frac{1}{6}R^2 + \frac{1}{4}R^2 + \frac{1}{4}R^2} \\ &\quad + R^2 + R^2 + \frac{1}{6}R^2 + \frac{1}{6}R^2 + \frac{1}{6}R^2 \\ &\leq 9R^2. \end{aligned} \tag{55}$$

Moreover, union bound for the probability events implies

$$\mathbb{P}\{E_T\} \geq \mathbb{P}\{E_{T-1} \cap E_① \cap E_④ \cap E_⑥\} = 1 - \mathbb{P}\{\overline{E}_{T-1} \cup \overline{E}_① \cup \overline{E}_④ \cup \overline{E}_⑥\} \geq 1 - \frac{T\beta}{K+1}. \tag{56}$$

This is exactly what we wanted to prove (see the paragraph after inequalities (21), (22)). Therefore, for all $k = 0, 1, \ldots, K+1$ we have $\mathbb{P}\{E_k\} \geq 1 - k\beta/(K+1)$., i.e., for $k = K+1$ we have that with probability at least $1 - \beta$ inequality

$$\begin{aligned} \mathtt{Gap}_R(\widetilde{x}_{\mathtt{avg}}^K) &= \max_{u \in B_R(x^*)} \left\{ \langle F(u), \widetilde{x}_{\mathtt{avg}}^K - u \rangle \right\} \\ &\leq \frac{1}{2\gamma(K+1)} \max_{u \in B_R(x^*)} \left\{ 2\gamma(K+1)\langle F(u), \widetilde{x}_{\mathtt{avg}}^t - u \rangle + \|x^{K+1} - u\|^2 \right\} \\ &\overset{(24)}{\leq} \frac{9R^2}{2\gamma(K+1)} \end{aligned}$$

holds. This concludes the proof. $\qquad\square$

**Corollary C.1.** *Let the assumptions of Theorem C.1 hold. Then, the following statements hold.*

1. **Large stepsize/large batch.** *The choice of stepsize and batchsize*

$$\gamma = \frac{1}{160L \ln\frac{6(K+1)}{\beta}}, \quad m = \max\left\{1, \frac{27(K+1)\sigma^2}{64L^2R^2 \ln\frac{6(K+1)}{\beta}}\right\} \tag{57}$$

*satisfies conditions (17) and (19). With such choice of $\gamma, m$, and the choice of $\lambda$ as in (18), the iterates produced by* clipped-SEG *after $K$ iterations with probability at least $1 - \beta$ satisfy*

$$\mathtt{Gap}_R(\widetilde{x}_{\mathtt{avg}}^K) \leq \frac{720LR^2 \ln\frac{6(K+1)}{\beta}}{K+1}. \tag{58}$$

*In particular, to guarantee $\mathtt{Gap}_R(\widetilde{x}_{\mathtt{avg}}^K) \leq \varepsilon$ with probability at least $1 - \beta$ for some $\varepsilon > 0$* clipped-SEG *requires,*

$$\mathcal{O}\left(\frac{LR^2}{\varepsilon} \ln\left(\frac{LR^2}{\varepsilon\beta}\right)\right) \text{ iterations,} \tag{59}$$

$$\mathcal{O}\left(\max\left\{\frac{LR^2}{\varepsilon}, \frac{\sigma^2R^2}{\varepsilon^2}\right\} \ln\left(\frac{LR^2}{\varepsilon\beta}\right)\right) \text{ oracle calls.} \tag{60}$$

2. **Small stepsize/small batch.** *The choice of stepsize and batchsize*

$$\gamma = \min\left\{\frac{1}{160L \ln\frac{6(K+1)}{\beta}}, \frac{R}{60\sigma\sqrt{3(K+1)\ln\frac{6(K+1)}{\beta}}}\right\}, \quad m = 1 \tag{61}$$

*satisfies conditions (17) and (19). With such choice of $\gamma, m$, and the choice of $\lambda$ as in (18), the iterates produced by* clipped-SEG *after $K$ iterations with probability at least $1 - \beta$ satisfy*

$$\mathtt{Gap}_R(\widetilde{x}_{\mathtt{avg}}^K) \leq \max\left\{\frac{720LR^2 \ln\frac{6(K+1)}{\beta}}{K+1}, \frac{270\sigma R\sqrt{\ln\frac{6(K+1)}{\beta}}}{\sqrt{K+1}}\right\}. \tag{62}$$

*In particular, to guarantee $\mathtt{Gap}_R(\widetilde{x}_{\mathtt{avg}}^K) \le \varepsilon$ with probability at least $1 - \beta$ for some $\varepsilon > 0$,* clipped-SEG *requires*

$$\mathcal{O}\left(\max\left\{\frac{LR^2}{\varepsilon}\ln\left(\frac{LR^2}{\varepsilon\beta}\right), \frac{\sigma^2 R^2}{\varepsilon^2}\ln\left(\frac{\sigma^2 R^2}{\varepsilon^2\beta}\right)\right\}\right) \quad \textit{iterations/oracle calls.} \qquad (63)$$

*Proof.*      1. **Large stepsize/large batch.** First of all, we verify that the choice of $\gamma$ and $m$ from (57) satisfies conditions (17) and (19): (17) trivially holds and (19) holds since

$$m = \max\left\{1, \frac{27(K+1)\sigma^2}{64L^2R^2\ln\frac{6(K+1)}{\beta}}\right\} = \max\left\{1, \frac{10800(K+1)\gamma^2\sigma^2\ln\frac{6(K+1)}{\beta}}{R^2}\right\}.$$

Therefore, applying Theorem C.1, we derive that with probability at least $1 - \beta$

$$\mathtt{Gap}_R(\widetilde{x}_{\mathtt{avg}}^K) \le \frac{9R^2}{2\gamma(K+1)} \overset{(57)}{=} \frac{720LR^2\ln\frac{4(K+1)}{\beta}}{K+1}.$$

To guarantee $\mathtt{Gap}_R(\widetilde{x}_{\mathtt{avg}}^K) \le \varepsilon$, we choose $K$ in such a way that the right-hand side of the above inequality is smaller than $\varepsilon$ that gives

$$K = \mathcal{O}\left(\frac{LR^2}{\varepsilon}\ln\left(\frac{LR^2}{\varepsilon\beta}\right)\right).$$

The total number of oracle calls equals

$$\begin{aligned}
2m(K+1) &\overset{(57)}{=} 2\max\left\{K+1, \frac{27(K+1)^2\sigma^2}{64L^2R^2\ln\frac{6(K+1)}{\beta}}\right\} \\
&= \mathcal{O}\left(\max\left\{\frac{LR^2}{\varepsilon}, \frac{\sigma^2 R^2}{\varepsilon^2}\right\}\ln\left(\frac{LR^2}{\varepsilon\beta}\right)\right).
\end{aligned}$$

2. **Small stepsize/small batch.** First of all, we verify that the choice of $\gamma$ and $m$ from (57) satisfies conditions (17) and (19):

$$\begin{aligned}
\gamma &= \min\left\{\frac{1}{160L\ln\frac{6(K+1)}{\beta}}, \frac{R}{60\sigma\sqrt{3(K+1)\ln\frac{6(K+1)}{\beta}}}\right\} \le \frac{1}{160L\ln\frac{6(K+1)}{\beta}}, \\
m &= 1 \overset{(61)}{\ge} \frac{10800(K+1)\gamma^2\sigma^2\ln\frac{6(K+1)}{\beta}}{R^2}.
\end{aligned}$$

Therefore, applying Theorem C.1, we derive that with probability at least $1 - \beta$

$$\begin{aligned}
\mathtt{Gap}_R(\widetilde{x}_{\mathtt{avg}}^K) &\le \frac{9R^2}{2\gamma(K+1)} \\
&\overset{(61)}{=} \max\left\{\frac{720LR^2\ln\frac{6(K+1)}{\beta}}{K+1}, \frac{270\sigma R\sqrt{\ln\frac{6(K+1)}{\beta}}}{\sqrt{K+1}}\right\}.
\end{aligned}$$

To guarantee $\mathtt{Gap}_R(\widetilde{x}_{\mathtt{avg}}^K) \le \varepsilon$, we choose $K$ in such a way that the right-hand side of the above inequality is smaller than $\varepsilon$ that gives

$$K = \mathcal{O}\left(\max\left\{\frac{LR^2}{\varepsilon}\ln\left(\frac{LR^2}{\varepsilon\beta}\right), \frac{\sigma^2 R^2}{\varepsilon^2}\ln\left(\frac{\sigma^2 R^2}{\varepsilon^2\beta}\right)\right\}\right).$$

The total number of oracle calls equals $2m(K+1) = 2(K+1)$.

□

## C.2 Star-Negative Comonotone Case

**Lemma C.2.** *Let Assumptions [1.2], [1.4] hold for $Q = B_{3R}(x^*) = \{x \in \mathbb{R}^d \mid \|x - x^*\| \leq 3R\}$, where $R \geq R_0 \stackrel{def}{=} \|x^0 - x^*\|$, and $\gamma_2 + 2\rho < \gamma_1 \leq 1/(2L)$. If $x^k$ and $\widetilde{x}^k$ lie in $B_{3R_0}(x^*)$ for all $k = 0, 1, \ldots, K$ for some $K \geq 0$, then the iterates produced by* clipped-SEG *satisfy*

$$
\begin{aligned}
\frac{\gamma_1 \gamma_2}{4(K+1)} \sum_{k=0}^{K} \|F(x^k)\|^2 &\leq \frac{\|x^0 - x^*\|^2 - \|x^{K+1} - x^*\|^2}{K+1} \\
&\quad + \frac{1}{K+1} \sum_{k=0}^{K} \left(\gamma_2^2 \|\omega_k\|^2 + 3\gamma_1 \gamma_2 \|\omega_k\|^2\right) \\
&\quad + \frac{2\gamma_2}{K+1} \sum_{k=0}^{K} \langle x^k - x^* - \gamma_2 F(\widetilde{x}^k), \theta_k \rangle
\end{aligned}
\tag{64}
$$

*where $\theta_k, \omega_k$ are defined in* (15)*,* (16)*.*

*Proof.* Using the update rule of clipped-SEG, we obtain

$$
\begin{aligned}
\|x^{k+1} - x^*\|^2 &= \|x^k - x^*\|^2 - 2\gamma_2 \langle x^k - x^*, \widetilde{F}_{\boldsymbol{\xi}_2^k}(\widetilde{x}^k)\rangle + \gamma_2^2 \|\widetilde{F}_{\boldsymbol{\xi}_2^k}(\widetilde{x}^k)\|^2 \\
&= \|x^k - x^*\|^2 - 2\gamma_2 \langle x^k - x^*, F(\widetilde{x}^k)\rangle + 2\gamma_2 \langle x^k - x^*, \theta_k\rangle \\
&\quad + \gamma_2^2 \|F(\widetilde{x}^k)\|^2 - 2\gamma_2 \langle F(\widetilde{x}^k), \theta_k\rangle + \gamma_2^2 \|\theta_k\|^2 \\
&= \|x^k - x^*\|^2 - 2\gamma_2 \langle \widetilde{x}^k - x^*, F(\widetilde{x}^k)\rangle - 2\gamma_2 \langle x^k - \widetilde{x}^k, F(\widetilde{x}^k)\rangle \\
&\quad + 2\gamma_2 \langle x^k - x^* - \gamma_2 F(\widetilde{x}^k), \theta_k\rangle + \gamma_2^2 \|F(\widetilde{x}^k)\|^2 + \gamma^2 \|\theta_k\|^2 \\
&\stackrel{(\text{SNC})}{\leq} \|x^k - x^*\|^2 + 2\gamma_2 \rho \|\widetilde{F}(\widetilde{x}^k)\|^2 - 2\gamma_1 \gamma_2 \langle \widetilde{F}_{\boldsymbol{\xi}_1^k}(x^k), F(\widetilde{x}^k)\rangle \\
&\quad + 2\gamma_2 \langle x^k - x^* - \gamma_2 F(\widetilde{x}^k), \theta_k\rangle + \gamma_2^2 \|F(\widetilde{x}^k)\|^2 + \gamma_2^2 \|\theta_k\|^2 \\
&\stackrel{(4)}{=} \|x^k - x^*\|^2 + \gamma_1 \gamma_2 \|\widetilde{F}_{\boldsymbol{\xi}_1^k}(x^k) - F(\widetilde{x}^k)\|^2 - \gamma_1 \gamma_2 \|F(\widetilde{x}^k)\|^2 - \gamma_1 \gamma_2 \|\widetilde{F}_{\boldsymbol{\xi}_1^k}(x^k)\|^2 \\
&\quad + 2\gamma_2 \langle x^k - x^* - \gamma_2 F(\widetilde{x}^k), \theta_k\rangle + \gamma_2 (2\rho + \gamma_2) \|F(\widetilde{x}^k)\|^2 + \gamma_2^2 \|\theta_k\|^2 \\
&\stackrel{(5)}{\leq} \|x^k - x^*\|^2 + 2\gamma_1 \gamma_2 \|\omega_k\|^2 + 2\gamma_1 \gamma_2 \|F(x^k) - F(\widetilde{x}^k)\|^2 - \gamma_1 \gamma_2 \|\widetilde{F}_{\boldsymbol{\xi}_1^k}(x^k)\|^2 \\
&\quad + 2\gamma_2 \langle x^k - x^* - \gamma_2 F(\widetilde{x}^k), \theta_k\rangle + \gamma_2 (2\rho + \gamma_2 - \gamma_1) \|F(\widetilde{x}^k)\|^2 + \gamma_2^2 \|\theta_k\|^2 \\
&\stackrel{(\text{Lip})}{\leq} \|x^k - x^*\|^2 - \gamma_1 \gamma_2 \left(1 - 2\gamma_1^2 L^2\right) \|\widetilde{F}_{\boldsymbol{\xi}_1^k}(x^k)\|^2 \\
&\quad + 2\gamma_2 \langle x^k - x^* - \gamma_2 F(\widetilde{x}^k), \theta_k\rangle + \gamma_2^2 \|\theta_k\|^2 + 2\gamma_1 \gamma_2 \|\omega_k\|^2,
\end{aligned}
$$

where in the last step we additionally use $x^k - \widetilde{x}^k = \gamma_1 \widetilde{F}_{\boldsymbol{\xi}_1^k}(x^k)$ after the application of Lipschitzness of $F$ and we use our assumption on $\gamma_1, \gamma_2, \rho$: $\gamma_2 + 2\rho \leq \gamma_1$. Since $\gamma_1 \leq 1/(2L)$, we have $\gamma_1 \gamma_2 \left(1 - 2\gamma_1^2 L^2\right) \|\widetilde{F}_{\boldsymbol{\xi}_1^k}(x^k)\|^2 \geq 0$ and, using (6) with $\alpha = 1$, we derive

$$
\begin{aligned}
\|x^{k+1} - x^*\|^2 &\leq \|x^k - x^*\|^2 - \frac{\gamma_1 \gamma_2}{2} \left(1 - 2\gamma_1^2 L^2\right) \|F(x^k)\|^2 \\
&\quad + 2\gamma_2 \langle x^k - x^* - \gamma_2 F(\widetilde{x}^k), \theta_k\rangle + \gamma_2^2 \|\theta_k\|^2 + 2\gamma_1 \gamma_2 \left(\frac{3}{2} - \gamma_1^2 L^2\right) \|\omega_k\|^2.
\end{aligned}
$$

Rearranging the terms and using $\frac{3}{2} - \gamma_1^2 L^2 \leq \frac{3}{2}$, $1 - 2\gamma_1^2 L^2 \geq 1/2$, we derive

$$
\begin{aligned}
\frac{\gamma_1 \gamma_2}{4} \|F(x^k)\|^2 &\leq \|x^k - x^*\|^2 - \|x^{k+1} - x^*\|^2 + \left(\gamma_2^2 \|\theta_k\|^2 + 3\gamma_1 \gamma_2 \|\omega_k\|^2\right) \\
&\quad + 2\gamma_2 \langle x^k - x^* - 2\gamma_2 F(\widetilde{x}^k), \theta_k\rangle.
\end{aligned}
$$

Finally, we sum up the above inequality for $k = 0, 1, \ldots, K$ and divide both sides of the result by $(K+1)$:

$$
\begin{aligned}
\frac{\gamma_1 \gamma_2}{4(K+1)} \sum_{k=0}^{K} \|F(x^k)\|^2 \quad \leq \quad & \frac{1}{K+1} \sum_{k=0}^{K} \left( \|x^k - x^*\|^2 - \|x^{k+1} - x^*\|^2 \right) + \frac{\gamma_2^2}{K+1} \sum_{k=0}^{K} \|\theta_k\|^2 \\
& + \frac{2\gamma_2}{K+1} \sum_{k=0}^{K} \langle x^k - x^* - \gamma_2 F(\widetilde{x}^k), \theta_k \rangle + \frac{3\gamma_1 \gamma_2}{K+1} \sum_{k=0}^{K} \|\omega_k\|^2 \\
= \quad & \frac{\|x^0 - x^*\|^2 - \|x^{K+1} - x^*\|^2}{K+1} \\
& + \frac{1}{K+1} \sum_{k=0}^{K} \left( \gamma_2^2 \|\theta_k\|^2 + 3\gamma_1 \gamma_2 \|\omega_k\|^2 \right) \\
& + \frac{2\gamma_2}{K+1} \sum_{k=0}^{K} \langle x^k - x^* - \gamma_2 F(\widetilde{x}^k), \theta_k \rangle.
\end{aligned}
$$

This finishes the proof. $\qquad \square$

**Theorem C.2.** *Let Assumptions 1.1, 1.2, 1.4 hold for $Q = B_{3R}(x^*)$, where $R \geq R_0 \overset{def}{=} \|x^0 - x^*\|$, and*

$$
\gamma_2 + 2\rho \leq \gamma_1 \leq \frac{1}{160 L \ln \frac{6(K+1)}{\beta}}, \tag{65}
$$

$$
\lambda_{1,k} \equiv \lambda_1 = \frac{R}{20\gamma_1 \ln \frac{6(K+1)}{\beta}}, \quad \lambda_{1,k} \equiv \lambda_2 = \frac{R}{20\gamma_2 \ln \frac{6(K+1)}{\beta}}, \tag{66}
$$

$$
m_{1,k} \equiv m_1 \geq \max \left\{ 1, \frac{216 \max\{\gamma_1 \gamma_2 (K+1), \sqrt{\gamma_1^3 \gamma_2 (K+1)} \ln \frac{6(K+1)}{\beta}\} \sigma^2}{R^2} \right\}, \tag{67}
$$

$$
m_{2,k} \equiv m_2 \geq \max \left\{ 1, \frac{3240(K+1)\gamma_2^2 \sigma^2 \ln \frac{6(K+1)}{\beta}}{R^2} \right\}, \tag{68}
$$

*for some $K \geq 0$ and $\beta \in (0,1]$ such that $\ln \frac{6(K+1)}{\beta} \geq 1$. Then, after $K$ iterations the iterates produced by* clipped-SEG *with probability at least $1 - \beta$ satisfy*

$$
\frac{1}{K+1} \sum_{k=0}^{K} \|F(x^k)\|^2 \leq \frac{36 R^2}{\gamma_1 \gamma_2 (K+1)}. \tag{69}
$$

*Proof.* As in the proof of Theorem C.1, we use the following notation: $R_k = \|x^k - x^*\|^2$, $k \geq 0$. We will derive (69) by induction. In particular, for each $k = 0, \ldots, K+1$ we define probability event $E_k$ as follows: inequalities

$$
R_t^2 \leq 4R^2 \tag{70}
$$

hold for $t = 0, 1, \ldots, k$ simultaneously. Our goal is to prove that $\mathbb{P}\{E_k\} \geq 1 - k\beta/(K+1)$ for all $k = 0, 1, \ldots, K+1$. We use the induction to show this statement. For $k = 0$ the statement is trivial since $R_0^2 \leq 4R^2$ by definition. Next, assume that the statement holds for $k = T - 1 \leq K$, i.e., we have $\mathbb{P}\{E_{T-1}\} \geq 1 - (T-1)\beta/(K+1)$. We need to prove that $\mathbb{P}\{E_T\} \geq 1 - T\beta/(K+1)$. First of all, since $R_t^2 \leq 4R^2$, we have $x^t \in B_{2R}(x^*)$. Operator $F$ is $L$-Lipschitz on $B_{3R}(x^*)$. Therefore, probability event $E_{T-1}$ implies

$$
\|F(x^t)\| \quad \leq \quad L\|x^t - x^*\| \overset{(70)}{\leq} 2LR \overset{(65),(66)}{\leq} \frac{\lambda_1}{2}. \tag{71}
$$

and

$$
\|\omega_t\|^2 \quad \overset{(5)}{\leq} \quad 2\|\widetilde{F}_{\boldsymbol{\xi}_1}(x^t)\|^2 + 2\|F(x^t)\|^2 \overset{(71)}{\leq} \frac{5}{2}\lambda_1^2 \overset{(66)}{\leq} \frac{R^2}{4\gamma_1^2} \tag{72}
$$

for all $t = 0, 1, \ldots, T - 1$.

Next, we show that probability event $E_{T-1}$ implies $\|\tilde{x}^t - x^*\| \leq 3R$ and derive useful inequalities related to $\theta_t$ for all $t = 0, 1, \ldots, T - 1$. Indeed, due to Lipschitzness of $F$ probability event $E_{T-1}$ implies

$$
\begin{aligned}
\|\tilde{x}^t - x^*\|^2 &= \|x^t - x^* - \gamma_1 \widetilde{F}_{\xi_1}(x^t)\|^2 \overset{(5)}{\leq} 2\|x^t - x^*\|^2 + 2\gamma_1^2 \|\widetilde{F}_{\xi_1}(x^t)\|^2 \\
&\overset{(5)}{\leq} 2R_t^2 + 4\gamma_1^2 \|F(x^t)\|^2 + 4\gamma_1^2 \|\omega_t\|^2 \\
&\overset{(\text{Lip})}{\leq} 2(1 + 2\gamma_1^2 L^2)R_t^2 + 4\gamma_1^2 \|\omega_t\|^2 \\
&\overset{(65),(72)}{\leq} 7R^2 \leq 9R^2
\end{aligned}
\tag{73}
$$

and

$$
\|F(\tilde{x}^t)\| \leq L\|\tilde{x}^t - x^*\| \leq \sqrt{7}LR \overset{(65),(66)}{\leq} \frac{\lambda_2}{2}
\tag{74}
$$

for all $t = 0, 1, \ldots, T - 1$.

That is, $E_{T-1}$ implies that $x^t, \tilde{x}^t \in B_{3R}(x^*)$ for all $t = 0, 1, \ldots, T - 1$. Applying Lemma C.2, we get that probability event $E_{T-1}$ implies

$$
\begin{aligned}
\frac{\gamma_1 \gamma_2}{4T} \sum_{l=0}^{T-1} \|F(x^l)\|^2 &\leq \frac{R^2 - R_T^2}{T} + \frac{2\gamma_2}{T} \sum_{l=0}^{T-1} \langle x^l - x^* - \gamma_2 F(\tilde{x}^l), \theta_l \rangle \\
&\quad + \frac{1}{T} \sum_{l=0}^{T-1} \left( \gamma_2^2 \|\theta_l\|^2 + 3\gamma_1 \gamma_2 \|\omega_l\|^2 \right)
\end{aligned}
\tag{75}
$$

$$
R_T^2 \leq R^2 + 2\gamma_2 \sum_{l=0}^{T-1} \langle x^l - x^* - \gamma_2 F(\tilde{x}^l), \theta_l \rangle + \sum_{l=0}^{T-1} \left( \gamma_2^2 \|\theta_l\|^2 + 3\gamma_1 \gamma_2 \|\omega_l\|^2 \right).
$$

To estimate the sums in the right-hand side, we introduce new vectors:

$$
\eta_t = \begin{cases} x^t - x^* - \gamma_2 F(\tilde{x}^t), & \text{if } \|x^t - x^* - \gamma_2 F(\tilde{x}^t)\| \leq \sqrt{7}(1 + \gamma_2 L)R, \\ 0, & \text{otherwise,} \end{cases}
\tag{76}
$$

for $t = 0, 1, \ldots, T - 1$. First of all, we point out that vectors $\zeta_t$ and $\eta_t$ are bounded with probability 1, i.e., with probability 1

$$
\|\eta_t\| \leq \sqrt{7}(1 + \gamma_2 L)R
\tag{77}
$$

for all $t = 0, 1, \ldots, T - 1$. Next, we notice that $E_{T-1}$ implies

$$
\begin{aligned}
\|x^t - x^* - \gamma_2 F(\tilde{x}^t)\| &\leq \|x^t - x^*\| + \gamma_2 \|F(\tilde{x}^t)\| \\
&\overset{(73),(74)}{\leq} \sqrt{7}(1 + \gamma_2 L)R
\end{aligned}
$$

for $t = 0, 1, \ldots, T - 1$, i.e., probability event $E_{T-1}$ implies $\eta_t = x^t - x^* - \gamma_2 F(\tilde{x}^t)$ for all $t = 0, 1, \ldots, T - 1$. Therefore, $E_{T-1}$ implies

$$
R_T^2 \leq R^2 + 2\gamma_2 \sum_{l=0}^{T-1} \langle \eta_l, \theta_l \rangle + \sum_{l=0}^{T-1} \left( \gamma_2^2 \|\theta_l\|^2 + 3\gamma_1 \gamma_2 \|\omega_l\|^2 \right).
$$

As in the monotone case, to continue the derivation, we introduce vectors $\theta_l^u, \theta_l^b, \omega_l^u, \omega_l^b$ defined as

$$
\theta_l^u \overset{\text{def}}{=} \mathbb{E}_{\xi_2^l} \left[ \widetilde{F}_{\xi_2^l}(\tilde{x}^l) \right] - \widetilde{F}_{\xi_2^l}(\tilde{x}^l), \quad \theta_l^b \overset{\text{def}}{=} F(\tilde{x}^l) - \mathbb{E}_{\xi_2^l} \left[ \widetilde{F}_{\xi_2^l}(\tilde{x}^l) \right],
\tag{78}
$$

$$
\omega_l^u \overset{\text{def}}{=} \mathbb{E}_{\xi_1^l} \left[ \widetilde{F}_{\xi_1^l}(x^l) \right] - \widetilde{F}_{\xi_1^l}(x^l), \quad \theta_l^b \overset{\text{def}}{=} F(x^l) - \mathbb{E}_{\xi_1^l} \left[ \widetilde{F}_{\xi_1^l}(x^l) \right],
\tag{79}
$$

for all $l = 0, \ldots, T-1$. By definition we have $\theta_l = \theta_l^u + \theta_l^b$, $\omega_l = \omega_l^u + \omega_l^b$ for all $l = 0, \ldots, T-1$. Using the introduced notation, we continue our derivation as follows: $E_{T-1}$ implies

$$R_T^2 \overset{(5)}{\leq} R^2 + \underbrace{2\gamma_2 \sum_{l=0}^{T-1} \langle \eta_l, \theta_l^u \rangle}_{①} + \underbrace{2\gamma_2 \sum_{l=0}^{T-1} \langle \eta_l, \theta_l^b \rangle}_{②} + \underbrace{2\gamma_2^2 \sum_{l=0}^{T-1} \mathbb{E}_{\boldsymbol{\xi}_2^l} \left[ \|\theta_l^u\|^2 \right]}_{③}$$

$$+ \underbrace{2\gamma_2^2 \sum_{l=0}^{T-1} \left( \|\theta_l^u\|^2 - \mathbb{E}_{\boldsymbol{\xi}_2^l} \left[ \|\theta_l^u\|^2 \right] \right)}_{④} + \underbrace{2\gamma_2^2 \sum_{l=0}^{T-1} \|\theta_l^b\|^2}_{⑤} + \underbrace{6\gamma_1\gamma_2 \sum_{l=0}^{T-1} \mathbb{E}_{\boldsymbol{\xi}_1^l} \left[ \|\omega_l^u\|^2 \right]}_{⑥}$$

$$+ \underbrace{6\gamma_1\gamma_2 \sum_{l=0}^{T-1} \left( \|\omega_l^u\|^2 - \mathbb{E}_{\boldsymbol{\xi}_1^l} \left[ \|\omega_l^u\|^2 \right] \right)}_{⑦} + \underbrace{6\gamma_1\gamma_2 \sum_{l=0}^{T-1} \|\omega_l^b\|^2}_{⑧}. \qquad (80)$$

The rest of the proof is based on deriving good enough upper bounds for ①, ②, ③, ④, ⑤, ⑥, ⑦, ⑧, i.e., we want to prove that ① + ② + ③ + ④ + ⑤ + ⑥ + ⑦ + ⑧ $\leq 8R^2$ with high probability.

Before we move on, we need to derive some useful inequalities for operating with $\theta_l^u, \theta_l^b, \omega_l^u, \omega_l^b$. First of all, Lemma B.2 implies that

$$\|\theta_l^u\| \leq 2\lambda_2, \quad \|\omega_l^u\| \leq 2\lambda_1 \qquad (81)$$

for all $l = 0, 1, \ldots, T-1$. Next, since $\{\xi_1^{i,l}\}_{i=1}^{m_1}, \{\xi_2^{i,l}\}_{i=1}^{m_2}$ are independently sampled from $\mathcal{D}$, we have $\mathbb{E}_{\boldsymbol{\xi}_1^l}[F_{\boldsymbol{\xi}_1^l}(x^l)] = F(x^l)$, $\mathbb{E}_{\boldsymbol{\xi}_2^l}[F_{\boldsymbol{\xi}_2^l}(\widetilde{x}^l)] = F(\widetilde{x}^l)$, and

$$\mathbb{E}_{\boldsymbol{\xi}_1^l} \left[ \|F_{\boldsymbol{\xi}_1^l}(x^l) - F(x^l)\|^2 \right] = \frac{1}{m_1^2} \sum_{i=1}^{m_1} \mathbb{E}_{\xi_1^{i,l}} \left[ \|F_{\xi_1^{i,l}}(x^l) - F(x^l)\|^2 \right] \overset{(1)}{\leq} \frac{\sigma^2}{m_1},$$

$$\mathbb{E}_{\boldsymbol{\xi}_2^l} \left[ \|F_{\boldsymbol{\xi}_2^l}(\widetilde{x}^l) - F(\widetilde{x}^l)\|^2 \right] = \frac{1}{m_2^2} \sum_{i=1}^{m_2} \mathbb{E}_{\xi_2^{i,l}} \left[ \|F_{\xi_2^{i,l}}(\widetilde{x}^l) - F(\widetilde{x}^l)\|^2 \right] \overset{(1)}{\leq} \frac{\sigma^2}{m_2},$$

for all $l = 0, 1, \ldots, T-1$. Moreover, as we already derived, probability event $E_{T-1}$ implies that $\|F(x^l)\| \leq \lambda_l/2$ and $\|F(\widetilde{x}^l)\| \leq \lambda_l/2$ for all $l = 0, 1, \ldots, T-1$ (see (71) and (74)). Therefore, in view of Lemma B.2, $E_{T-1}$ implies that

$$\|\theta_l^b\| \leq \frac{4\sigma^2}{m_2\lambda_2}, \quad \|\omega_l^b\| \leq \frac{4\sigma^2}{m_1\lambda_1}, \qquad (82)$$

$$\mathbb{E}_{\boldsymbol{\xi}_2^l} \left[ \|\theta_l\|^2 \right] \leq \frac{18\sigma^2}{m_2}, \quad \mathbb{E}_{\boldsymbol{\xi}_1^l} \left[ \|\omega_l\|^2 \right] \leq \frac{18\sigma^2}{m_1}, \qquad (83)$$

$$\mathbb{E}_{\boldsymbol{\xi}_2^l} \left[ \|\theta_l^u\|^2 \right] \leq \frac{18\sigma^2}{m_2}, \quad \mathbb{E}_{\boldsymbol{\xi}_1^l} \left[ \|\omega_l^u\|^2 \right] \leq \frac{18\sigma^2}{m_1}, \qquad (84)$$

for all $l = 0, 1, \ldots, T-1$.

**Upper bound for ①.** Since $\mathbb{E}_{\boldsymbol{\xi}_2^l}[\theta_l^u] = 0$, we have

$$\mathbb{E}_{\boldsymbol{\xi}_2^l} \left[ 2\gamma_2 \langle \eta_l, \theta_l^u \rangle \right] = 0.$$

Next, the summands in ① are bounded with probability 1:

$$|2\gamma_2 \langle \eta_l, \theta_l^u \rangle| \leq 2\gamma_2 \|\eta_l\| \cdot \|\theta_l^u\| \overset{(77),(81)}{\leq} 4\sqrt{7}\gamma_2(1 + \gamma_2 L)R\lambda_l \overset{(65),(66)}{\leq} \frac{R^2}{\ln \frac{6(K+1)}{\beta}} \overset{\text{def}}{=} c. \qquad (85)$$

Moreover, these summands have bounded conditional variances $\sigma_l^2 \overset{\text{def}}{=} \mathbb{E}_{\boldsymbol{\xi}_2^l} \left[ 4\gamma_2^2 \langle \eta_l, \theta_l^u \rangle^2 \right]$:

$$\sigma_l^2 \leq \mathbb{E}_{\boldsymbol{\xi}_2^l} \left[ 4\gamma_2^2 \|\eta_l\|^2 \cdot \|\theta_l^u\|^2 \right] \overset{(77)}{\leq} 28\gamma_2^2(1 + \gamma_2 L)^2 R^2 \mathbb{E}_{\boldsymbol{\xi}_2^l} \left[ \|\theta_l^u\|^2 \right] \overset{(65)}{\leq} 30\gamma_2^2 R^2 \mathbb{E}_{\boldsymbol{\xi}_2^l} \left[ \|\theta_l^u\|^2 \right]. \qquad (86)$$

That is, sequence $\{2\gamma_2\langle\eta_l, \theta_l^u\rangle\}_{l\geq 0}$ is a bounded martingale difference sequence having bounded conditional variances $\{\sigma_l^2\}_{l\geq 0}$. Applying Bernstein's inequality (Lemma B.1) with $X_l = 2\gamma_2\langle\eta_l, \theta_l^u\rangle$, $c$ defined in (85), $b = R^2$, $G = \frac{R^4}{6\ln\frac{6(K+1)}{\beta}}$, we get that

$$\mathbb{P}\left\{|\text{①}| > R^2 \text{ and } \sum_{l=0}^{T-1}\sigma_l^2 \leq \frac{R^4}{6\ln\frac{6(K+1)}{\beta}}\right\} \leq 2\exp\left(-\frac{b^2}{2G + 2cb/3}\right) = \frac{\beta}{3(K+1)}.$$

In other words, $\mathbb{P}\{E_{\text{①}}\} \geq 1 - \frac{\beta}{3(K+1)}$, where probability event $E_{\text{①}}$ is defined as

$$E_{\text{①}} = \left\{\text{either} \quad \sum_{l=0}^{T-1}\sigma_l^2 > \frac{R^4}{6\ln\frac{6(K+1)}{\beta}} \quad \text{or} \quad |\text{①}| \leq R^2\right\}. \tag{87}$$

Moreover, we notice here that probability event $E_{T-1}$ implies that

$$\sum_{l=0}^{T-1}\sigma_l^2 \overset{(86)}{\leq} 30\gamma_2^2 R^2\sum_{l=0}^{T-1}\mathbb{E}_{\boldsymbol{\xi}_2^l}\left[\|\theta_l^u\|^2\right] \overset{(84),T\leq K+1}{\leq} \frac{540\gamma_2^2 R^2\sigma^2(K+1)}{m_2} \overset{(68)}{\leq} \frac{R^4}{6\ln\frac{6(K+1)}{\beta}}. \tag{88}$$

**Upper bound for ②.** Probability event $E_{T-1}$ implies

$$\text{②} \quad \leq \quad 2\gamma_2\sum_{l=0}^{T-1}\|\eta_l\|\cdot\|\theta_l^b\| \overset{(77),(82),T\leq K+1}{\leq} \frac{8\sqrt{7}\gamma_2(1+\gamma_2 L)\sigma^2 R(K+1)}{m_2\lambda_2}$$

$$\overset{(65),(66)}{=} \quad \frac{161\sqrt{7}\gamma_2^2\sigma^2(K+1)\ln\frac{6(K+1)}{\beta}}{m_2} \overset{(68)}{\leq} R^2. \tag{89}$$

**Upper bound for ③.** Probability event $E_{T-1}$ implies

$$\text{③} = 2\gamma_2^2\sum_{l=0}^{T-1}\mathbb{E}_{\boldsymbol{\xi}_2^l}\left[\|\theta_l^u\|^2\right] \overset{(84),T\leq K+1}{\leq} \frac{36\gamma_2^2\sigma^2(K+1)}{m_2} \overset{(68)}{\leq} R^2. \tag{90}$$

**Upper bound for ④.** We have

$$2\gamma_2^2\mathbb{E}_{\boldsymbol{\xi}_2^l}\left[\|\theta_l^u\|^2 - \mathbb{E}_{\boldsymbol{\xi}_2^l}\left[\|\theta_l^u\|^2\right]\right] = 0.$$

Next, the summands in ④ are bounded with probability 1:

$$2\gamma_2^2\left|\|\theta_l^u\|^2 - \mathbb{E}_{\boldsymbol{\xi}_2^l}\left[\|\theta_l^u\|^2\right]\right| \quad \leq \quad 2\gamma_2^2\left(\|\theta_l^u\|^2 + \mathbb{E}_{\boldsymbol{\xi}_2^l}\left[\|\theta_l^u\|^2\right]\right) \overset{(81)}{\leq} 16\gamma_2^2\lambda_2^2$$

$$\overset{(66)}{\leq} \quad \frac{R^2}{\ln\frac{6(K+1)}{\beta}} \overset{\text{def}}{=} c. \tag{91}$$

Moreover, these summands have bounded conditional variances $\widetilde{\sigma}_l^2 \overset{\text{def}}{=} 4\gamma_2^4\mathbb{E}_{\boldsymbol{\xi}_2^l}\left[\left(\|\theta_l^u\|^2 - \mathbb{E}_{\boldsymbol{\xi}_2^l}\left[\|\theta_l^u\|^2\right]\right)^2\right]$:

$$\widetilde{\sigma}_l^2 \overset{(91)}{\leq} \frac{2\gamma_2^2 R^2}{\ln\frac{6(K+1)}{\beta}}\mathbb{E}_{\boldsymbol{\xi}_2^l}\left[\left|\|\theta_l^u\|^2 - \mathbb{E}_{\boldsymbol{\xi}_2^l}\left[\|\theta_l^u\|^2\right]\right|\right] \leq \frac{4\gamma_2^2 R^2}{\ln\frac{6(K+1)}{\beta}}\mathbb{E}_{\boldsymbol{\xi}_2^l}\left[\|\theta_l^u\|^2\right] \tag{92}$$

That is, sequence $\{\|\theta_l^u\|^2 - \mathbb{E}_{\boldsymbol{\xi}_2^l}[\|\theta_l^u\|^2]\}_{l\geq 0}$ is a bounded martingale difference sequence having bounded conditional variances $\{\widetilde{\sigma}_l^2\}_{l\geq 0}$. Applying Bernstein's inequality (Lemma B.1) with $X_l = \|\theta_l^u\|^2 - \mathbb{E}_{\boldsymbol{\xi}_2^l}[\|\theta_l^u\|^2]$, $c$ defined in (91), $b = R^2$, $G = \frac{R^4}{6\ln\frac{6(K+1)}{\beta}}$, we get that

$$\mathbb{P}\left\{|\text{④}| > R^2 \text{ and } \sum_{l=0}^{T-1}\widetilde{\sigma}_l^2 \leq \frac{R^4}{6\ln\frac{6(K+1)}{\beta}}\right\} \leq 2\exp\left(-\frac{b^2}{2G + 2cb/3}\right) = \frac{\beta}{3(K+1)}.$$

In other words, $\mathbb{P}\{E_④\} \geq 1 - \frac{\beta}{3(K+1)}$, where probability event $E_④$ is defined as

$$E_④ = \left\{ \text{either} \quad \sum_{l=0}^{T-1} \widetilde{\sigma}_l^2 > \frac{R^4}{6 \ln \frac{6(K+1)}{\beta}} \quad \text{or} \quad |④| \leq R^2 \right\}. \tag{93}$$

Moreover, we notice here that probability event $E_{T-1}$ implies that

$$\sum_{l=0}^{T-1} \widetilde{\sigma}_l^2 \overset{(92)}{\leq} \frac{4\gamma_2^2 R^2}{\ln \frac{6(K+1)}{\beta}} \sum_{l=0}^{T-1} \mathbb{E}_{\boldsymbol{\xi}_2^l} \left[ \|\theta_l^u\|^2 \right] \overset{(84), T \leq K+1}{\leq} \frac{72\gamma_2^2 R^2 \sigma^2(K+1)}{m_2 \ln \frac{6(K+1)}{\beta}}$$

$$\overset{(68)}{\leq} \frac{R^4}{6 \ln \frac{6(K+1)}{\beta}}. \tag{94}$$

**Upper bound for ⑤.** Probability event $E_{T-1}$ implies

$$⑤ = 2\gamma_2^2 \sum_{l=0}^{T-1} \|\theta_l^b\|^2 \overset{(82), T \leq K+1}{\leq} \frac{32\gamma_2^2 \sigma^4(K+1)}{m_2^2 \lambda_2^2} \overset{(66)}{=} \frac{12800\gamma_2^4 \sigma^4(K+1) \ln^2 \frac{6(K+1)}{\beta}}{m_2^2 R^2} \overset{(68)}{\leq} R^2. \tag{95}$$

**Upper bound for ⑥.** Probability event $E_{T-1}$ implies

$$⑥ = 6\gamma_1\gamma_2 \sum_{l=0}^{T-1} \mathbb{E}_{\boldsymbol{\xi}_1^l} \left[ \|\omega_l^u\|^2 \right] \overset{(84), T \leq K+1}{\leq} \frac{108\gamma_1\gamma_2\sigma^2(K+1)}{m_1} \overset{(67)}{\leq} R^2. \tag{96}$$

**Upper bound for ⑦.** We have

$$6\gamma_1\gamma_2 \mathbb{E}_{\boldsymbol{\xi}_1^l} \left[ \|\omega_l^u\|^2 - \mathbb{E}_{\boldsymbol{\xi}_1^l} \left[ \|\omega_l^u\|^2 \right] \right] = 0.$$

Next, the summands in ⑦ are bounded with probability 1:

$$6\gamma_1\gamma_2 \left| \|\omega_l^u\|^2 - \mathbb{E}_{\boldsymbol{\xi}_1^l} \left[ \|\omega_l^u\|^2 \right] \right| \leq 6\gamma_1\gamma_2 \left( \|\omega_l^u\|^2 + \mathbb{E}_{\boldsymbol{\xi}_1^l} \left[ \|\omega_l^u\|^2 \right] \right) \overset{(81)}{\leq} 48\gamma_1\gamma_2\lambda_1^2$$

$$\overset{(66)}{\leq} \frac{\gamma_2 R^2}{\gamma_1 \ln \frac{6(K+1)}{\beta}} \overset{\gamma_2 \leq \gamma_1}{\leq} \frac{R^2}{\ln \frac{6(K+1)}{\beta}} \overset{\text{def}}{=} c. \tag{97}$$

Moreover, these summands have bounded conditional variances $\widehat{\sigma}_l^2 \overset{\text{def}}{=}$ $36\gamma_1^2\gamma_2^2 \mathbb{E}_{\boldsymbol{\xi}_1^l} \left[ \left( \|\omega_l^u\|^2 - \mathbb{E}_{\boldsymbol{\xi}_1^l} \left[ \|\omega_l^u\|^2 \right] \right)^2 \right]$:

$$\widehat{\sigma}_l^2 \overset{(97)}{\leq} \frac{6\gamma_2^2 R^2}{\ln \frac{6(K+1)}{\beta}} \mathbb{E}_{\boldsymbol{\xi}_1^l} \left[ \left| \|\omega_l^u\|^2 - \mathbb{E}_{\boldsymbol{\xi}_1^l} \left[ \|\omega_l^u\|^2 \right] \right| \right] \leq \frac{12\gamma_2^2 R^2}{\ln \frac{6(K+1)}{\beta}} \mathbb{E}_{\boldsymbol{\xi}_1^l} \left[ \|\omega_l^u\|^2 \right] \tag{98}$$

That is, sequence $\{\|\omega_l^u\|^2 - \mathbb{E}_{\boldsymbol{\xi}_1^l}[\|\omega_l^u\|^2]\}_{l \geq 0}$ is a bounded martingale difference sequence having bounded conditional variances $\{\widehat{\sigma}_l^2\}_{l \geq 0}$. Applying Bernstein's inequality (Lemma B.1) with $X_l = \|\omega_l^u\|^2 - \mathbb{E}_{\boldsymbol{\xi}_1^l}[\|\omega_l^u\|^2]$, $c$ defined in (97), $b = R^2$, $G = \frac{R^4}{6 \ln \frac{6(K+1)}{\beta}}$, we get that

$$\mathbb{P} \left\{ |⑦| > R^2 \text{ and } \sum_{l=0}^{T-1} \widehat{\sigma}_l^2 \leq \frac{R^4}{6 \ln \frac{6(K+1)}{\beta}} \right\} \leq 2\exp\left( -\frac{b^2}{2G + \frac{2cb}{3}} \right) = \frac{\beta}{3(K+1)}.$$

In other words, $\mathbb{P}\{E_⑦\} \geq 1 - \frac{\beta}{3(K+1)}$, where probability event $E_⑦$ is defined as

$$E_⑦ = \left\{ \text{either} \quad \sum_{l=0}^{T-1} \widehat{\sigma}_l^2 > \frac{R^4}{6 \ln \frac{6(K+1)}{\beta}} \quad \text{or} \quad |⑦| \leq R^2 \right\}. \tag{99}$$

Moreover, we notice here that probability event $E_{T-1}$ implies that

$$\sum_{l=0}^{T-1} \widehat{\sigma}_l^2 \overset{(98)}{\leq} \frac{12\gamma_2^2 R^2}{\ln \frac{6(K+1)}{\beta}} \sum_{l=0}^{T-1} \mathbb{E}_{\boldsymbol{\xi}_1^l} \left[ \|\omega_l^u\|^2 \right] \overset{(84), T \leq K+1}{\leq} \frac{216\gamma_2^2 R^2 \sigma^2(K+1)}{m_1 \ln \frac{6(K+1)}{\beta}}$$

$$\overset{(67)}{\leq} \frac{R^4}{6 \ln \frac{6(K+1)}{\beta}}. \tag{100}$$

**Upper bound for ⑧.** Probability event $E_{T-1}$ implies

$$
⑧ = 6\gamma_1\gamma_2 \sum_{l=0}^{T-1} \|\omega_l^b\|^2 \overset{(82),T\leq K+1}{\leq} \frac{96\gamma_1\gamma_2\sigma^4(K+1)}{m_1^2\lambda_1^2}
$$

$$
\overset{(66)}{=} \frac{38400\gamma_1^3\gamma_2\sigma^4(K+1)\ln^2\frac{6(K+1)}{\beta}}{m_1^2 R^2} \overset{(67)}{\leq} R^2. \tag{101}
$$

**Final derivation.** Putting all bounds together, we get that $E_{T-1}$ implies

$$
R_T^2 \overset{(80)}{\leq} R^2 + ① + ② + ③ + ④ + ⑤ + ⑥ + ⑦ + ⑧,
$$

$$
② \overset{(89)}{\leq} R^2, \quad ③ \overset{(90)}{\leq} R^2, \quad ⑤ \overset{(95)}{\leq} R^2, \quad ⑥ \overset{(96)}{\leq} R^2, \quad ⑧ \overset{(101)}{\leq} R^2,
$$

$$
\sum_{l=0}^{T-1}\sigma_l^2 \overset{(88)}{\leq} \frac{R^4}{6\ln\frac{6(K+1)}{\beta}}, \quad \sum_{l=0}^{T-1}\widetilde{\sigma}_l^2 \overset{(94)}{\leq} \frac{R^4}{6\ln\frac{6(K+1)}{\beta}}, \quad \sum_{l=0}^{T-1}\widehat{\sigma}_l^2 \overset{(100)}{\leq} \frac{R^4}{6\ln\frac{6(K+1)}{\beta}}.
$$

Moreover, in view of (87), (93), (99), and our induction assumption, we have

$$
\mathbb{P}\{E_{T-1}\} \geq 1 - \frac{(T-1)\beta}{K+1},
$$

$$
\mathbb{P}\{E_①\} \geq 1 - \frac{\beta}{3(K+1)}, \quad \mathbb{P}\{E_④\} \geq 1 - \frac{\beta}{3(K+1)}, \quad \mathbb{P}\{E_⑦\} \geq 1 - \frac{\beta}{3(K+1)},
$$

where probability events $E_①$, $E_④$, and $E_⑦$ are defined as

$$
E_① = \left\{ \text{either} \quad \sum_{l=0}^{T-1}\sigma_l^2 > \frac{R^4}{6\ln\frac{6(K+1)}{\beta}} \quad \text{or} \quad |①| \leq R^2 \right\},
$$

$$
E_④ = \left\{ \text{either} \quad \sum_{l=0}^{T-1}\widetilde{\sigma}_l^2 > \frac{R^4}{6\ln\frac{6(K+1)}{\beta}} \quad \text{or} \quad |④| \leq R^2 \right\},
$$

$$
E_⑦ = \left\{ \text{either} \quad \sum_{l=0}^{T-1}\widehat{\sigma}_l^2 > \frac{R^4}{6\ln\frac{6(K+1)}{\beta}} \quad \text{or} \quad |⑦| \leq R^2 \right\}.
$$

Putting all of these inequalities together, we obtain that probability event $E_{T-1} \cap E_① \cap E_④ \cap E_⑦$ implies

$$
R_T^2 \overset{(80)}{\leq} R^2 + ① + ② + ③ + ④ + ⑤ + ⑥ + ⑦ + ⑧
$$
$$
\leq 9R^2.
$$

Moreover, union bound for the probability events implies

$$
\mathbb{P}\{E_T\} \geq \mathbb{P}\{E_{T-1} \cap E_① \cap E_④ \cap E_⑦\} = 1 - \mathbb{P}\{\overline{E}_{T-1} \cup \overline{E}_① \cup \overline{E}_④ \cup \overline{E}_⑦\} \geq 1 - \frac{T\beta}{K+1}. \tag{102}
$$

This is exactly what we wanted to prove (see the paragraph after inequality (70)). In particular, $E_{K+1}$ implies

$$
\frac{1}{K+1}\sum_{k=0}^{K+1}\|F(x^k)\|^2 \overset{(75)}{\leq} \frac{4(R^2 - R_{K+1}^2)}{\gamma_1\gamma_2(K+1)} + \frac{4(① + ② + ③ + ④ + ⑤ + ⑥ + ⑦ + ⑧)}{\gamma_1\gamma_2(K+1)}
$$
$$
\leq \frac{36R^2}{\gamma_1\gamma_2(K+1)}.
$$

This finishes the proof. $\qquad\square$

**Corollary C.2.** *Let the assumptions of Theorem C.2 hold and*

$$
\rho \leq \frac{1}{640L\ln\frac{6(K+1)}{\beta}}. \tag{103}
$$

*Then, the choice of step-sizes and batch-sizes*

$$2\gamma_2 = \gamma_1 = \frac{1}{160L \ln \frac{6(K+1)}{\beta}}, \quad m_1 = m_2 = \max\left\{1, \frac{81(K+1)\sigma^2}{640L^2R^2 \ln \frac{6(K+1)}{\beta}}\right\} \tag{104}$$

*satisfies conditions* (65), (67), (68)*. With such choice of* $\gamma, m_1, m_2$*, and the choice of* $\lambda_1, \lambda_2$ *as in* (66)*, the iterates produced by* clipped-SEG *after* $K$ *iterations with probability at least* $1 - \beta$ *satisfy*

$$\frac{1}{K+1} \sum_{k=0}^{K} \|F(x^k)\|^2 \leq \frac{1843200L^2R^2 \ln^2 \frac{6(K+1)}{\beta}}{K+1}. \tag{105}$$

*In particular, to guarantee* $\frac{1}{K+1} \sum_{k=0}^{K} \|F(x^k)\|^2 \leq \varepsilon$ *with probability at least* $1 - \beta$ *for some* $\varepsilon > 0$
clipped-SEG *requires,*

$$\mathcal{O}\left(\frac{L^2R^2}{\varepsilon} \ln^2\left(\frac{L^2R^2}{\varepsilon\beta}\right)\right) \text{ iterations,} \tag{106}$$

$$\mathcal{O}\left(\max\left\{\frac{L^2R^2}{\varepsilon} \ln^2\left(\frac{L^2R^2}{\varepsilon\beta}\right), \frac{L^2\sigma^2R^2}{\varepsilon^2} \ln^3\left(\frac{LR^2}{\varepsilon\beta}\right)\right\}\right) \text{ oracle calls.} \tag{107}$$

*Proof.* First of all, we verify that the choice of $\gamma_1, \gamma_2$ and $m_1, m_2$ from (104) satisfies conditions (65), (67), (68). Inequality (65) holds since

$$\gamma_2 + 2\rho \overset{(104)}{=} \frac{1}{320L \ln \frac{6(K+1)}{\beta}} + 2\rho \overset{(103)}{\leq} \frac{1}{320L \ln \frac{6(K+1)}{\beta}} + \frac{1}{320L \ln \frac{6(K+1)}{\beta}} \overset{(104)}{=} \gamma_1$$

and (67), (68) are satisfied since

$$
\begin{aligned}
m_1 &= \max\left\{1, \frac{81(K+1)\sigma^2}{640L^2R^2 \ln \frac{6(K+1)}{\beta}}\right\} \\
&\geq \max\left\{1, \frac{216 \max\{\gamma_1\gamma_2(K+1), \sqrt{\gamma_1^3\gamma_2(K+1)} \ln \frac{6(K+1)}{\beta}\}\sigma^2}{R^2}\right\}, \\
m_2 &= \max\left\{1, \frac{81(K+1)\sigma^2}{640L^2R^2 \ln \frac{6(K+1)}{\beta}}\right\} \geq \max\left\{1, \frac{3240(K+1)\gamma_2^2\sigma^2 \ln \frac{6(K+1)}{\beta}}{R^2}\right\}.
\end{aligned}
$$

Therefore, applying Theorem C.2, we derive that with probability at least $1 - \beta$

$$\frac{1}{K+1} \sum_{k=0}^{K} \|F(x^k)\|^2 \leq \frac{36R^2}{\gamma_1\gamma_2(K+1)} \overset{(104)}{=} \frac{1843200L^2R^2 \ln^2 \frac{6(K+1)}{\beta}}{K+1}.$$

To guarantee $\frac{1}{K+1} \sum_{k=0}^{K} \|F(x^k)\|^2 \leq \varepsilon$, we choose $K$ in such a way that the right-hand side of the above inequality is smaller than $\varepsilon$ that gives

$$K = \mathcal{O}\left(\frac{L^2R^2}{\varepsilon} \ln^2\left(\frac{L^2R^2}{\varepsilon\beta}\right)\right).$$

The total number of oracle calls equals

$$
\begin{aligned}
2m(K+1) &\overset{(104)}{=} 2\max\left\{K+1, \frac{81(K+1)^2\sigma^2}{640L^2R^2 \ln \frac{6(K+1)}{\beta}}\right\} \\
&= \mathcal{O}\left(\max\left\{\frac{L^2R^2}{\varepsilon} \ln^2\left(\frac{L^2R^2}{\varepsilon\beta}\right), \frac{L^2\sigma^2R^2}{\varepsilon^2} \ln^3\left(\frac{L^2R^2}{\varepsilon\beta}\right)\right\}\right).
\end{aligned}
$$

$\square$

## C.3  Quasi-Strongly Monotone Case

**Lemma C.3.** *Let Assumptions [1.2], [1.5] hold for $Q = B_{3R}(x^*) = \{x \in \mathbb{R}^d \mid \|x - x^*\| \leq 3R\}$, where $R \geq R_0 \overset{\text{def}}{=} \|x^0 - x^*\|$, and $\gamma_1 = \gamma_2 = \gamma$, $0 < \gamma \leq 1/2(L+2\mu)$. If $x^k$ and $\widetilde{x}^k \overset{\text{def}}{=} x^k - \gamma F(x^k)$ lie in $\overline{B}_{3R}(x^*)$ for all $k = 0, 1, \ldots, K$ for some $K \geq 0$, then the iterates produced by* clipped-SEG *satisfy*

$$
\begin{aligned}
\|x^{K+1} - x^*\|^2 &\leq (1 - \gamma\mu)^{K+1}\|x^0 - x^*\|^2 - 4\gamma^3\mu\sum_{k=0}^{K}(1-\gamma\mu)^{K-k}\langle F(x^k), \omega_k\rangle \\
&\quad + 2\gamma\sum_{k=0}^{K}(1-\gamma\mu)^{K-k}\langle x^k - x^* - \gamma F(\widetilde{x}^k), \theta_k\rangle \\
&\quad + \gamma^2\sum_{k=0}^{K}(1-\gamma\mu)^{K-k}\left(\|\theta_k\|^2 + 4\|\omega_k\|^2\right),
\end{aligned}
\tag{108}
$$

*where $\theta_k, \omega_k$ are defined in* (15)*,* (16)*.*

*Proof.* Using the update rule of clipped-SEG, we obtain

$$
\begin{aligned}
\|x^{k+1} - x^*\|^2 &= \|x^k - x^*\|^2 - 2\gamma\langle x^k - x^*, \widetilde{F}_{\boldsymbol{\xi}_2^k}(\widetilde{x}^k)\rangle + \gamma^2\|\widetilde{F}_{\boldsymbol{\xi}_2^k}(\widetilde{x}^k)\|^2 \\
&= \|x^k - x^*\|^2 - 2\gamma\langle x^k - x^*, F(\widetilde{x}^k)\rangle + 2\gamma\langle x^k - x^*, \theta_k\rangle \\
&\quad + \gamma^2\|F(\widetilde{x}^k)\|^2 - 2\gamma^2\langle F(\widetilde{x}^k), \theta_k\rangle + \gamma^2\|\theta_k\|^2 \\
&= \|x^k - x^*\|^2 - 2\gamma\langle \widetilde{x}^k - x^*, F(\widetilde{x}^k)\rangle - 2\gamma\langle x^k - \widetilde{x}^k, F(\widetilde{x}^k)\rangle \\
&\quad + 2\gamma\langle x^k - x^* - \gamma F(\widetilde{x}^k), \theta_k\rangle + \gamma^2\|F(\widetilde{x}^k)\|^2 + \gamma^2\|\theta_k\|^2.
\end{aligned}
$$

Since $F$ is $\mu$-quasi strongly monotone, we have

$$
\begin{aligned}
-2\gamma\langle \widetilde{x}^k - x^*, F(\widetilde{x}^k)\rangle &\leq -2\gamma\mu\|\widetilde{x}^k - x^*\|^2 \overset{(6)}{\leq} -\gamma\mu\|x^k - x^*\|^2 + 2\gamma\mu\|\widetilde{x}^k - x^k\|^2 \\
&= -\gamma\mu\|x^k - x^*\|^2 + 2\gamma^3\mu\|\widetilde{F}_{\boldsymbol{\xi}_1}(x^k)\|^2 \\
&= -\gamma\mu\|x^k - x^*\|^2 + 2\gamma^3\mu\|F(x^k)\|^2 - 4\gamma^3\mu\langle F(x^k), \omega_k\rangle + 2\gamma^3\mu\|\omega_k\|^2.
\end{aligned}
$$

Moreover, $-2\gamma\langle x^k - \widetilde{x}^k, F(\widetilde{x}^k)\rangle$ can be rewritten as

$$
\begin{aligned}
-2\gamma\langle x^k - \widetilde{x}^k, F(\widetilde{x}^k)\rangle &= -2\gamma^2\langle \widetilde{F}_{\boldsymbol{\xi}_1}(x^k), F(\widetilde{x}^k)\rangle \\
&= \gamma^2\|\widetilde{F}_{\boldsymbol{\xi}_1}(x^k) - F(x^k)\|^2 - \gamma^2\|\widetilde{F}_{\boldsymbol{\xi}_1}(x^k)\|^2 - \gamma^2\|F(\widetilde{x}^k)\|^2.
\end{aligned}
$$

Putting all together, we get

$$
\begin{aligned}
\|x^{k+1} - x^*\|^2 \quad\leq\quad & (1-\gamma\mu)\|x^k - x^*\|^2 + 2\gamma^3\mu\|F(x^k)\|^2 - 4\gamma^3\mu\langle F(x^k), \omega_k\rangle + 2\gamma^3\mu\|\omega_k\|^2 \\
& +\gamma^2\|\widetilde{F}_{\boldsymbol{\xi}_1}(x^k) - F(x^k)\|^2 - \gamma^2\|\widetilde{F}_{\boldsymbol{\xi}_1}(x^k)\|^2 - \gamma^2\|F(\widetilde{x}^k)\|^2 \\
& +2\gamma\langle x^k - x^* - \gamma F(\widetilde{x}^k), \theta_k\rangle + \gamma^2\|F(\widetilde{x}^k)\|^2 + \gamma^2\|\theta_k\|^2 \\
\overset{(5)}{\leq}\quad & (1-\gamma\mu)\|x^k - x^*\|^2 + 2\gamma^3\mu\|F(x^k)\|^2 - 4\gamma^3\mu\langle F(x^k), \omega_k\rangle + 2\gamma^3\mu\|\omega_k\|^2 \\
& +2\gamma^2\|\omega_k\|^2 + 2\gamma^2\|F(x^k) - F(\widetilde{x}^k)\|^2 - \gamma^2\|\widetilde{F}_{\boldsymbol{\xi}_1}(x^k)\|^2 \\
& +2\gamma\langle x^k - x^* - \gamma F(\widetilde{x}^k), \theta_k\rangle + \gamma^2\|\theta_k\|^2 \\
\overset{(\text{Lip})}{\leq}\quad & (1-\gamma\mu)\|x^k - x^*\|^2 + 2\gamma^3\mu\|F(x^k)\|^2 - 4\gamma^3\mu\langle F(x^k), \omega_k\rangle \\
& +2\gamma^2(1+\gamma\mu)\|\omega_k\|^2 - \gamma^2(1-2\gamma^2L^2)\|\widetilde{F}_{\boldsymbol{\xi}_1}(x^k)\|^2 \\
& +2\gamma\langle x^k - x^* - \gamma F(\widetilde{x}^k), \theta_k\rangle + \gamma^2\|\theta_k\|^2 \\
\overset{(6)}{\leq}\quad & (1-\gamma\mu)\|x^k - x^*\|^2 - \gamma^2\left(\frac{1}{2} - \gamma^2L^2 - 2\gamma\mu\right)\|F(x^k)\|^2 \\
& -4\gamma^3\mu\langle F(x^k), \omega_k\rangle + \gamma^2(3 - 2\gamma^2L^2 + 2\gamma\mu)\|\omega_k\|^2 \\
& +2\gamma\langle x^k - x^* - \gamma F(\widetilde{x}^k), \theta_k\rangle + \gamma^2\|\theta_k\|^2 \\
\leq\quad & (1-\gamma\mu)\|x^k - x^*\|^2 - 4\gamma^3\mu\langle F(x^k), \omega_k\rangle \\
& +2\gamma\langle x^k - x^* - \gamma F(\widetilde{x}^k), \theta_k\rangle + \gamma^2\left(\|\theta_k\|^2 + 4\|\omega_k\|^2\right),
\end{aligned}
$$

where in the last step we apply $0 < \gamma \leq 1/2(L+2\mu)$. Unrolling the recurrence, we obtain (108). $\qquad\square$

**Theorem C.3.** *Let Assumptions 1.1, 1.2, 1.5, hold for $Q = B_{3R}(x^*) = \{x \in \mathbb{R}^d \mid \|x - x^*\| \leq 3R\}$, where $R \geq R_0 \overset{def}{=} \|x^0 - x^*\|$, and $\gamma_1 = \gamma_2 = \gamma$,*

$$
0 < \gamma \quad\leq\quad \frac{1}{650L\ln\frac{6(K+1)}{\beta}}, \tag{109}
$$

$$
\lambda_{1,k} = \lambda_{2,k} = \lambda_k \quad=\quad \frac{\exp(-\gamma\mu(1 + k/2))R}{120\gamma\ln\frac{6(K+1)}{\beta}}, \tag{110}
$$

$$
m_{1,k} = m_{2,k} = m_k \quad\geq\quad \max\left\{1, \frac{264600\gamma^2(K+1)\sigma^2\ln\frac{6(K+1)}{\beta}}{\exp(-\gamma\mu k)R^2}\right\}, \tag{111}
$$

*for some $K \geq 0$ and $\beta \in (0,1]$ such that $\ln\frac{6(K+1)}{\beta} \geq 1$. Then, after $K$ iterations the iterates produced by* clipped-SEG *with probability at least $1 - \beta$ satisfy*

$$
\|x^{K+1} - x^*\|^2 \leq 2\exp(-\gamma\mu(K+1))R^2. \tag{112}
$$

*Proof.* As in the proof of Theorem C.1, we use the following notation: $R_k = \|x^k - x^*\|^2$, $k \geq 0$. We will derive (112) by induction. In particular, for each $k = 0, \ldots, K+1$ we define probability event $E_k$ as follows: inequalities

$$
R_t^2 \leq 2\exp(-\gamma\mu t)R^2 \tag{113}
$$

hold for $t = 0, 1, \ldots, k$ simultaneously. Our goal is to prove that $\mathbb{P}\{E_k\} \geq 1 - k\beta/(K+1)$ for all $k = 0, 1, \ldots, K+1$. We use the induction to show this statement. For $k = 0$ the statement is trivial since $R_0^2 \leq 2R^2$ by definition. Next, assume that the statement holds for $k = T - 1 \leq K$, i.e., we have $\mathbb{P}\{E_{T-1}\} \geq 1 - (T-1)\beta/(K+1)$. We need to prove that $\mathbb{P}\{E_T\} \geq 1 - T\beta/(K+1)$. First of all, since $R_t^2 \leq 2\exp(-\gamma\mu t)R^2 \leq 9R^2$, we have $x^t \in B_{3R}(x^*)$. Operator $F$ is $L$-Lipschitz on $B_{3R}(x^*)$. Therefore, probability event $E_{T-1}$ implies

$$
\|F(x^t)\| \quad\leq\quad L\|x^t - x^*\| \overset{(113)}{\leq} \sqrt{2}L\exp(-\gamma\mu t/2)R \overset{(109),(110)}{\leq} \frac{\lambda_t}{2}. \tag{114}
$$

and

$$
\|\omega_t\|^2 \quad\overset{(5)}{\leq}\quad 2\|\widetilde{F}_{\boldsymbol{\xi}_1}(x^t)\|^2 + 2\|F(x^t)\|^2 \overset{(114)}{\leq} \frac{5}{2}\lambda_t^2 \overset{(110)}{\leq} \frac{\exp(-\gamma\mu t)R^2}{4\gamma^2} \tag{115}
$$

for all $t = 0, 1, \ldots, T - 1$.

Next, we show that probability event $E_{T-1}$ implies $\|\widetilde{x}^t - x^*\| \leq 3R$ and derive useful inequalities related to $\theta_t$ for all $t = 0, 1, \ldots, T - 1$. Indeed, due to Lipschitzness of $F$ probability event $E_{T-1}$ implies

$$
\begin{aligned}
\|\widetilde{x}^t - x^*\|^2 \quad &= \quad \|x^t - x^* - \gamma \widetilde{F}_{\boldsymbol{\xi}_1}(x^t)\|^2 \overset{(5)}{\leq} 2\|x^t - x^*\|^2 + 2\gamma^2 \|\widetilde{F}_{\boldsymbol{\xi}_1}(x^t)\|^2 \\
&\overset{(5)}{\leq} \quad 2R_t^2 + 4\gamma^2 \|F(x^t)\|^2 + 4\gamma^2 \|\omega_t\|^2 \\
&\overset{(Lip)}{\leq} \quad 2(1 + 2\gamma^2 L^2)R_t^2 + 4\gamma^2 \|\omega_t\|^2 \\
&\overset{(109),(115)}{\leq} \quad 7\exp(-\gamma\mu t)R^2 \leq 9R^2
\end{aligned}
\tag{116}
$$

and

$$
\|F(\widetilde{x}^t)\| \quad \leq \quad L\|\widetilde{x}^t - x^*\| \leq \sqrt{7}L\exp(-\gamma\mu t/2)R \overset{(109),(110)}{\leq} \frac{\lambda_t}{2}
\tag{117}
$$

for all $t = 0, 1, \ldots, T - 1$.

That is, $E_{T-1}$ implies that $x^t, \widetilde{x}^t \in B_{3R}(x^*)$ for all $t = 0, 1, \ldots, T - 1$. Applying Lemma C.3 and $(1 - \gamma\mu)^T \leq \exp(-\gamma\mu T)$, we get that probability event $E_{T-1}$ implies

$$
\begin{aligned}
R_T^2 \quad \leq \quad &\exp(-\gamma\mu T)R^2 - 4\gamma^3\mu \sum_{l=0}^{T-1}(1 - \gamma\mu)^{T-1-l}\langle F(x^l), \omega_l\rangle \\
&+ 2\gamma \sum_{l=0}^{T-1}(1 - \gamma\mu)^{T-1-l}\langle x^l - x^* - \gamma F(\widetilde{x}^l), \theta_l\rangle \\
&+ \gamma^2 \sum_{l=0}^{T-1}(1 - \gamma\mu)^{T-1-l}\left(\|\theta_l\|^2 + 4\|\omega_l\|^2\right).
\end{aligned}
$$

To estimate the sums in the right-hand side, we introduce new vectors:

$$
\zeta_t = \begin{cases} F(x^t), & \text{if } \|F(x^t)\| \leq \sqrt{2}L\exp(-\gamma\mu t/2)R, \\ 0, & \text{otherwise,} \end{cases}
\tag{118}
$$

$$
\eta_t = \begin{cases} x^t - x^* - \gamma F(\widetilde{x}^t), & \text{if } \|x^t - x^* - \gamma F(\widetilde{x}^t)\| \leq \sqrt{7}(1 + \gamma L)\exp(-\gamma\mu t/2)R, \\ 0, & \text{otherwise,} \end{cases}
\tag{119}
$$

for $t = 0, 1, \ldots, T - 1$. First of all, we point out that vectors $\zeta_t$ and $\eta_t$ are bounded with probability 1, i.e., with probability 1

$$
\|\zeta_t\| \leq \sqrt{2}L\exp(-\gamma\mu t/2)R, \quad \|\eta_t\| \leq \sqrt{7}(1 + \gamma L)\exp(-\gamma\mu t/2)R
\tag{120}
$$

for all $t = 0, 1, \ldots, T - 1$. Next, we notice that $E_{T-1}$ implies $\|F(x^t)\| \leq \sqrt{2}L\exp(-\gamma\mu t/2)R$ (due to (114)) and

$$
\begin{aligned}
\|x^t - x^* - \gamma F(\widetilde{x}^t)\| \quad &\leq \quad \|x^t - x^*\| + \gamma\|F(\widetilde{x}^t)\| \\
&\overset{(116),(117)}{\leq} \quad \sqrt{7}(1 + \gamma L)\exp(-\gamma\mu t/2)R
\end{aligned}
$$

for $t = 0, 1, \ldots, T - 1$, i.e., probability event $E_{T-1}$ implies $\zeta_t = F(x^t)$ and $\eta_t = x^t - x^* - \gamma F(\widetilde{x}^t)$ for all $t = 0, 1, \ldots, T - 1$. Therefore, $E_{T-1}$ implies

$$
\begin{aligned}
R_T^2 \quad \leq \quad &\exp(-\gamma\mu T)R^2 - 4\gamma^3\mu \sum_{l=0}^{T-1}(1 - \gamma\mu)^{T-1-l}\langle \zeta_l, \omega_l\rangle \\
&+ 2\gamma \sum_{l=0}^{T-1}(1 - \gamma\mu)^{T-1-l}\langle \eta_l, \theta_l\rangle + \gamma^2 \sum_{l=0}^{T-1}(1 - \gamma\mu)^{T-1-l}\left(\|\theta_l\|^2 + 4\|\omega_l\|^2\right).
\end{aligned}
$$

As in the monotone case, to continue the derivation, we introduce vectors $\theta_l^u, \theta_l^b, \omega_l^u, \omega_l^b$ defined as

$$\theta_l^u \stackrel{\text{def}}{=} \mathbb{E}_{\boldsymbol{\xi}_2^l}\left[\widetilde{F}_{\boldsymbol{\xi}_2^l}(\widetilde{x}^l)\right] - \widetilde{F}_{\boldsymbol{\xi}_2^l}(\widetilde{x}^l), \quad \theta_l^b \stackrel{\text{def}}{=} F(\widetilde{x}^l) - \mathbb{E}_{\boldsymbol{\xi}_2^l}\left[\widetilde{F}_{\boldsymbol{\xi}_2^l}(\widetilde{x}^l)\right], \tag{121}$$

$$\omega_l^u \stackrel{\text{def}}{=} \mathbb{E}_{\boldsymbol{\xi}_1^l}\left[\widetilde{F}_{\boldsymbol{\xi}_1^l}(x^l)\right] - \widetilde{F}_{\boldsymbol{\xi}_1^l}(x^l), \quad \omega_l^b \stackrel{\text{def}}{=} F(x^l) - \mathbb{E}_{\boldsymbol{\xi}_1^l}\left[\widetilde{F}_{\boldsymbol{\xi}_1^l}(x^l)\right], \tag{122}$$

for all $l = 0, \ldots, T - 1$. By definition we have $\theta_l = \theta_l^u + \theta_l^b$, $\omega_l = \omega_l^u + \omega_l^b$ for all $l = 0, \ldots, T - 1$. Using the introduced notation, we continue our derivation as follows: $E_{T-1}$ implies

$$R_T^2 \stackrel{(5)}{\leq} \exp(-\gamma\mu T)R^2 \underbrace{-4\gamma^3\mu\sum_{l=0}^{T-1}(1-\gamma\mu)^{T-1-l}\langle\zeta_l, \omega_l^u\rangle}_{\text{①}} \underbrace{-4\gamma^3\mu\sum_{l=0}^{T-1}(1-\gamma\mu)^{T-1-l}\langle\zeta_l, \omega_l^b\rangle}_{\text{②}}$$

$$+ \underbrace{2\gamma\sum_{l=0}^{T-1}(1-\gamma\mu)^{T-1-l}\langle\eta_l, \theta_l^u\rangle}_{\text{③}} + \underbrace{2\gamma\sum_{l=0}^{T-1}(1-\gamma\mu)^{T-1-l}\langle\eta_l, \theta_l^b\rangle}_{\text{④}}$$

$$+ \underbrace{2\gamma^2\sum_{l=0}^{T-1}(1-\gamma\mu)^{T-1-l}\left(\mathbb{E}_{\boldsymbol{\xi}_2^l}\left[\|\theta_l^u\|^2\right] + 4\mathbb{E}_{\boldsymbol{\xi}_1^l}\left[\|\omega_l^u\|^2\right]\right)}_{\text{⑤}}$$

$$+ \underbrace{2\gamma^2\sum_{l=0}^{T-1}(1-\gamma\mu)^{T-1-l}\left(\|\theta_l^u\|^2 + 4\|\omega_l^u\|^2 - \mathbb{E}_{\boldsymbol{\xi}_2^l}\left[\|\theta_l^u\|^2\right] - 4\mathbb{E}_{\boldsymbol{\xi}_1^l}\left[\|\omega_l^u\|^2\right]\right)}_{\text{⑥}}$$

$$+ \underbrace{2\gamma^2\sum_{l=0}^{T-1}(1-\gamma\mu)^{T-1-l}\left(\|\theta_l^b\|^2 + 4\|\omega_l^b\|^2\right)}_{\text{⑦}}. \tag{123}$$

The rest of the proof is based on deriving good enough upper bounds for ①, ②, ③, ④, ⑤, ⑥, ⑦, i.e., we want to prove that ① + ② + ③ + ④ + ⑤ + ⑥ + ⑦ $\leq \exp(-\gamma\mu T)R^2$ with high probability.

Before we move on, we need to derive some useful inequalities for operating with $\theta_l^u, \theta_l^b, \omega_l^u, \omega_l^b$. First of all, Lemma B.2 implies that

$$\|\theta_l^u\| \leq 2\lambda_l, \quad \|\omega_l^u\| \leq 2\lambda_l \tag{124}$$

for all $l = 0, 1, \ldots, T - 1$. Next, since $\{\xi_1^{i,l}\}_{i=1}^{m_l}$, $\{\xi_2^{i,l}\}_{i=1}^{m_l}$ are independently sampled from $\mathcal{D}$, we have $\mathbb{E}_{\boldsymbol{\xi}_1^l}[F_{\boldsymbol{\xi}_1^l}(x^l)] = F(x^l)$, $\mathbb{E}_{\boldsymbol{\xi}_2^l}[F_{\boldsymbol{\xi}_2^l}(\widetilde{x}^l)] = F(\widetilde{x}^l)$, and

$$\mathbb{E}_{\boldsymbol{\xi}_1^l}\left[\|F_{\boldsymbol{\xi}_1^l}(x^l) - F(x^l)\|^2\right] = \frac{1}{m_l^2}\sum_{i=1}^{m_l}\mathbb{E}_{\xi_1^{i,l}}\left[\|F_{\xi_1^{i,l}}(x^l) - F(x^l)\|^2\right] \stackrel{(1)}{\leq} \frac{\sigma^2}{m_l},$$

$$\mathbb{E}_{\boldsymbol{\xi}_2^l}\left[\|F_{\boldsymbol{\xi}_2^l}(\widetilde{x}^l) - F(\widetilde{x}^l)\|^2\right] = \frac{1}{m_l^2}\sum_{i=1}^{m_l}\mathbb{E}_{\xi_2^{i,l}}\left[\|F_{\xi_2^{i,l}}(\widetilde{x}^l) - F(\widetilde{x}^l)\|^2\right] \stackrel{(1)}{\leq} \frac{\sigma^2}{m_l},$$

for all $l = 0, 1, \ldots, T - 1$. Moreover, as we already derived, probability event $E_{T-1}$ implies that $\|F(x^l)\| \leq \lambda_l/2$ and $\|F(\widetilde{x}^l)\| \leq \lambda_l/2$ for all $l = 0, 1, \ldots, T - 1$ (see (114) and (117)). Therefore, in view of Lemma B.2, $E_{T-1}$ implies that

$$\|\theta_l^b\| \leq \frac{4\sigma^2}{m_l\lambda_l}, \quad \|\omega_l^b\| \leq \frac{4\sigma^2}{m_l\lambda_l}, \tag{125}$$

$$\mathbb{E}_{\boldsymbol{\xi}_2^l}\left[\|\theta_l\|^2\right] \leq \frac{18\sigma^2}{m_l}, \quad \mathbb{E}_{\boldsymbol{\xi}_1^l}\left[\|\omega_l\|^2\right] \leq \frac{18\sigma^2}{m_l}, \tag{126}$$

$$\mathbb{E}_{\boldsymbol{\xi}_2^l}\left[\|\theta_l^u\|^2\right] \leq \frac{18\sigma^2}{m_l}, \quad \mathbb{E}_{\boldsymbol{\xi}_1^l}\left[\|\omega_l^u\|^2\right] \leq \frac{18\sigma^2}{m_l}, \tag{127}$$

for all $l = 0, 1, \ldots, T - 1$.

**Upper bound for ①.** Since $\mathbb{E}_{\boldsymbol{\xi}_1^l}[\omega_l^u] = 0$, we have

$$\mathbb{E}_{\boldsymbol{\xi}_1^l}\left[-4\gamma^3\mu(1-\gamma\mu)^{T-1-l}\langle\zeta_l,\omega_l^u\rangle\right] = 0.$$

Next, the summands in ① are bounded with probability 1:

$$
\begin{aligned}
|-4\gamma^3\mu(1-\gamma\mu)^{T-1-l}\langle\zeta_l,\omega_l^u\rangle| &\leq 4\gamma^3\mu\exp(-\gamma\mu(T-1-l))\|\zeta_l\|\cdot\|\omega_l^u\| \\
&\overset{(120),(124)}{\leq} 8\sqrt{2}\gamma^3\mu L\exp(-\gamma\mu(T-1-{}^l\!/_2))R\lambda_l \\
&\overset{(109),(110)}{\leq} \frac{\exp(-\gamma\mu T)R^2}{7\ln\frac{6(K+1)}{\beta}} \overset{\text{def}}{=} c.
\end{aligned}
\tag{128}
$$

Moreover, these summands have bounded conditional variances $\sigma_l^2 \overset{\text{def}}{=} \mathbb{E}_{\boldsymbol{\xi}_1^l}\left[16\gamma^6\mu^2(1-\gamma\mu)^{2T-2-2l}\langle\zeta_l,\omega_l^u\rangle^2\right]$:

$$
\begin{aligned}
\sigma_l^2 &\leq \mathbb{E}_{\boldsymbol{\xi}_1^l}\left[16\gamma^6\mu^2\exp(-\gamma\mu(2T-2-2l))\|\zeta_l\|^2\cdot\|\omega_l^u\|^2\right] \\
&\overset{(120)}{\leq} 36\gamma^6\mu^2 L^2\exp(-\gamma\mu(2T-2-l))R^2\mathbb{E}_{\boldsymbol{\xi}_1^l}\left[\|\omega_l^u\|^2\right] \\
&\overset{(109)}{\leq} \frac{4\gamma^2\exp(-\gamma\mu(2T-l))R^2}{2809\ln\frac{6(K+1)}{\beta}}\mathbb{E}_{\boldsymbol{\xi}_1^l}\left[\|\omega_l^u\|^2\right].
\end{aligned}
\tag{129}
$$

That is, sequence $\{-4\gamma^3\mu(1-\gamma\mu)^{T-1-l}\langle\zeta_l,\omega_l^u\rangle\}_{l\geq0}$ is a bounded martingale difference sequence having bounded conditional variances $\{\sigma_l^2\}_{l\geq0}$. Applying Bernstein's inequality (Lemma B.1) with $X_l = -4\gamma^3\mu(1-\gamma\mu)^{T-1-l}\langle\zeta_l,\omega_l^u\rangle$, $c$ defined in (128), $b = \frac{1}{7}\exp(-\gamma\mu T)R^2$, $G = \frac{\exp(-2\gamma\mu T)R^4}{294\ln\frac{6(K+1)}{\beta}}$, we get that

$$\mathbb{P}\left\{|①| > \frac{1}{7}\exp(-\gamma\mu T)R^2 \text{ and } \sum_{l=0}^{T-1}\sigma_l^2 \leq \frac{\exp(-2\gamma\mu T)R^4}{294\ln\frac{6(K+1)}{\beta}}\right\} \leq 2\exp\left(-\frac{b^2}{2G+{}^{2cb}\!/_3}\right) = \frac{\beta}{3(K+1)}.$$

In other words, $\mathbb{P}\{E_①\} \geq 1 - \frac{\beta}{3(K+1)}$, where probability event $E_①$ is defined as

$$E_① = \left\{\text{either} \quad \sum_{l=0}^{T-1}\sigma_l^2 > \frac{\exp(-2\gamma\mu T)R^4}{294\ln\frac{6(K+1)}{\beta}} \quad \text{or} \quad |①| \leq \frac{1}{7}\exp(-\gamma\mu T)R^2\right\}.
\tag{130}$$

Moreover, we notice here that probability event $E_{T-1}$ implies that

$$
\begin{aligned}
\sum_{l=0}^{T-1}\sigma_l^2 &\overset{(129)}{\leq} \frac{4\gamma^2\exp(-2\gamma\mu T)R^2}{2809\ln\frac{6(K+1)}{\beta}}\sum_{l=0}^{T-1}\frac{\mathbb{E}_{\boldsymbol{\xi}_1^l}\left[\|\omega_l^u\|^2\right]}{\exp(-\gamma\mu l)} \\
&\overset{(127),T\leq K+1}{\leq} \frac{72\gamma^2\exp(-2\gamma\mu T)R^2\sigma^2}{2809\ln\frac{6(K+1)}{\beta}}\sum_{l=0}^{K}\frac{1}{m_l\exp(-\gamma\mu l)} \\
&\overset{(111)}{\leq} \frac{\exp(-2\gamma\mu T)R^4}{294\ln\frac{6(K+1)}{\beta}}.
\end{aligned}
\tag{131}
$$

**Upper bound for ②.** Probability event $E_{T-1}$ implies

$$
\begin{aligned}
② &\leq 4\gamma^3\mu\sum_{l=0}^{T-1}\exp(-\gamma\mu(T-1-l))\|\zeta_l\|\cdot\|\omega_l^b\| \\
&\overset{(120),(125)}{\leq} 16\sqrt{2}\exp(-\gamma\mu(T-1))\gamma^3\mu LR\sum_{l=0}^{T-1}\frac{\sigma^2}{m_l\lambda_l\exp(-\gamma\mu l/2)} \\
&\overset{(110)}{=} 1920\sqrt{2}\exp(-\gamma\mu(T-2))\gamma^4\mu L\sum_{l=0}^{T-1}\frac{\sigma^2\ln\frac{6(K+1)}{\beta}}{m_l\exp(-\gamma\mu l)} \\
&\overset{(109),(111),T\leq K+1}{\leq} \frac{1}{7}\exp(-\gamma\mu T)R^2.
\end{aligned}
\tag{132}
$$

**Upper bound for ③.** Since $\mathbb{E}_{\boldsymbol{\xi}_2^l}[\theta_l^u] = 0$, we have

$$\mathbb{E}_{\boldsymbol{\xi}_2^l}\left[2\gamma(1-\gamma\mu)^{T-1-l}\langle\eta_l,\theta_l^u\rangle\right] = 0.$$

Next, the summands in ③ are bounded with probability 1:

$$
\begin{aligned}
|2\gamma(1-\gamma\mu)^{T-1-l}\langle\eta_l,\theta_l^u\rangle| &\leq 2\gamma\exp(-\gamma\mu(T-1-l))\|\eta_l\|\cdot\|\theta_l^u\| \\
&\overset{(120),(124)}{\leq} 4\sqrt{7}\gamma(1+\gamma L)\exp(-\gamma\mu(T-1-{}^l\!/_2))R\lambda_l \\
&\overset{(109),(110)}{\leq} \frac{\exp(-\gamma\mu T)R^2}{7\ln\frac{6(K+1)}{\beta}} \overset{\text{def}}{=} c.
\end{aligned}
\tag{133}
$$

Moreover, these summands have bounded conditional variances $\widetilde{\sigma}_l^2 \overset{\text{def}}{=} \mathbb{E}_{\boldsymbol{\xi}_2^l}\left[4\gamma^2(1-\gamma\mu)^{2T-2-2l}\langle\eta_l,\theta_l^u\rangle^2\right]$:

$$
\begin{aligned}
\widetilde{\sigma}_l^2 &\leq \mathbb{E}_{\boldsymbol{\xi}_2^l}\left[4\gamma^2\exp(-\gamma\mu(2T-2-2l))\|\eta_l\|^2\cdot\|\theta_l^u\|^2\right] \\
&\overset{(120)}{\leq} 49\gamma^2(1+\gamma L)^2\exp(-\gamma\mu(2T-2-l))R^2\mathbb{E}_{\boldsymbol{\xi}_2^l}\left[\|\theta_l^u\|^2\right] \\
&\overset{(109)}{\leq} 50\gamma^2\exp(-\gamma\mu(2T-l))R^2\mathbb{E}_{\boldsymbol{\xi}_2^l}\left[\|\theta_l^u\|^2\right].
\end{aligned}
\tag{134}
$$

That is, sequence $\{2\gamma(1-\gamma\mu)^{T-1-l}\langle\eta_l,\theta_l^u\rangle\}_{l\geq0}$ is a bounded martingale difference sequence having bounded conditional variances $\{\widetilde{\sigma}_l^2\}_{l\geq0}$. Applying Bernstein's inequality (Lemma B.1) with $X_l = 2\gamma(1-\gamma\mu)^{T-1-l}\langle\eta_l,\theta_l^u\rangle$, $c$ defined in (133), $b = \frac{1}{7}\exp(-\gamma\mu T)R^2$, $G = \frac{\exp(-2\gamma\mu T)R^4}{294\ln\frac{6(K+1)}{\beta}}$, we get that

$$\mathbb{P}\left\{|③| > \frac{1}{7}\exp(-\gamma\mu T)R^2 \text{ and } \sum_{l=0}^{T-1}\widetilde{\sigma}_l^2 \leq \frac{\exp(-2\gamma\mu T)R^4}{294\ln\frac{6(K+1)}{\beta}}\right\} \leq 2\exp\left(-\frac{b^2}{2G+{}^{2cb}\!/_3}\right) = \frac{\beta}{3(K+1)}.$$

In other words, $\mathbb{P}\{E_③\} \geq 1 - \frac{\beta}{3(K+1)}$, where probability event $E_③$ is defined as

$$E_③ = \left\{\text{either} \quad \sum_{l=0}^{T-1}\widetilde{\sigma}_l^2 > \frac{\exp(-2\gamma\mu T)R^4}{294\ln\frac{6(K+1)}{\beta}} \quad \text{or} \quad |③| \leq \frac{1}{7}\exp(-\gamma\mu T)R^2\right\}.
\tag{135}$$

Moreover, we notice here that probability event $E_{T-1}$ implies that

$$
\begin{aligned}
\sum_{l=0}^{T-1}\widetilde{\sigma}_l^2 &\overset{(134)}{\leq} 50\gamma^2\exp(-2\gamma\mu T)R^2\sum_{l=0}^{T-1}\frac{\mathbb{E}_{\boldsymbol{\xi}_2^l}\left[\|\theta_l^u\|^2\right]}{\exp(-\gamma\mu l)} \\
&\overset{(127),T\leq K+1}{\leq} 900\gamma^2\exp(-2\gamma\mu T)R^2\sigma^2\sum_{l=0}^{K}\frac{1}{m_l\exp(-\gamma\mu l)} \\
&\overset{(111)}{\leq} \frac{\exp(-2\gamma\mu T)R^4}{294\ln\frac{6(K+1)}{\beta}}.
\end{aligned}
\tag{136}
$$

**Upper bound for ④.** Probability event $E_{T-1}$ implies

$$
\begin{aligned}
④ &\leq 2\gamma\exp(-\gamma\mu(T-1))\sum_{l=0}^{T-1}\frac{\|\eta_l\|\cdot\|\theta_l^b\|}{\exp(-\gamma\mu l)} \\
&\overset{(120),(125)}{\leq} 8\sqrt{7}\gamma(1+\gamma L)\exp(-\gamma\mu(T-1))R\sum_{l=0}^{T-1}\frac{\sigma^2}{m_l\lambda_l\exp(-\gamma\mu l/2)} \\
&\overset{(110)}{\leq} 960\sqrt{7}\gamma^2(1+\gamma L)\exp(-\gamma\mu(T-2))\sum_{l=0}^{T-1}\frac{\sigma^2\ln\frac{6(K+1)}{\beta}}{m_l\exp(-\gamma\mu l)} \\
&\overset{(111),T\leq K+1}{\leq} \frac{1}{7}\exp(-\gamma\mu T)R^2.
\end{aligned}
\tag{137}
$$

**Upper bound for ⑤.** Probability event $E_{T-1}$ implies

$$⑤ \quad = \quad 2\gamma^2 \exp(-\gamma\mu(T-1)) \sum_{l=0}^{T-1} \frac{\mathbb{E}_{\boldsymbol{\xi}_2^l}\left[\|\theta_l^u\|^2\right] + 4\mathbb{E}_{\boldsymbol{\xi}_1^l}\left[\|\omega_l^u\|^2\right]}{\exp(-\gamma\mu l)}$$

$$\overset{(127)}{\leq} \quad 180\gamma^2 \exp(-\gamma\mu(T-1)) \sum_{l=0}^{T-1} \frac{\sigma^2}{m_l \exp(-\gamma\mu l)}$$

$$\overset{(111),T\leq K+1}{\leq} \quad \frac{1}{7} \exp(-\gamma\mu T) R^2. \tag{138}$$

**Upper bound for ⑥.** First of all, we have

$$2\gamma^2(1-\gamma\mu)^{T-1-l}\mathbb{E}_{\boldsymbol{\xi}_1^l,\boldsymbol{\xi}_2^l}\left[\|\theta_l^u\|^2 + 4\|\omega_l^u\|^2 - \mathbb{E}_{\boldsymbol{\xi}_2^l}\left[\|\theta_l^u\|^2\right] - 4\mathbb{E}_{\boldsymbol{\xi}_1^l}\left[\|\omega_l^u\|^2\right]\right] = 0.$$

Next, the summands in ⑥ are bounded with probability 1:

$$2\gamma^2(1-\gamma\mu)^{T-1-l}\left|\|\theta_l^u\|^2 + 4\|\omega_l^u\|^2 - \mathbb{E}_{\boldsymbol{\xi}_2^l}\left[\|\theta_l^u\|^2\right] - 4\mathbb{E}_{\boldsymbol{\xi}_1^l}\left[\|\omega_l^u\|^2\right]\right| \quad \overset{(124)}{\leq} \quad \frac{80\gamma^2 \exp(-\gamma\mu T)\lambda_l^2}{\exp(-\gamma\mu(1+l))}$$

$$\overset{(110)}{\leq} \quad \frac{\exp(-\gamma\mu T)R^2}{7\ln\frac{6(K+1)}{\beta}}$$

$$\overset{\text{def}}{=} \quad c. \tag{139}$$

Moreover, these summands have bounded conditional variances $\widehat{\sigma}_l^2 \overset{\text{def}}{=}$
$\mathbb{E}_{\boldsymbol{\xi}_1^l,\boldsymbol{\xi}_2^l}\left[4\gamma^4(1-\gamma\mu)^{2T-2-2l}\left|\|\theta_l^u\|^2 + 4\|\omega_l^u\|^2 - \mathbb{E}_{\boldsymbol{\xi}_2^l}\left[\|\theta_l^u\|^2\right] - 4\mathbb{E}_{\boldsymbol{\xi}_1^l}\left[\|\omega_l^u\|^2\right]\right|^2\right]$:

$$\widehat{\sigma}_l^2 \quad \overset{(139)}{\leq} \quad \frac{2\gamma^2 \exp(-2\gamma\mu T)R^2}{7\exp(-\gamma\mu(1+l))\ln\frac{6(K+1)}{\beta}}\mathbb{E}_{\boldsymbol{\xi}_1^l,\boldsymbol{\xi}_2^l}\left[\left|\|\theta_l^u\|^2 + 4\|\omega_l^u\|^2 - \mathbb{E}_{\boldsymbol{\xi}_2^l}\left[\|\theta_l^u\|^2\right] - 4\mathbb{E}_{\boldsymbol{\xi}_1^l}\left[\|\omega_l^u\|^2\right]\right|\right]$$

$$\leq \quad \frac{4\gamma^2 \exp(-2\gamma\mu T)R^2}{7\exp(-\gamma\mu(1+l))\ln\frac{6(K+1)}{\beta}}\mathbb{E}_{\boldsymbol{\xi}_1^l,\boldsymbol{\xi}_2^l}\left[\|\theta_l^u\|^2 + 4\|\omega_l^u\|^2\right]. \tag{140}$$

That is, sequence $\left\{2\gamma^2(1-\gamma\mu)^{T-1-l}\left(\|\theta_l^u\|^2 + 4\|\omega_l^u\|^2 - \mathbb{E}_{\boldsymbol{\xi}_2^l}\left[\|\theta_l^u\|^2\right] - 4\mathbb{E}_{\boldsymbol{\xi}_1^l}\left[\|\omega_l^u\|^2\right]\right)\right\}_{l\geq 0}$
is a bounded martingale difference sequence having bounded conditional variances
$\{\widehat{\sigma}_l^2\}_{l\geq 0}$. Applying Bernstein's inequality (Lemma B.1) with $X_l = 2\gamma^2(1-\gamma\mu)^{T-1-l}\left(\|\theta_l^u\|^2 + 4\|\omega_l^u\|^2 - \mathbb{E}_{\boldsymbol{\xi}_2^l}\left[\|\theta_l^u\|^2\right] - 4\mathbb{E}_{\boldsymbol{\xi}_1^l}\left[\|\omega_l^u\|^2\right]\right)$, $c$ defined in (139), $b = \frac{1}{7}\exp(-\gamma\mu T)R^2$, $G = \frac{\exp(-2\gamma\mu T)R^4}{294\ln\frac{6(K+1)}{\beta}}$, we get that

$$\mathbb{P}\left\{|⑥| > \frac{1}{7}\exp(-\gamma\mu T)R^2 \text{ and } \sum_{l=0}^{T-1}\widehat{\sigma}_l^2 \leq \frac{\exp(-2\gamma\mu T)R^4}{294\ln\frac{6(K+1)}{\beta}}\right\} \leq 2\exp\left(-\frac{b^2}{2G + 2cb/3}\right) = \frac{\beta}{3(K+1)}.$$

In other words, $\mathbb{P}\{E_⑥\} \geq 1 - \frac{\beta}{3(K+1)}$, where probability event $E_⑥$ is defined as

$$E_⑥ = \left\{\text{either} \quad \sum_{l=0}^{T-1}\widehat{\sigma}_l^2 > \frac{\exp(-2\gamma\mu T)R^4}{294\ln\frac{6(K+1)}{\beta}} \quad \text{or} \quad |⑥| \leq \frac{1}{7}\exp(-\gamma\mu T)R^2\right\}. \tag{141}$$

Moreover, we notice here that probability event $E_{T-1}$ implies that

$$\sum_{l=0}^{T-1}\widehat{\sigma}_l^2 \quad \overset{(140)}{\leq} \quad \frac{4\gamma^2 \exp(-\gamma\mu(2T-1))R^2}{7\ln\frac{6(K+1)}{\beta}} \sum_{l=0}^{T-1}\frac{\mathbb{E}_{\boldsymbol{\xi}_1^l,\boldsymbol{\xi}_2^l}\left[\|\theta_l^u\|^2 + 4\|\omega_l^u\|^2\right]}{\exp(-\gamma\mu l)}$$

$$\overset{(127),T\leq K+1}{\leq} \quad \frac{360\gamma^2 \exp(-\gamma\mu(2T-1))R^2\sigma^2}{7\ln\frac{6(K+1)}{\beta}} \sum_{l=0}^{K}\frac{1}{m_l \exp(-\gamma\mu l)}$$

$$\overset{(111)}{\leq} \quad \frac{\exp(-2\gamma\mu T)R^4}{294\ln\frac{6(K+1)}{\beta}}. \tag{142}$$

**Upper bound for ⑦.** Probability event $E_{T-1}$ implies

$$
\begin{aligned}
⑦ \quad &= \quad 2\gamma^2 \sum_{l=0}^{T-1} \exp(-\gamma\mu(T-1-l)) \left( \|\theta_l^b\|^2 + 4\|\omega_l^b\|^2 \right) \\
&\overset{(125)}{\leq} \quad 160\gamma^2 \exp(-\gamma\mu(T-1)) \sum_{l=0}^{T-1} \frac{\sigma^4}{m_l^2 \lambda_l^2 \exp(-\gamma\mu l)} \\
&\overset{(110)}{=} \quad 2304000\gamma^4 \exp(-\gamma\mu(T-3)) \sum_{l=0}^{T-1} \frac{\sigma^4 \ln^2 \frac{6(K+1)}{\beta}}{m_l^2 R^2 \exp(-2\gamma\mu l)} \\
&\overset{(111), T \leq K+1}{\leq} \quad \frac{1}{7} \exp(-\gamma\mu T) R^2. \quad\quad\quad\quad (143)
\end{aligned}
$$

**Final derivation.** Putting all bounds together, we get that $E_{T-1}$ implies

$$
R_T^2 \overset{(123)}{\leq} \exp(-\gamma\mu T) R^2 + ① + ② + ③ + ④ + ⑤ + ⑥ + ⑦,
$$

$$
② \overset{(132)}{\leq} \frac{1}{7} \exp(-\gamma\mu T) R^2, \quad ④ \overset{(137)}{\leq} \frac{1}{7} \exp(-\gamma\mu T) R^2,
$$

$$
⑤ \overset{(138)}{\leq} \frac{1}{7} \exp(-\gamma\mu T) R^2, \quad ⑦ \overset{(143)}{\leq} \frac{1}{7} \exp(-\gamma\mu T) R^2,
$$

$$
\sum_{l=0}^{T-1} \sigma_l^2 \overset{(131)}{\leq} \frac{\exp(-2\gamma\mu T) R^4}{294 \ln \frac{6(K+1)}{\beta}}, \quad \sum_{l=0}^{T-1} \widetilde{\sigma}_l^2 \overset{(136)}{\leq} \frac{\exp(-2\gamma\mu T) R^4}{294 \ln \frac{6(K+1)}{\beta}}, \quad \sum_{l=0}^{T-1} \widehat{\sigma}_l^2 \overset{(142)}{\leq} \frac{\exp(-2\gamma\mu T) R^4}{294 \ln \frac{6(K+1)}{\beta}}.
$$

Moreover, in view of (130), (135), (141), and our induction assumption, we have

$$
\mathbb{P}\{E_{T-1}\} \geq 1 - \frac{(T-1)\beta}{K+1},
$$

$$
\mathbb{P}\{E_①\} \geq 1 - \frac{\beta}{3(K+1)}, \quad \mathbb{P}\{E_③\} \geq 1 - \frac{\beta}{3(K+1)}, \quad \mathbb{P}\{E_⑥\} \geq 1 - \frac{\beta}{3(K+1)},
$$

where probability events $E_①$, $E_③$, and $E_⑥$ are defined as

$$
E_① = \left\{ \text{either} \quad \sum_{l=0}^{T-1} \sigma_l^2 > \frac{\exp(-2\gamma\mu T) R^4}{294 \ln \frac{6(K+1)}{\beta}} \quad \text{or} \quad |①| \leq \frac{1}{7} \exp(-\gamma\mu T) R^2 \right\},
$$

$$
E_③ = \left\{ \text{either} \quad \sum_{l=0}^{T-1} \widetilde{\sigma}_l^2 > \frac{\exp(-2\gamma\mu T) R^4}{294 \ln \frac{6(K+1)}{\beta}} \quad \text{or} \quad |③| \leq \frac{1}{7} \exp(-\gamma\mu T) R^2 \right\},
$$

$$
E_⑥ = \left\{ \text{either} \quad \sum_{l=0}^{T-1} \widehat{\sigma}_l^2 > \frac{\exp(-2\gamma\mu T) R^4}{294 \ln \frac{6(K+1)}{\beta}} \quad \text{or} \quad |⑥| \leq \frac{1}{7} \exp(-\gamma\mu T) R^2 \right\}.
$$

Putting all of these inequalities together, we obtain that probability event $E_{T-1} \cap E_① \cap E_③ \cap E_⑥$ implies

$$
\begin{aligned}
R_T^2 \quad &\overset{(123)}{\leq} \quad \exp(-\gamma\mu T) R^2 + ① + ② + ③ + ④ + ⑤ + ⑥ + ⑦ \\
&\leq \quad 2 \exp(-\gamma\mu T) R^2.
\end{aligned}
$$

Moreover, union bound for the probability events implies

$$
\mathbb{P}\{E_T\} \geq \mathbb{P}\{E_{T-1} \cap E_① \cap E_③ \cap E_⑥\} = 1 - \mathbb{P}\{\overline{E}_{T-1} \cup \overline{E}_① \cup \overline{E}_③ \cup \overline{E}_⑥\} \geq 1 - \frac{T\beta}{K+1}. \quad (144)
$$

This is exactly what we wanted to prove (see the paragraph after inequality (113)). In particular, with probability at least $1 - \beta$ satisfy we have

$$
\|x^{K+1} - x^*\|^2 \leq 2 \exp(-\gamma\mu(K+1)) R^2,
$$

which finishes the proof. $\qquad\qquad\qquad\qquad\qquad\qquad\qquad\qquad\qquad\qquad\qquad\qquad\square$

**Corollary C.3.** *Let the assumptions of Theorem C.3 hold. Then, the following statements hold.*

1. ***Large stepsize/large batch.*** *The choice of stepsize and batchsize*

$$\gamma = \frac{1}{650L \ln \frac{6(K+1)}{\beta}}, \quad m_k = \max\left\{1, \frac{264600\gamma^2(K+1)\sigma^2 \ln \frac{6(K+1)}{\beta}}{\exp(-\gamma\mu k)R^2}\right\} \quad (145)$$

*satisfies conditions (109) and (111). With such choice of $\gamma, m_k$, and the choice of $\lambda_k$ as in (110), the iterates produced by* clipped-SEG *after $K$ iterations with probability at least $1 - \beta$ satisfy*

$$\|x^{K+1} - x^*\|^2 \leq 2\exp\left(-\frac{\mu(K+1)}{650L \ln \frac{6(K+1)}{\beta}}\right) R^2. \quad (146)$$

*In particular, to guarantee $\|x^{K+1} - x^*\|^2 \leq \varepsilon$ with probability at least $1 - \beta$ for some $\varepsilon > 0$* clipped-SEG *requires*

$$\mathcal{O}\left(\frac{L}{\mu} \ln\left(\frac{R^2}{\varepsilon}\right) \ln\left(\frac{L}{\mu\beta} \ln\left(\frac{R^2}{\varepsilon}\right)\right)\right) \text{ iterations,} \quad (147)$$

$$\mathcal{O}\left(\max\left\{\frac{L}{\mu}, \frac{\sigma^2}{\mu^2\varepsilon}\right\} \ln\left(\frac{R^2}{\varepsilon}\right) \ln\left(\frac{L}{\mu\beta} \ln\left(\frac{R^2}{\varepsilon}\right)\right)\right) \text{ oracle calls.} \quad (148)$$

2. ***Small stepsize/small batch.*** *The choice of stepsize and batchsize*

$$\gamma = \min\left\{\frac{1}{650L \ln \frac{6(K+1)}{\beta}}, \frac{\ln(B_K)}{\mu(K+1)}\right\}, \quad m_k \equiv 1 \quad (149)$$

*satisfies conditions (109) and (111), where $B_K = \max\left\{2, \frac{(K+1)\mu^2 R^2}{264600\sigma^2 \ln\left(\frac{6(K+1)}{\beta}\right) \ln^2(B_K)}\right\} =$*

$$\mathcal{O}\left(\max\left\{2, \frac{(K+1)\mu^2 R^2}{264600\sigma^2 \ln\left(\frac{6(K+1)}{\beta}\right) \ln^2\left(\max\left\{2, \frac{(K+1)\mu^2 R^2}{264600\sigma^2 \ln\left(\frac{6(K+1)}{\beta}\right)}\right\}\right)}\right\}\right).$$ *With such choice*

*of $\gamma, m_k$, and the choice of $\lambda_k$ as in (110), the iterates produced by* clipped-SEG *after $K$ iterations with probability at least $1 - \beta$ satisfy*

$$\|x^{K+1} - x^*\|^2 \leq \max\left\{2\exp\left(-\frac{\mu(K+1)}{650L \ln \frac{6(K+1)}{\beta}}\right) R^2, \frac{529200\sigma^2 \ln\left(\frac{6(K+1)}{\beta}\right) \ln^2(B_K)}{\mu^2(K+1)}\right\}. \quad (150)$$

*In particular, to guarantee $\|x^{K+1} - x^*\|^2 \leq \varepsilon$ with probability at least $1 - \beta$ for some $\varepsilon > 0$* clipped-SEG *requires*

$$\mathcal{O}\left(\max\left\{\frac{L}{\mu} \ln\left(\frac{R^2}{\varepsilon}\right) \ln\left(\frac{L}{\mu\beta} \ln\left(\frac{R^2}{\varepsilon}\right)\right), \frac{\sigma^2}{\mu^2\varepsilon} \ln\left(\frac{\sigma^2}{\mu^2\varepsilon\beta}\right) \ln^2(B_\varepsilon)\right\}\right) \quad (151)$$

*iterations/oracle calls, where*

$$B_\varepsilon = \max\left\{2, \frac{R^2}{\varepsilon \ln\left(\frac{\sigma^2}{\mu^2\varepsilon\beta}\right) \ln^2\left(\max\left\{2, \frac{R^2}{\varepsilon \ln\left(\frac{\sigma^2}{\mu^2\varepsilon\beta}\right)}\right\}\right)}\right\}.$$

*Proof.*     1. **Large stepsize/large batch.** First of all, it is easy to see that the choice of $\gamma$ and $m_k$ from (145) satisfies conditions (109) and (111). Therefore, applying Theorem C.3, we derive that with probability at least $1 - \beta$

$$\|x^{K+1} - x^*\|^2 \leq 2\exp(-\gamma\mu(K+1))R^2 \overset{(145)}{=} 2\exp\left(-\frac{\mu(K+1)}{650L \ln \frac{6(K+1)}{\beta}}\right) R^2.$$

To guarantee $\|x^{K+1} - x^*\|^2 \le \varepsilon$, we choose $K$ in such a way that the right-hand side of the above inequality is smaller than $\varepsilon$ that gives

$$K = \mathcal{O}\left(\frac{L}{\mu}\ln\left(\frac{R^2}{\varepsilon}\right)\ln\left(\frac{L}{\mu\beta}\ln\left(\frac{R^2}{\varepsilon}\right)\right)\right).$$

The total number of oracle calls equals

$$\sum_{k=0}^{K} 2m_k \overset{(145)}{=} 2\sum_{k=0}^{K} \max\left\{1, \frac{264600\gamma^2(K+1)\sigma^2\ln\frac{6(K+1)}{\beta}}{\exp(-\gamma\mu k)R^2}\right\}$$

$$= \mathcal{O}\left(\max\left\{K, \frac{\gamma(K+1)\exp(\gamma\mu(K+1))\sigma^2\ln\frac{6(K+1)}{\beta}}{\mu R^2}\right\}\right)$$

$$= \mathcal{O}\left(\max\left\{\frac{L}{\mu}, \frac{\sigma^2}{\mu^2\varepsilon}\right\}\ln\left(\frac{R^2}{\varepsilon}\right)\ln\left(\frac{L}{\mu\beta}\ln\left(\frac{R^2}{\varepsilon}\right)\right)\right).$$

2. **Small stepsize/small batch.** First of all, we verify that the choice of $\gamma$ and $m_k$ from (149) satisfies conditions (109) and (111): (109) trivially holds and (111) holds since for all $k = 0, \ldots, K$

$$\frac{264600\gamma^2(K+1)\sigma^2\ln\frac{6(K+1)}{\beta}}{\exp(-\gamma\mu k)R^2} \le \frac{264600\gamma^2(K+1)\sigma^2\ln\frac{6(K+1)}{\beta}}{\exp(-\gamma\mu(K+1))R^2}$$

$$\overset{(149)}{\le} \frac{264600\ln^2(B_K)\exp(\gamma\mu(K+1))\sigma^2\ln\frac{6(K+1)}{\beta}}{\mu^2(K+1)R^2}$$

$$\overset{(149)}{\le} 1.$$

Therefore, applying Theorem C.3, we derive that with probability at least $1 - \beta$

$$\|x^{K+1} - x^*\|^2 \le 2\exp(-\gamma\mu(K+1))R^2$$

$$\overset{(149)}{=} \max\left\{2\exp\left(-\frac{\mu(K+1)}{650L\ln\frac{6(K+1)}{\beta}}\right)R^2, \frac{2R^2}{B_K}\right\}$$

$$= \max\left\{2\exp\left(-\frac{\mu(K+1)}{650L\ln\frac{6(K+1)}{\beta}}\right)R^2, \frac{529200\sigma^2\ln\left(\frac{6(K+1)}{\beta}\right)\ln^2(B_K)}{\mu^2(K+1)}\right\}.$$

To guarantee $\|x^{K+1} - x^*\|^2 \le \varepsilon$, we choose $K$ in such a way that the right-hand side of the above inequality is smaller than $\varepsilon$ that gives $K$ of the order

$$\mathcal{O}\left(\max\left\{\frac{L}{\mu}\ln\left(\frac{R^2}{\varepsilon}\right)\ln\left(\frac{L}{\mu\beta}\ln\left(\frac{R^2}{\varepsilon}\right)\right), \frac{\sigma^2}{\mu^2\varepsilon}\ln\left(\frac{\sigma^2}{\mu^2\varepsilon\beta}\right)\ln^2(B_\varepsilon)\right\}\right),$$

where

$$B_\varepsilon = \max\left\{2, \frac{R^2}{\varepsilon\ln\left(\frac{\sigma^2}{\mu^2\varepsilon\beta}\right)\ln^2\left(\max\left\{2, \frac{R^2}{\varepsilon\ln\left(\frac{\sigma^2}{\mu^2\varepsilon\beta}\right)}\right\}\right)}\right\}.$$

The total number of oracle calls equals $\sum_{k=0}^{K} 2m_k = 2(K+1)$.

$\square$

# D    Clipped Stochastic Gradient Descent-Ascent: Missing Proofs and Details

## D.1    Monotone Star-Cocoercive Case

**Lemma D.1.** *Let Assumption 1.3 hold for $Q = B_{2R}(x^*)$, where $R \geq R_0 \stackrel{def}{=} \|x^0 - x^*\|$ and $0 < \gamma \leq {}^2/\ell$. If $x^k$ lies in $B_{2R}(x^*)$ for all $k = 0, 1, \ldots, K$ for some $K \geq 0$, then for all $u \in B_{3R}(x^*)$ the iterates produced by* clipped-SGDA *satisfy*

$$2\gamma\langle F(u), x_{avg}^K - u\rangle \quad \leq \quad \frac{\|x^0 - u\|^2 - \|x^{K+1} - u\|^2}{K+1}$$

$$+\frac{2\gamma}{K+1}\sum_{k=0}^{K}\langle x^k - u - \gamma F(x^k), \omega_k\rangle$$

$$+\frac{\gamma^2}{K+1}\sum_{k=0}^{K}\left(\|F(x^k)\|^2 + \|\omega_k\|^2\right), \tag{152}$$

$$x_{avg}^K \quad \stackrel{def}{=} \quad \frac{1}{K+1}\sum_{k=0}^{K}x^k, \tag{153}$$

$$\omega_k \quad \stackrel{def}{=} \quad F(x^k) - \widetilde{F}_{\boldsymbol{\xi}^k}(x^k). \tag{154}$$

*Proof.* Using the update rule of clipped-SGDA, we obtain

$$\|x^{k+1} - u\|^2 = \|x^k - u\|^2 - 2\gamma\langle x^k - u, \widetilde{F}_{\boldsymbol{\xi}^k}(x^k)\rangle + \gamma^2\|\widetilde{F}_{\boldsymbol{\xi}^k}(x^k)\|^2$$

$$= \|x^k - u\|^2 - 2\gamma\langle x^k - u, F(x^k)\rangle + 2\gamma\langle x^k - u, \omega_k\rangle$$

$$+\gamma^2\|F(x^k)\|^2 - 2\gamma^2\langle F(x^k), \omega_k\rangle + \gamma^2\|\omega_k\|^2$$

$$\stackrel{\text{(Mon)}}{\leq} \|x^k - u\|^2 - 2\gamma\langle x^k - u, F(u)\rangle + 2\gamma\langle x^k - u - \gamma F(x^k), \omega_k\rangle$$

$$+\gamma^2\left(\|F(x^k)\|^2 + \|\omega_k\|^2\right).$$

Rearranging the terms, we derive

$$2\gamma\langle F(u), x^k - u\rangle \leq \|x^k - u\|^2 - \|x^{k+1} - u\|^2 + 2\gamma\langle x^k - u - \gamma F(x^k), \omega_k\rangle$$

$$+\gamma^2\left(\|F(x^k)\|^2 + \|\omega_k\|^2\right).$$

Finally, we sum up the above inequality for $k = 0, 1, \ldots, K$ and divide both sides of the result by $(K+1)$:

$$2\gamma\langle F(u), x_{avg}^K - u\rangle \quad \leq \quad \frac{1}{K+1}\sum_{k=0}^{K}\left(\|x^k - u\|^2 - \|x^{k+1} - u\|^2\right)$$

$$+\frac{2\gamma}{K+1}\sum_{k=0}^{K}\langle x^k - u - \gamma F(x^k), \omega_k\rangle$$

$$+\frac{\gamma^2}{K+1}\sum_{k=0}^{K}\left(\|F(x^k)\|^2 + \|\omega_k\|^2\right)$$

$$= \quad \frac{\|x^0 - u\|^2 - \|x^{K+1} - u\|^2}{K+1}$$

$$+\frac{2\gamma}{K+1}\sum_{k=0}^{K}\langle x^k - u - \gamma F(x^k), \omega_k\rangle$$

$$+\frac{\gamma^2}{K+1}\sum_{k=0}^{K}\left(\|F(x^k)\|^2 + \|\omega_k\|^2\right).$$

This finishes the proof. $\qquad\square$

We also derive the following lemma, which we use in the analysis of the star-cocoercive case as well.

**Lemma D.2.** *Let Assumption 1.6 hold for $Q = B_{2R}(x^*)$, where $R \geq R_0 \stackrel{def}{=} \|x^0 - x^*\|$ and $0 < \gamma \leq 2/\ell$. If $x^k$ lies in $B_{2R}(x^*)$ for all $k = 0, 1, \ldots, K$ for some $K \geq 0$, then the iterates produced by* clipped-SGDA *satisfy*

$$
\frac{\gamma}{K+1}\left(\frac{2}{\ell} - \gamma\right)\sum_{k=0}^{K}\|F(x^k)\|^2 \quad \leq \quad \frac{\|x^0 - x^*\|^2 - \|x^{K+1} - x^*\|^2}{K+1}
$$

$$
+\frac{2\gamma}{K+1}\sum_{k=0}^{K}\langle x^k - x^* - \gamma F(x^k), \omega_k\rangle
$$

$$
+\frac{\gamma^2}{K+1}\sum_{k=0}^{K}\|\omega_k\|^2, \tag{155}
$$

*where $\omega_k$ is defined in (154).*

*Proof.* Using the update rule of clipped-SGDA, we obtain

$$
\begin{aligned}
\|x^{k+1} - x^*\|^2 &= \|x^k - x^*\|^2 - 2\gamma\langle x^k - x^*, \widetilde{F}_{\boldsymbol{\xi}^k}(x^k)\rangle + \gamma^2\|\widetilde{F}_{\boldsymbol{\xi}^k}(x^k)\|^2 \\
&= \|x^k - x^*\|^2 - 2\gamma\langle x^k - x^*, F(x^k)\rangle + 2\gamma\langle x^k - x^*, \omega_k\rangle \\
&\quad +\gamma^2\|F(x^k)\|^2 - 2\gamma^2\langle F(x^k), \omega_k\rangle + \gamma^2\|\omega_k\|^2 \\
&\stackrel{(SC)}{\leq} \|x^k - x^*\|^2 + 2\gamma\langle x^k - x^*, \omega_k\rangle - 2\gamma^2\langle F(x^k), \omega_k\rangle \\
&\quad +\gamma\left(\gamma - \frac{2}{\ell}\right)\|F(x^k)\|^2 + \gamma^2\|\omega_k\|^2.
\end{aligned}
$$

Since $0 < \gamma \leq 2/\ell$, we have $\gamma\left(2/\ell - \gamma\right)\|F(x^k)\|^2 \geq 0$ and, rearranging the terms, we derive

$$
\begin{aligned}
\gamma\left(\frac{2}{\ell} - \gamma\right)\|F(x^k)\|^2 &\leq \|x^k - x^*\|^2 - \|x^{k+1} - x^*\|^2 + 2\gamma\langle x^k - x^*, \omega_k\rangle \\
&\quad -2\gamma^2\langle F(x^k), \omega_k\rangle + \gamma^2\|\omega_k\|^2.
\end{aligned}
$$

Finally, we sum up the above inequality for $k = 0, 1, \ldots, K$ and divide both sides of the result by $(K+1)$:

$$
\begin{aligned}
\frac{\gamma}{K+1}\left(\frac{2}{\ell} - \gamma\right)\sum_{k=0}^{K}\|F(x^k)\|^2 &\leq \frac{1}{K+1}\sum_{k=0}^{K}\left(\|x^k - x^*\|^2 - \|x^{k+1} - x^*\|^2\right) + \frac{\gamma^2}{K+1}\sum_{k=0}^{K}\|\omega_k\|^2 \\
&\quad +\frac{2\gamma}{K+1}\sum_{k=0}^{K}\langle x^k - x^*, \omega_k\rangle - \frac{2\gamma^2}{K+1}\sum_{k=0}^{K}\langle F(x^k), \omega_k\rangle \\
&= \frac{\|x^0 - x^*\|^2 - \|x^{K+1} - x^*\|^2}{K+1} + \frac{\gamma^2}{K+1}\sum_{k=0}^{K}\|\omega_k\|^2 \\
&\quad +\frac{2\gamma}{K+1}\sum_{k=0}^{K}\langle x^k - x^*, \omega_k\rangle - \frac{2\gamma^2}{K+1}\sum_{k=0}^{K}\langle F(x^k), \omega_k\rangle.
\end{aligned}
$$

This finishes the proof. $\qquad\square$

**Theorem D.1.** *Let Assumptions 1.1, 1.3, 1.6, hold for $Q = B_{2R}(x^*)$, where $R \geq R_0 \stackrel{def}{=} \|x^0 - x^*\|$, and*

$$
\gamma \leq \frac{1}{170\ell \ln\frac{6(K+1)}{\beta}}, \tag{156}
$$

$$
\lambda = \frac{R}{60\gamma \ln\frac{6(K+1)}{\beta}}, \tag{157}
$$

$$
m \geq \max\left\{1, \frac{97200(K+1)\gamma^2\sigma^2\ln\frac{6(K+1)}{\beta}}{R^2}\right\}, \tag{158}
$$

*for some $K \geq 0$ and $\beta \in (0, 1]$ such that $\ln \frac{6(K+1)}{\beta} \geq 1$. Then, after $K$ iterations the iterates produced by* clipped-SGDA *with probability at least $1 - \beta$ satisfy*

$$Gap_R(x_{avg}^K) \leq \frac{9R^2}{2\gamma(K+1)}. \tag{159}$$

*Proof.* We introduce new notation: $R_k = \|x^k - x^*\|$ for all $k \geq 0$. The proof is based on the induction. In particular, for each $k = 0, \ldots, K + 1$ we define the probability event $E_k$ as follows: inequalities

$$\|x^t - x^*\|^2 \leq 2R^2 \quad \text{and} \quad \gamma \left\| \sum_{l=0}^{t-1} \omega_l \right\| \leq R \tag{160}$$

hold for $t = 0, 1, \ldots, k$ simultaneously. Our goal is to prove that $\mathbb{P}\{E_k\} \geq 1 - {k\beta}/{(K+1)}$ for all $k = 0, 1, \ldots, K + 1$. We use the induction to show this statement. For $k = 0$ the statement is trivial since $R_0^2 \leq 2R^2$ by definition and $\sum_{l=0}^{-1} \omega_l = 0$. Next, assume that the statement holds for $k = T \leq K$, i.e., we have $\mathbb{P}\{E_T\} \geq 1 - {T\beta}/{(K+1)}$. We need to prove that $\mathbb{P}\{E_{T+1}\} \geq 1 - {(T+1)\beta}/{(K+1)}$. Let us notice that probability event $E_T$ implies $x^t \in B_{2R}(x^*)$ for all $t = 0, 1, \ldots, T$. This means that the assumptions of Lemma D.2 hold and we have that probability event $E_T$ implies ($\gamma < 1/\ell$)

$$\frac{\gamma}{\ell(T+1)} \sum_{t=0}^{T} \|F(x^t)\|^2 \quad \leq \quad \frac{\|x^0 - x^*\|^2 - \|x^{T+1} - x^*\|^2}{T+1}$$

$$+ \frac{2\gamma}{T+1} \sum_{t=0}^{T} \langle x^t - x^* - \gamma F(x^t), \omega_t \rangle$$

$$+ \frac{\gamma^2}{T+1} \sum_{t=0}^{T} \|\omega_t\|^2 \tag{161}$$

and

$$\|F(x^t)\| \overset{\text{(SC)}}{\leq} \ell \|x^t - x^*\| \overset{\text{(160)}}{\leq} \sqrt{2}\ell R \overset{\text{(156),(157)}}{\leq} \frac{\lambda}{2} \tag{162}$$

for all $t = 0, 1, \ldots, T$. From (161) we have

$$R_{T+1}^2 \leq R_0^2 + 2\gamma \sum_{t=0}^{T} \langle x^t - x^* - \gamma F(x^t), \omega_t \rangle + \gamma^2 \sum_{t=0}^{T} \|\omega_t\|^2.$$

Next, we notice that

$$\|x^t - x^* - \gamma F(x^t)\| \quad \leq \quad \|x^t - x^*\| + \gamma\|F(x^t)\| \overset{\text{(SC),(160)}}{\leq} 2R + \gamma\ell\|x^t - x^*\|$$

$$\overset{\text{(160)}}{\leq} \quad 2R + 2R\gamma\ell \overset{\text{(156)}}{\leq} 3R, \tag{163}$$

for all $t = 0, 1, \ldots, T$. Consider random vectors

$$\eta_t = \begin{cases} x^t - x^* - \gamma F(x^t), & \text{if } \|x^t - x^* - \gamma F(x^t)\| \leq 3R, \\ 0, & \text{otherwise,} \end{cases}$$

for all $t = 0, 1, \ldots, T$. We notice that $\eta_t$ is bounded with probability 1:

$$\|\eta_t\| \leq 3R \tag{164}$$

for all $t = 0, 1, \ldots, T$. Moreover, in view of (163), probability event $E_T$ implies $\eta_t = x^t - x^* - \gamma F(x^t)$ for all $t = 0, 1, \ldots, T$. Therefore, $E_T$ implies

$$R_{T+1}^2 \leq R^2 + 2\gamma \sum_{t=0}^{T} \langle \eta_t, \omega_t \rangle + \gamma^2 \sum_{t=0}^{T} \|\omega_t\|^2.$$

To continue our derivation we introduce new notation:

$$\omega_t^u \stackrel{\text{def}}{=} \mathbb{E}_{\boldsymbol{\xi}^t}\left[\widetilde{F}_{\boldsymbol{\xi}^t}(x^t)\right] - \widetilde{F}_{\boldsymbol{\xi}^t}(x^t), \quad \omega_t^b \stackrel{\text{def}}{=} F(x^t) - \mathbb{E}_{\boldsymbol{\xi}^t}\left[\widetilde{F}_{\boldsymbol{\xi}^t}(x^t)\right] \tag{165}$$

By definition we have $\omega_t = \omega_t^u + \omega_t^b$ for all $t = 0, \ldots, T$. Using the introduced notation, we continue our derivation as follows: $E_T$ implies

$$
\begin{aligned}
R_{T+1}^2 &\leq R^2 + \underbrace{2\gamma \sum_{t=0}^{T}\langle \eta_t, \omega_t^u \rangle}_{\text{①}} + \underbrace{2\gamma \sum_{t=0}^{T}\langle \eta_t, \omega_t^b \rangle}_{\text{②}} + \underbrace{2\gamma^2 \sum_{t=0}^{T}\left(\mathbb{E}_{\boldsymbol{\xi}^t}\left[\|\omega_t^u\|^2\right]\right)}_{\text{③}} \\
&\quad + \underbrace{2\gamma^2 \sum_{t=0}^{T}\left(\|\omega_t^u\|^2 - \mathbb{E}_{\boldsymbol{\xi}^t}\left[\|\omega_t^u\|^2\right]\right)}_{\text{④}} + \underbrace{2\gamma^2 \sum_{t=0}^{T}\left(\|\omega_t^b\|^2\right)}_{\text{⑤}}.
\end{aligned}
\tag{166}
$$

We emphasize that the above inequality does not rely on monotonicity of $F$.

As we notice above, $E_T$ implies $x^t \in B_{2R}(x^*)$ for all $t = 0, 1, \ldots, T$. This means that the assumptions of Lemma D.1 hold and we have that probability event $E_T$ implies

$$
\begin{aligned}
2\gamma(T+1)\mathtt{Gap}_R(x_{\text{avg}}^T) &\leq \max_{u \in B_R(x^*)}\left\{\|x^0 - u\|^2 + 2\gamma \sum_{t=0}^{T}\langle x^t - u - \gamma F(x^t), \omega_t\rangle\right\} \\
&\quad + \gamma^2 \sum_{t=0}^{T}\left(\|F(x^t)\|^2 + \|\omega_t\|^2\right), \\
&= \max_{u \in B_R(x^*)}\left\{\|x^0 - u\|^2 + 2\gamma \sum_{t=0}^{T}\langle x^* - u, \omega_t\rangle\right\} \\
&\quad + 2\gamma \sum_{t=0}^{T}\langle x^t - x^* - \gamma F(x^t), \omega_t\rangle \\
&\quad + \gamma^2 \sum_{t=0}^{T}\left(\|F(x^t)\|^2 + \|\omega_t\|^2\right).
\end{aligned}
$$

We notice that $E_T$ implies $\eta_t = x^t - x^* - \gamma F(x^t)$ for all $t = 0, 1, \ldots, T$ as well as (161) and $\gamma < 1/\ell$. Therefore, probability event $E_T$ implies

$$
\begin{aligned}
2\gamma(T+1)\mathtt{Gap}_R(x_{\text{avg}}^T) &\leq \max_{u \in B_R(x^*)}\left\{\|x^0 - u\|^2\right\} + 2\gamma \max_{u \in B_R(x^*)}\left\{\sum_{t=0}^{T}\langle x^* - u, \omega_t\rangle\right\} \\
&\quad + 2\gamma \sum_{t=0}^{T}\langle \eta_t, \omega_t\rangle + \frac{\gamma}{\ell}\sum_{t=0}^{T}\|F(x^t)\|^2 + \gamma^2 \sum_{t=0}^{T}\|\omega_t\|^2 \\
&\leq 4R^2 + 2\gamma \max_{u \in B_R(x^*)}\left\{\left\langle x^* - u, \sum_{t=0}^{T}\omega_t\right\rangle\right\} \\
&\quad + R^2 + 4\gamma \sum_{t=0}^{T}\langle \eta_t, \omega_t\rangle + 2\gamma^2 \sum_{t=0}^{T}\|\omega_t\|^2 \\
&\leq 5R^2 + 2\gamma R\left\|\sum_{t=0}^{T}\omega_t\right\| + 2 \cdot (\text{①} + \text{②} + \text{③} + \text{④} + \text{⑤}), \tag{167}
\end{aligned}
$$

where ①, ②, ③, ④, ⑤ are defined in (166).

The rest of the proof is based on deriving good enough upper bounds for ①, ②, ③, ④, ⑤, i.e., we want to prove that $\text{①} + \text{②} + \text{③} + \text{④} + \text{⑤} \leq R^2$ and $2\gamma R\left\|\sum_{t=0}^{T}\omega_t\right\| \leq 2R^2$ with high probability.

Before we move on, we need to derive some useful inequalities for operating with $\omega_t^u, \omega_t^b$. First of all, Lemma B.2 implies that

$$\|\omega_t^u\| \le 2\lambda \tag{168}$$

for all $t = 0, 1, \ldots, T$. Next, since $\{\xi^{i,t}\}_{i=1}^m$ are independently sampled from $\mathcal{D}$, we have $\mathbb{E}_{\boldsymbol{\xi}^t}[F_{\boldsymbol{\xi}^t}(x^t)] = F(x^t)$, and

$$\mathbb{E}_{\boldsymbol{\xi}^t}\left[\|F_{\boldsymbol{\xi}^t}(x^t) - F(x^t)\|^2\right] = \frac{1}{m^2}\sum_{i=1}^m \mathbb{E}_{\xi^{i,t}}\left[\|F_{\xi^{i,t}}(x^t) - F(x^t)\|^2\right] \overset{(1)}{\le} \frac{\sigma^2}{m},$$

for all $l = 0, 1, \ldots, T$. Therefore, in view of Lemma B.2, $E_T$ implies that

$$\|\omega_t^b\| \le \frac{4\sigma^2}{m\lambda}, \tag{169}$$

$$\mathbb{E}_{\boldsymbol{\xi}^t}\left[\|\omega_t\|^2\right] \le \frac{18\sigma^2}{m}, \tag{170}$$

$$\mathbb{E}_{\boldsymbol{\xi}^t}\left[\|\omega_t^u\|^2\right] \le \frac{18\sigma^2}{m} \tag{171}$$

for all $l = 0, 1, \ldots, T$.

**Upper bound for ①.** Since $\mathbb{E}_{\boldsymbol{\xi}^t}[\omega_t^u] = 0$, we have

$$\mathbb{E}_{\boldsymbol{\xi}^t}\left[2\gamma\langle\eta_t, \omega_t^u\rangle\right] = 0.$$

Next, the summands in ① are bounded with probability 1:

$$|2\gamma\langle\eta_t, \omega_t^u\rangle| \le 2\gamma\|\eta_t\| \cdot \|\omega_t^u\| \overset{(164),(168)}{\le} 12\gamma R\lambda \overset{(157)}{\le} \frac{R^2}{5\ln\frac{6(K+1)}{\beta}} \overset{\text{def}}{=} c. \tag{172}$$

Moreover, these summands have bounded conditional variances $\sigma_t^2 \overset{\text{def}}{=} \mathbb{E}_{\boldsymbol{\xi}^t}\left[4\gamma^2\langle\eta_t, \omega_t^u\rangle^2\right]$:

$$\sigma_t^2 \le \mathbb{E}_{\boldsymbol{\xi}^t}\left[4\gamma^2\|\eta_t\|^2 \cdot \|\omega_t^u\|^2\right] \overset{(164)}{\le} 36\gamma^2 R^2 \mathbb{E}_{\boldsymbol{\xi}^t}\left[\|\omega_t^u\|^2\right]. \tag{173}$$

That is, sequence $\{2\gamma\langle\eta_t, \omega_t^u\rangle\}_{t\ge0}$ is a bounded martingale difference sequence having bounded conditional variances $\{\sigma_t^2\}_{t\ge0}$. Applying Bernstein's inequality (Lemma B.1) with $X_t = 2\gamma\langle\eta_t, \omega_t^u\rangle$, $c$ defined in (172), $b = \frac{R^2}{5}$, $G = \frac{R^4}{150\ln\frac{6(K+1)}{\beta}}$, we get that

$$\mathbb{P}\left\{|①| > \frac{R^2}{5} \text{ and } \sum_{t=0}^T \sigma_t^2 \le \frac{R^4}{150\ln\frac{6(K+1)}{\beta}}\right\} \le 2\exp\left(-\frac{b^2}{2G + \frac{2cb}{3}}\right) = \frac{\beta}{3(K+1)}.$$

In other words, $\mathbb{P}\{E_①\} \ge 1 - \frac{\beta}{3(K+1)}$, where probability event $E_①$ is defined as

$$E_① = \left\{\text{either} \quad \sum_{t=0}^T \sigma_t^2 > \frac{R^4}{150\ln\frac{6(K+1)}{\beta}} \quad \text{or} \quad |①| \le \frac{R^2}{5}\right\}. \tag{174}$$

Moreover, we notice here that probability event $E_T$ implies that

$$\sum_{t=0}^T \sigma_t^2 \overset{(173)}{\le} 36\gamma^2 R^2 \sum_{t=0}^T \mathbb{E}_{\boldsymbol{\xi}^t}\left[\|\omega_t^u\|^2\right] \overset{(171), T\le K+1}{\le} \frac{648\gamma^2 R^2\sigma^2(K+1)}{m} \overset{(158)}{\le} \frac{R^4}{150\ln\frac{6(K+1)}{\beta}}. \tag{175}$$

**Upper bound for ②.** Probability event $E_T$ implies

$$\begin{aligned}
② &\le 2\gamma\sum_{t=0}^T \|\eta_l\| \cdot \|\omega_t^b\| \overset{(164),(169), T\le K+1}{\le} \frac{24\gamma\sigma^2 R(K+1)}{m\lambda} \\
&\overset{(157)}{=} \frac{1440\gamma^2\sigma^2(K+1)\ln\frac{6(K+1)}{\beta}}{m} \overset{(158)}{\le} \frac{R^2}{5}.
\end{aligned} \tag{176}$$

**Upper bound for ③.** Probability event $E_T$ implies

$$③ = 2\gamma^2 \sum_{t=0}^{T} \mathbb{E}_{\boldsymbol{\xi}^t} \left[ \|\omega_t^u\|^2 \right] \overset{(171),T\leq K+1}{\leq} \frac{36\gamma^2\sigma^2(K+1)}{m} \overset{(158)}{\leq} \frac{R^2}{5}. \tag{177}$$

**Upper bound for ④.** We have

$$2\gamma^2 \mathbb{E}_{\boldsymbol{\xi}^t} \left[ \|\omega_t^u\|^2 - \mathbb{E}_{\boldsymbol{\xi}^t} \left[ \|\omega_t^u\|^2 \right] \right] = 0.$$

Next, the summands in ④ are bounded with probability 1:

$$\begin{aligned}
2\gamma^2 \left| \|\omega_t^u\|^2 - \mathbb{E}_{\boldsymbol{\xi}^t} \left[ \|\omega_t^u\|^2 \right] \right| &\leq 2\gamma^2 \left( \|\omega_t^u\|^2 + \mathbb{E}_{\boldsymbol{\xi}^t} \left[ \|\omega_t^u\|^2 \right] \right) \overset{(168)}{\leq} 16\gamma^2\lambda^2 \\
&\overset{(157)}{\leq} \frac{R^2}{225 \ln \frac{6(K+1)}{\beta}} \leq \frac{R^2}{5 \ln \frac{6(K+1)}{\beta}} \overset{\text{def}}{=} c.
\end{aligned} \tag{178}$$

Moreover, these summands have bounded conditional variances $\widetilde{\sigma}_t^2 \overset{\text{def}}{=}$ $4\gamma^4 \mathbb{E}_{\boldsymbol{\xi}^t} \left[ \left( \|\omega_t^u\|^2 - \mathbb{E}_{\boldsymbol{\xi}^t} \left[ \|\omega_t^u\|^2 \right] \right)^2 \right]$:

$$\widetilde{\sigma}_t^2 \overset{(178)}{\leq} \frac{2\gamma^2 R^2}{225 \ln \frac{6(K+1)}{\beta}} \mathbb{E}_{\boldsymbol{\xi}^t} \left[ \left| \|\omega_t^u\|^2 - \mathbb{E}_{\boldsymbol{\xi}^t} \left[ \|\omega_t^u\|^2 \right] \right| \right] \leq \frac{4\gamma^2 R^2}{225 \ln \frac{6(K+1)}{\beta}} \mathbb{E}_{\boldsymbol{\xi}^t} \left[ \|\omega_t^u\|^2 \right]. \tag{179}$$

That is, sequence $\{\|\omega_t^u\|^2 - \mathbb{E}_{\boldsymbol{\xi}^t}[\|\omega_t^u\|^2]\}_{t\geq 0}$ is a bounded martingale difference sequence having bounded conditional variances $\{\widetilde{\sigma}_t^2\}_{t\geq 0}$. Applying Bernstein's inequality (Lemma B.1) with $X_t = \|\omega_t^u\|^2 - \mathbb{E}_{\boldsymbol{\xi}^t}[\|\omega_t^u\|^2]$, $c$ defined in (178), $b = \frac{R^2}{5}$, $G = \frac{R^4}{150 \ln \frac{6(K+1)}{\beta}}$, we get that

$$\mathbb{P} \left\{ |④| > \frac{R^2}{5} \text{ and } \sum_{t=0}^{T} \widetilde{\sigma}_t^2 \leq \frac{R^4}{150 \ln \frac{6(K+1)}{\beta}} \right\} \leq 2\exp\left( -\frac{b^2}{2G + 2cb/3} \right) = \frac{\beta}{3(K+1)}.$$

In other words, $\mathbb{P}\{E_④\} \geq 1 - \frac{\beta}{3(K+1)}$, where probability event $E_④$ is defined as

$$E_④ = \left\{ \text{either} \quad \sum_{t=0}^{T} \widetilde{\sigma}_t^2 > \frac{R^4}{150 \ln \frac{6(K+1)}{\beta}} \quad \text{or} \quad |④| \leq \frac{R^2}{5} \right\}. \tag{180}$$

Moreover, we notice here that probability event $E_T$ implies that

$$\begin{aligned}
\sum_{t=0}^{T} \widetilde{\sigma}_t^2 &\overset{(179)}{\leq} \frac{4\gamma^2 R^2}{225 \ln \frac{6(K+1)}{\beta}} \sum_{t=0}^{T} \mathbb{E}_{\boldsymbol{\xi}^t} \left[ \|\omega_t^u\|^2 \right] \overset{(171),T\leq K+1}{\leq} \frac{8\gamma^2 R^2 \sigma^2 (K+1)}{25m \ln \frac{6(K+1)}{\beta}} \\
&\overset{(158)}{\leq} \frac{R^4}{150 \ln \frac{6(K+1)}{\beta}}.
\end{aligned} \tag{181}$$

**Upper bound for ⑤.** Probability event $E_T$ implies

$$\begin{aligned}
⑤ &= 2\gamma^2 \sum_{t=0}^{T} \|\omega_t^b\|^2 \overset{(169),T\leq K+1}{\leq} \frac{32\gamma^2\sigma^4(K+1)}{m^2\lambda^2} \overset{(157)}{=} \frac{115200\gamma^4\sigma^4(K+1)\ln^2 \frac{6(K+1)}{\beta}}{m^2 R^2} \\
&\overset{(158)}{\leq} \frac{R^2}{5}.
\end{aligned} \tag{182}$$

**Upper bound for $\gamma \left\| \sum_{t=0}^{T} \omega_t \right\|$.** To handle this term, we introduce new notation:

$$\zeta_l = \begin{cases} \gamma \sum_{r=0}^{l-1} \omega_r, & \text{if } \left\| \gamma \sum_{r=0}^{l-1} \omega_r \right\| \leq R, \\ 0, & \text{otherwise} \end{cases}$$

for $l = 1, 2, \ldots, T-1$. By definition, we have

$$\|\zeta_l\| \leq R. \tag{183}$$

Therefore, in view of (160), probability event $E_T$ implies

$$
\begin{aligned}
\gamma \left\| \sum_{l=0}^{T} \omega_l \right\| &= \sqrt{\gamma^2 \left\| \sum_{l=0}^{T} \omega_l \right\|^2} \\
&= \sqrt{\gamma^2 \sum_{l=0}^{T} \|\omega_l\|^2 + 2\gamma \sum_{l=0}^{T} \left\langle \gamma \sum_{r=0}^{l-1} \omega_r, \omega_l \right\rangle} \\
&= \sqrt{\gamma^2 \sum_{l=0}^{T} \|\omega_l\|^2 + 2\gamma \sum_{l=0}^{T} \langle \zeta_l, \omega_l \rangle} \\
&\overset{(166)}{\leq} \sqrt{③ + ④ + ⑤ + \underbrace{2\gamma \sum_{l=0}^{T} \langle \zeta_l, \omega_l^u \rangle}_{⑥} + \underbrace{2\gamma \sum_{l=0}^{T} \langle \zeta_l, \omega_l^b \rangle}_{⑦}}.
\end{aligned} \tag{184}
$$

Following similar steps as before, we bound ⑥ and ⑦.

**Upper bound for ⑥.** Since $\mathbb{E}_{\boldsymbol{\xi}^t}[\omega_t^u] = 0$, we have

$$\mathbb{E}_{\boldsymbol{\xi}^t}\left[2\gamma \langle \zeta_t, \omega_t^u \rangle \right] = 0.$$

Next, the summands in ⑥ are bounded with probability 1:

$$|2\gamma \langle \zeta_t, \omega_t^u \rangle| \leq 2\gamma \|\zeta_t\| \cdot \|\omega_t^u\| \overset{(183),(168)}{\leq} 4\gamma R\lambda \overset{(157)}{\leq} \frac{R^2}{5 \ln \frac{6(K+1)}{\beta}} \overset{\text{def}}{=} c. \tag{185}$$

Moreover, these summands have bounded conditional variances $\widehat{\sigma}_t^2 \overset{\text{def}}{=} \mathbb{E}_{\boldsymbol{\xi}^t}\left[4\gamma^2 \langle \zeta_t, \omega_t^u \rangle^2\right]$:

$$\widehat{\sigma}_t^2 \leq \mathbb{E}_{\boldsymbol{\xi}^t}\left[4\gamma^2 \|\zeta_t\|^2 \cdot \|\omega_t^u\|^2\right] \overset{(164)}{\leq} 4\gamma^2 R^2 \mathbb{E}_{\boldsymbol{\xi}^t}\left[\|\omega_t^u\|^2\right]. \tag{186}$$

That is, sequence $\{2\gamma \langle \zeta_t, \omega_t^u \rangle\}_{t \geq 0}$ is a bounded martingale difference sequence having bounded conditional variances $\{\widehat{\sigma}_t^2\}_{t \geq 0}$. Applying Bernstein's inequality (Lemma B.1) with $X_t = 2\gamma \langle \zeta_t, \omega_t^u \rangle$, $c$ defined in (172), $b = \frac{R^2}{5}$, $G = \frac{R^4}{150 \ln \frac{6(K+1)}{\beta}}$, we get that

$$\mathbb{P}\left\{|⑥| > \frac{R^2}{5} \text{ and } \sum_{t=0}^{T} \widehat{\sigma}_t^2 \leq \frac{R^4}{150 \ln \frac{6(K+1)}{\beta}}\right\} \leq 2\exp\left(-\frac{b^2}{2G + 2cb/3}\right) = \frac{\beta}{3(K+1)}.$$

In other words, $\mathbb{P}\{E_⑥\} \geq 1 - \frac{\beta}{3(K+1)}$, where probability event $E_⑥$ is defined as

$$E_⑥ = \left\{\text{either} \quad \sum_{t=0}^{T} \widehat{\sigma}_t^2 > \frac{R^4}{150 \ln \frac{6(K+1)}{\beta}} \quad \text{or} \quad |⑥| \leq \frac{R^2}{5}\right\}. \tag{187}$$

Moreover, we notice here that probability event $E_T$ implies that

$$\sum_{t=0}^{T} \widehat{\sigma}_t^2 \overset{(186)}{\leq} 4\gamma^2 R^2 \sum_{t=0}^{T} \mathbb{E}_{\boldsymbol{\xi}^t}\left[\|\omega_t^u\|^2\right] \overset{(171),T \leq K+1}{\leq} \frac{72\gamma^2 R^2 \sigma^2 (K+1)}{m} \overset{(158)}{\leq} \frac{R^4}{150 \ln \frac{6(K+1)}{\beta}}. \tag{188}$$

**Upper bound for ⑦.** Probability event $E_T$ implies

$$
\begin{aligned}
⑦ &\leq 2\gamma \sum_{t=0}^{T} \|\zeta_t\| \cdot \|\omega_t^b\| \overset{(183),(169),T \leq K+1}{\leq} \frac{8\gamma \sigma^2 R(K+1)}{m\lambda} \\
&\overset{(157)}{=} \frac{480\gamma^2 \sigma^2 (K+1) \ln \frac{6(K+1)}{\beta}}{m} \overset{(158)}{\leq} \frac{R^2}{5}.
\end{aligned} \tag{189}
$$

**Final derivation.** Putting all bounds together, we get that $E_T$ implies

$$R_{T+1}^2 \overset{(166)}{\leq} R^2 + ① + ② + ③ + ④ + ⑤,$$

$$2\gamma(T+1)\mathrm{Gap}_R(x_{\mathrm{avg}}^T) \overset{(167)}{\leq} 5R^2 + 2\gamma R \left\| \sum_{t=0}^T \omega_t \right\| + 2 \cdot (① + ② + ③ + ④ + ⑤),$$

$$\gamma \left\| \sum_{l=0}^T \omega_l \right\| \overset{(184)}{\leq} \sqrt{③ + ④ + ⑤ + ⑥ + ⑦},$$

$$② \overset{(176)}{\leq} \frac{R^2}{5}, \quad ③ \overset{(177)}{\leq} \frac{R^2}{5}, \quad ⑤ \overset{(182)}{\leq} \frac{R^2}{5}, \quad ⑦ \overset{(189)}{\leq} \frac{R^2}{5},$$

$$\sum_{t=0}^T \sigma_t^2 \overset{(175)}{\leq} \frac{R^4}{150 \ln \frac{6(K+1)}{\beta}}, \quad \sum_{t=0}^T \widetilde{\sigma}_t^2 \overset{(181)}{\leq} \frac{R^4}{150 \ln \frac{6(K+1)}{\beta}}, \quad \sum_{t=0}^T \widehat{\sigma}_t^2 \overset{(188)}{\leq} \frac{R^4}{150 \ln \frac{6(K+1)}{\beta}}.$$

Moreover, in view of (174), (180), (189), and our induction assumption, we have

$$\mathbb{P}\{E_T\} \geq 1 - \frac{T\beta}{K+1},$$

$$\mathbb{P}\{E_①\} \geq 1 - \frac{\beta}{3(K+1)}, \quad \mathbb{P}\{E_④\} \geq 1 - \frac{\beta}{3(K+1)}, \quad \mathbb{P}\{E_⑥\} \geq 1 - \frac{\beta}{3(K+1)},$$

where probability events $E_①$, $E_④$, and $E_⑥$ are defined as

$$E_① = \left\{ \text{either} \quad \sum_{t=0}^T \sigma_t^2 > \frac{R^4}{150 \ln \frac{6(K+1)}{\beta}} \quad \text{or} \quad |①| \leq \frac{R^2}{5} \right\},$$

$$E_④ = \left\{ \text{either} \quad \sum_{t=0}^T \widetilde{\sigma}_t^2 > \frac{R^4}{150 \ln \frac{6(K+1)}{\beta}} \quad \text{or} \quad |④| \leq \frac{R^2}{5} \right\},$$

$$E_⑥ = \left\{ \text{either} \quad \sum_{t=0}^T \widehat{\sigma}_t^2 > \frac{R^4}{150 \ln \frac{6(K+1)}{\beta}} \quad \text{or} \quad |⑥| \leq \frac{R^2}{5} \right\}.$$

Putting all of these inequalities together, we obtain that probability event $E_T \cap E_① \cap E_④ \cap E_⑥$ implies

$$\begin{aligned} R_{T+1}^2 &\leq R^2 + ① + ② + ③ + ④ + ⑤ \leq 2R^2, \\ \gamma \left\| \sum_{l=0}^T \omega_l \right\| &\leq \sqrt{③ + ④ + ⑤ + ⑥ + ⑦} \leq R, \\ 2\gamma(T+1)\mathrm{Gap}_R(x_{\mathrm{avg}}^T) &\leq 5R^2 + 2\gamma R \left\| \sum_{t=0}^T \omega_t \right\| + 2 \cdot (① + ② + ③ + ④ + ⑤) \\ &\leq 9R^2. \end{aligned}$$

Moreover, union bound for the probability events implies

$$\mathbb{P}\{E_{T+1}\} \geq \mathbb{P}\{E_T \cap E_① \cap E_④ \cap E_⑥\} = 1 - \mathbb{P}\{\overline{E}_T \cup \overline{E}_① \cup \overline{E}_④ \cup \overline{E}_⑥\} \geq 1 - \frac{T\beta}{K+1}.$$

This is exactly what we wanted to prove (see the paragraph after inequality (160)). In particular, $E_K$ implies

$$\mathrm{Gap}_R(x_{\mathrm{avg}}^K) \leq \frac{9R^2}{2\gamma(K+1)},$$

which finishes the proof. $\qquad\square$

**Corollary D.1.** *Let the assumptions of Theorem D.1 hold. Then, the following statements hold.*

1. **Large stepsize/large batch.** *The choice of stepsize and batchsize*

$$\gamma = \frac{1}{170\ell \ln \frac{6(K+1)}{\beta}}, \quad m = \max \left\{ 1, \frac{972(K+1)\sigma^2}{289\ell^2 R^2 \ln \frac{6(K+1)}{\beta}} \right\} \tag{190}$$

*satisfies conditions* (156) *and* (158). *With such choice of* $\gamma, m$, *and the choice of* $\lambda$ *as in* (157), *the iterates produced by* clipped-SGDA *after* $K$ *iterations with probability at least* $1 - \beta$ *satisfy*

$$\mathsf{Gap}(x_{avg}^K) \le \frac{765\ell R^2 \ln \frac{6(K+1)}{\beta}}{K+1}. \tag{191}$$

*In particular, to guarantee* $\mathsf{Gap}(x_{avg}^K) \le \varepsilon$ *with probability at least* $1 - \beta$ *for some* $\varepsilon > 0$ clipped-SGDA *requires,*

$$\mathcal{O}\left( \frac{\ell R^2}{\varepsilon} \ln \left( \frac{\ell R^2}{\varepsilon \beta} \right) \right) \quad iterations, \tag{192}$$

$$\mathcal{O}\left( \max \left\{ \frac{\ell R^2}{\varepsilon}, \frac{\sigma^2 R^2}{\varepsilon^2} \right\} \ln \left( \frac{\ell R^2}{\varepsilon \beta} \right) \right) \quad oracle\ calls. \tag{193}$$

2. **Small stepsize/small batch.** *The choice of stepsize and batchsize*

$$\gamma = \min \left\{ \frac{1}{170\ell \ln \frac{6(K+1)}{\beta}}, \frac{R}{180\sigma \sqrt{3(K+1) \ln \frac{6(K+1)}{\beta}}} \right\}, \quad m = 1 \tag{194}$$

*satisfies conditions* (156) *and* (158). *With such choice of* $\gamma, m$, *and the choice of* $\lambda$ *as in* (157), *the iterates produced by* clipped-SGDA *after* $K$ *iterations with probability at least* $1 - \beta$ *satisfy*

$$\mathsf{Gap}(x_{avg}^K) \le \max \left\{ \frac{765\ell R^2 \ln \frac{6(K+1)}{\beta}}{K+1}, \frac{810\sigma R \sqrt{3 \ln \frac{6(K+1)}{\beta}}}{\sqrt{K+1}} \right\}. \tag{195}$$

*In particular, to guarantee* $\mathsf{Gap}(x_{avg}^K) \le \varepsilon$ *with probability at least* $1 - \beta$ *for some* $\varepsilon > 0$, clipped-SGDA *requires*

$$\mathcal{O}\left( \max \left\{ \frac{\ell R^2}{\varepsilon} \ln \left( \frac{\ell R^2}{\varepsilon \beta} \right), \frac{\sigma^2 R^2}{\varepsilon^2} \ln \left( \frac{\sigma^2 R^2}{\varepsilon^2 \beta} \right) \right\} \right) \quad iterations/oracle\ calls. \tag{196}$$

*Proof.*      1. **Large stepsize/large batch.** First of all, we verify that the choice of $\gamma$ and $m$ from (190) satisfies conditions (156) and (158): (156) trivially holds and (158) holds since

$$m = \max \left\{ 1, \frac{972(K+1)\sigma^2}{289\ell^2 R^2 \ln \frac{6(K+1)}{\beta}} \right\} = \max \left\{ 1, \frac{97200(K+1)\gamma^2\sigma^2 \ln \frac{6(K+1)}{\beta}}{R^2} \right\}.$$

Therefore, applying Theorem D.1, we derive that with probability at least $1 - \beta$

$$\mathsf{Gap}(x_{avg}^K) \le \frac{9R^2}{2\gamma(K+1)} \overset{(190)}{\le} \frac{765\ell R^2 \ln \frac{6(K+1)}{\beta}}{K+1}.$$

To guarantee $\mathsf{Gap}(x_{avg}^K) \le \varepsilon$, we choose $K$ in such a way that the right-hand side of the above inequality is smaller than $\varepsilon$ that gives

$$K = \mathcal{O}\left( \frac{\ell R^2}{\varepsilon} \ln \left( \frac{\ell R^2}{\varepsilon \beta} \right) \right).$$

The total number of oracle calls equals

$$m(K+1) \overset{(190)}{=} \max \left\{ K+1, \frac{972(K+1)^2\sigma^2}{289\ell^2 R^2 \ln \frac{6(K+1)}{\beta}} \right\}$$

$$= \mathcal{O}\left( \max \left\{ \frac{\ell R^2}{\varepsilon}, \frac{\sigma^2 R^2}{\varepsilon^2} \right\} \ln \left( \frac{\ell R^2}{\varepsilon \beta} \right) \right).$$

2. **Small stepsize/small batch.** First of all, we verify that the choice of $\gamma$ and $m$ from (194) satisfies conditions (156) and (158):

$$\gamma = \min\left\{\frac{1}{170\ell\ln\frac{6(K+1)}{\beta}}, \frac{R}{180\sigma\sqrt{3(K+1)\ln\frac{6(K+1)}{\beta}}}\right\} \leq \frac{1}{170\ell\ln\frac{6(K+1)}{\beta}},$$

$$m = 1 \stackrel{(194)}{\geq} \frac{97200(K+1)\gamma^2\sigma^2\ln\frac{6(K+1)}{\beta}}{R^2}.$$

Therefore, applying Theorem D.1, we derive that with probability at least $1 - \beta$

$$\begin{aligned}
\mathtt{Gap}(x_{\mathrm{avg}}^K) &\leq \frac{9R^2}{2\gamma(K+1)}\\
&\stackrel{(194)}{=} \max\left\{\frac{765\ell R^2\ln\frac{6(K+1)}{\beta}}{K+1}, \frac{810\sigma R\sqrt{3\ln\frac{6(K+1)}{\beta}}}{\sqrt{K+1}}\right\}.
\end{aligned}$$

To guarantee $\mathtt{Gap}(x_{\mathrm{avg}}^K) \leq \varepsilon$, we choose $K$ in such a way that the right-hand side of the above inequality is smaller than $\varepsilon$ that gives

$$K = \mathcal{O}\left(\max\left\{\frac{\ell R^2}{\varepsilon}\ln\left(\frac{\ell R^2}{\varepsilon\beta}\right), \frac{\sigma^2 R^2}{\varepsilon^2}\ln\left(\frac{\sigma^2 R^2}{\varepsilon^2\beta}\right)\right\}\right).$$

The total number of oracle calls equals $K + 1$.

$\square$

## D.2 Star-Cocoercive Case

**Theorem D.2.** *Let Assumptions 1.1, 1.6, hold for $Q = B_{2R}(x^*)$, where $R \geq R_0 \stackrel{def}{=} \|x^0 - x^*\|$, and*

$$\gamma \leq \frac{1}{170\ell\ln\frac{4(K+1)}{\beta}}, \tag{197}$$

$$\lambda = \frac{R}{60\gamma\ln\frac{4(K+1)}{\beta}}, \tag{198}$$

$$m \geq \max\left\{1, \frac{97200(K+1)\gamma^2\sigma^2\ln\frac{4(K+1)}{\beta}}{R^2}\right\}, \tag{199}$$

*for some $K \geq 0$ and $\beta \in (0, 1]$ such that $\ln\frac{4(K+1)}{\beta} \geq 1$. Then, after $K$ iterations the iterates produced by* clipped-SGDA *with probability at least $1 - \beta$ satisfy*

$$\frac{1}{K+1}\sum_{k=0}^{K}\|F(x^k)\|^2 \leq \frac{2\ell R^2}{\gamma(K+1)}. \tag{200}$$

*Proof.* We introduce new notation: $R_k = \|x^k - x^*\|$ for all $k \geq 0$. The proof is based on deriving via induction that $R_k^2 \leq CR^2$ for some numerical constant $C > 0$. In particular, for each $k = 0, \ldots, K+1$ we define probability event $E_k$ as follows: inequalities

$$\|x^t - x^*\|^2 \leq 2R^2, \tag{201}$$

hold for $t = 0, 1, \ldots, k$ simultaneously. Our goal is to prove that $\mathbb{P}\{E_k\} \geq 1 - {k\beta}/{(K+1)}$ for all $k = 0, 1, \ldots, K+1$. We notice that inequalities (161) and (166) are derived without assuming monotonicity of $F$. Therefore, following exactly the same step as in the proof of Theorem D.1 (up to

the replacement of $\ln \frac{6(K+1)}{\beta}$ by $\ln \frac{4(K+1)}{\beta}$), we get that

$$R_{T+1}^2 \overset{(166)}{\leq} R^2 + ① + ② + ③ + ④ + ⑤,$$

$$② \overset{(176)}{\leq} \frac{R^2}{5}, \quad ③ \overset{(177)}{\leq} \frac{R^2}{5}, \quad ⑤ \overset{(182)}{\leq} \frac{R^2}{5},$$

$$\sum_{t=0}^{T} \sigma_t^2 \overset{(175)}{\leq} \frac{R^4}{150 \ln \frac{4(K+1)}{\beta}}, \quad \sum_{t=0}^{T} \widetilde{\sigma}_t^2 \overset{(181)}{\leq} \frac{R^4}{150 \ln \frac{4(K+1)}{\beta}}.$$

Moreover, in view of (174), (180), and our induction assumption, we have

$$\mathbb{P}\{E_T\} \geq 1 - \frac{T\beta}{K+1},$$

$$\mathbb{P}\{E_①\} \geq 1 - \frac{\beta}{2(K+1)}, \quad \mathbb{P}\{E_④\} \geq 1 - \frac{\beta}{2(K+1)},$$

where probability events $E_①$, and $E_④$ are defined as

$$E_① = \left\{ \text{either} \quad \sum_{t=0}^{T} \sigma_t^2 > \frac{R^4}{150 \ln \frac{4(K+1)}{\beta}} \quad \text{or} \quad |①| \leq \frac{R^2}{5} \right\},$$

$$E_④ = \left\{ \text{either} \quad \sum_{t=0}^{T} \widetilde{\sigma}_t^2 > \frac{R^4}{150 \ln \frac{4(K+1)}{\beta}} \quad \text{or} \quad |④| \leq \frac{R^2}{5} \right\}.$$

Putting all of these inequalities together, we obtain that probability event $E_{T-1} \cap E_① \cap E_④$ implies

$$R_{T+1}^2 \leq R^2 + ① + ② + ③ + ④ + ⑤ \leq 2R^2.$$

Moreover, union bound for the probability events implies

$$\mathbb{P}\{E_{T+1}\} \geq \mathbb{P}\{E_T \cap E_① \cap E_④\} = 1 - \mathbb{P}\{\overline{E}_T \cup \overline{E}_① \cup \overline{E}_④\} \geq 1 - \frac{T\beta}{K+1}. \tag{202}$$

This is exactly what we wanted to prove (see the paragraph after inequality (201)). In particular, $E_K$ implies

$$\frac{1}{K+1} \sum_{k=0}^{K} \|F(x^k)\|^2 \overset{(161)}{\leq} \frac{\ell(R^2 - R_{K+1}^2)}{\gamma(K+1)} + \frac{\ell(① + ② + ③ + ④ + ⑤)}{\gamma(K+1)}$$

$$\leq \frac{2\ell R^2}{\gamma(K+1)}.$$

This finishes the proof.

$\square$

**Corollary D.2.** *Let the assumptions of Theorem D.2 hold. Then, the following statements hold.*

1. ***Large stepsize/large batch.*** *The choice of stepsize and batchsize*

$$\gamma = \frac{1}{170\ell \ln \frac{4(K+1)}{\beta}}, \quad m = \max\left\{1, \frac{972(K+1)\sigma^2}{289\ell^2 R^2 \ln \frac{4(K+1)}{\beta}}\right\} \tag{203}$$

*satisfies conditions (197) and (199). With such choice of $\gamma, m$, and the choice of $\lambda$ as in (198), the iterates produced by* clipped-SGDA *after $K$ iterations with probability at least $1 - \beta$ satisfy*

$$\frac{1}{K+1} \sum_{k=0}^{K} \|F(x^k)\|^2 \leq \frac{340\ell^2 R^2 \ln \frac{4(K+1)}{\beta}}{K+1}. \tag{204}$$

*In particular, to guarantee $\frac{1}{K+1} \sum_{k=0}^{K} \|F(x^k)\|^2 \leq \varepsilon$ with probability at least $1 - \beta$ for some $\varepsilon > 0$* clipped-SGDA *requires,*

$$\mathcal{O}\left(\frac{\ell^2 R^2}{\varepsilon} \ln\left(\frac{\ell^2 R^2}{\varepsilon \beta}\right)\right) \text{ iterations,} \tag{205}$$

$$\mathcal{O}\left(\max\left\{\frac{\ell^2 R^2}{\varepsilon}, \frac{\ell^2 \sigma^2 R^2}{\varepsilon^2}\right\} \ln\left(\frac{\ell^2 R^2}{\varepsilon \beta}\right)\right) \text{ oracle calls.} \tag{206}$$

2. ***Small stepsize/small batch.*** *The choice of stepsize and batchsize*

$$\gamma = \min\left\{\frac{1}{170\ell \ln\frac{4(K+1)}{\beta}}, \frac{R}{180\sigma\sqrt{3(K+1)\ln\frac{4(K+1)}{\beta}}}\right\}, \quad m = 1 \qquad (207)$$

*satisfies conditions* (197) *and* (199). *With such choice of* $\gamma, m$, *and the choice of* $\lambda$ *as in* (198), *the iterates produced by* clipped-SGDA *after* $K$ *iterations with probability at least* $1 - \beta$ *satisfy*

$$\frac{1}{K+1}\sum_{k=0}^{K}\|F(x^k)\|^2 \le \max\left\{\frac{340\ell^2 R^2 \ln\frac{4(K+1)}{\beta}}{K+1}, \frac{360\ell\sigma R\sqrt{3\ln\frac{4(K+1)}{\beta}}}{\sqrt{K+1}}\right\}. \quad (208)$$

*In particular, to guarantee* $\frac{1}{K+1}\sum_{k=0}^{K}\|F(x^k)\|^2 \le \varepsilon$ *with probability at least* $1 - \beta$ *for some* $\varepsilon > 0$, clipped-SGDA *requires*

$$\mathcal{O}\left(\max\left\{\frac{\ell^2 R^2}{\varepsilon}\ln\left(\frac{\ell^2 R^2}{\varepsilon\beta}\right), \frac{\ell^2\sigma^2 R^2}{\varepsilon^2}\ln\left(\frac{\ell^2\sigma^2 R^2}{\varepsilon^2\beta}\right)\right\}\right) \quad \text{iterations/oracle calls.} \quad (209)$$

*Proof.* 1. **Large stepsize/large batch.** First of all, we verify that the choice of $\gamma$ and $m$ from (203) satisfies conditions (197) and (199): (197) trivially holds and (199) holds since

$$m = \max\left\{1, \frac{972(K+1)\sigma^2}{289\ell^2 R^2 \ln\frac{4(K+1)}{\beta}}\right\} = \max\left\{1, \frac{97200(K+1)\gamma^2\sigma^2 \ln\frac{4(K+1)}{\beta}}{R^2}\right\}.$$

Therefore, applying Theorem D.2, we derive that with probability at least $1 - \beta$

$$\frac{1}{K+1}\sum_{k=0}^{K}\|F(x^k)\|^2 \le \frac{2\ell R^2}{\gamma(K+1)} \overset{(203)}{\le} \frac{340\ell^2 R^2 \ln\frac{4(K+1)}{\beta}}{K+1}.$$

To guarantee $\frac{1}{K+1}\sum_{k=0}^{K}\|F(x^k)\|^2 \le \varepsilon$, we choose $K$ in such a way that the right-hand side of the above inequality is smaller than $\varepsilon$ that gives

$$K = \mathcal{O}\left(\frac{\ell^2 R^2}{\varepsilon}\ln\left(\frac{\ell^2 R^2}{\varepsilon\beta}\right)\right).$$

The total number of oracle calls equals

$$m(K+1) \overset{(203)}{=} \max\left\{K+1, \frac{972(K+1)^2\sigma^2}{289\ell^2 R^2 \ln\frac{4(K+1)}{\beta}}\right\}$$

$$= \mathcal{O}\left(\max\left\{\frac{\ell^2 R^2}{\varepsilon}, \frac{\ell^2\sigma^2 R^2}{\varepsilon^2}\right\}\ln\left(\frac{\ell^2 R^2}{\varepsilon\beta}\right)\right).$$

2. **Small stepsize/small batch.** First of all, we verify that the choice of $\gamma$ and $m$ from (207) satisfies conditions (197) and (199):

$$\gamma = \min\left\{\frac{1}{170\ell \ln\frac{4(K+1)}{\beta}}, \frac{R}{180\sigma\sqrt{3(K+1)\ln\frac{4(K+1)}{\beta}}}\right\} \le \frac{1}{170\ell \ln\frac{4(K+1)}{\beta}},$$

$$m = 1 \overset{(207)}{\ge} \frac{97200(K+1)\gamma^2\sigma^2 \ln\frac{4(K+1)}{\beta}}{R^2}.$$

Therefore, applying Theorem D.2, we derive that with probability at least $1 - \beta$

$$\frac{1}{K+1}\sum_{k=0}^{K}\|F(x^k)\|^2 \le \frac{2\ell R^2}{\gamma(K+1)}$$

$$\overset{(207)}{=} \max\left\{\frac{340\ell^2 R^2 \ln\frac{4(K+1)}{\beta}}{K+1}, \frac{360\ell\sigma R\sqrt{3\ln\frac{4(K+1)}{\beta}}}{\sqrt{K+1}}\right\}.$$

To guarantee $\frac{1}{K+1} \sum_{k=0}^{K} \|F(x^k)\|^2 \le \varepsilon$, we choose $K$ in such a way that the right-hand side of the above inequality is smaller than $\varepsilon$ that gives

$$K = \mathcal{O}\left( \max\left\{ \frac{\ell^2 R^2}{\varepsilon} \ln\left(\frac{\ell^2 R^2}{\varepsilon \beta}\right), \frac{\ell^2 \sigma^2 R^2}{\varepsilon^2} \ln\left(\frac{\ell^2 \sigma^2 R^2}{\varepsilon^2 \beta}\right) \right\} \right).$$

The total number of oracle calls equals $K + 1$.

$\square$

### D.3 Quasi-Strongly Monotone Star-Cocoercive Case

**Lemma D.3.** *Let Assumptions 1.5, 1.6 hold for $Q = B_{2R}(x^*)$, where $R \ge R_0 \stackrel{def}{=} \|x^0 - x^*\|$, and $0 < \gamma \le 1/\ell$. If $x^k$ lies in $B_{2R}(x^*)$ for all $k = 0, 1, \dots, K$ for some $K \ge 0$, then the iterates produced by* clipped-SGDA *satisfy*

$$\|x^{K+1} - x^*\|^2 \le (1 - \gamma\mu)^{K+1}\|x^0 - x^*\|^2 + 2\gamma \sum_{k=0}^{K}(1 - \gamma\mu)^{K-k}\langle x^k - x^* - \gamma F(x^k), \omega_k \rangle$$

$$+ \gamma^2 \sum_{k=0}^{K}(1 - \gamma\mu)^{K-k}\|\omega_k\|^2, \tag{210}$$

*where $\omega_k$ is defined in (154).*

*Proof.* Using the update rule of clipped-SGDA, we obtain

$$
\begin{aligned}
\|x^{k+1} - x^*\|^2 &= \|x^k - x^*\|^2 - 2\gamma\langle x^k - x^*, \widetilde{F}_{\boldsymbol{\xi}^k}(x^k) \rangle + \gamma^2 \|\widetilde{F}_{\boldsymbol{\xi}^k}(x^k)\|^2 \\
&= \|x^k - x^*\|^2 - 2\gamma\langle x^k - x^*, F(x^k) \rangle + 2\gamma\langle x^k - x^*, \omega_k \rangle \\
&\quad + \gamma^2\|F(x^k)\|^2 - 2\gamma^2\langle F(x^k), \omega_k \rangle + \gamma^2\|\omega_k\|^2 \\
&= \|x^k - x^*\|^2 + 2\gamma\langle x^k - x^* - \gamma F(x^k), \omega_k \rangle \\
&\quad - 2\gamma\langle x^k - x^*, F(x^k) \rangle + \gamma^2\|F(x^k)\|^2 + \gamma^2\|\omega_k\|^2 \\
&\overset{\text{(SC)}}{\le} \|x^k - x^*\|^2 + 2\gamma\langle x^k - x^* - \gamma F(x^k), \omega_k \rangle \\
&\quad - 2\gamma\langle x^k - x^*, F(x^k) \rangle + \gamma^2\ell\langle x^k - x^*, F(x^k) \rangle + \gamma^2\|\omega_k\|^2 \\
&= \|x^k - x^*\|^2 + 2\gamma\langle x^k - x^* - \gamma F(x^k), \omega_k \rangle \\
&\quad - 2\gamma\left(1 - \frac{\gamma\ell}{2}\right)\langle x^k - x^*, F(x^k) \rangle + \gamma^2\|\omega_k\|^2 \\
&\overset{\text{(QSM)},\gamma\le\frac{1}{\ell}}{\le} \|x^k - x^*\|^2 + 2\gamma\langle x^k - x^* - \gamma F(x^k), \omega_k \rangle \\
&\quad - 2\gamma\mu\left(1 - \frac{\gamma\ell}{2}\right)\|x^k - x^*\|^2 + \gamma^2\|\omega_k\|^2 \\
&\overset{\gamma\le\frac{1}{\ell}}{\le} (1 - \gamma\mu)\|x^k - x^*\|^2 + 2\gamma\langle x^k - x^* - \gamma F(x^k), \omega_k \rangle + \gamma^2\|\omega_k\|^2.
\end{aligned}
$$

Unrolling the recurrence, we obtain (210). $\square$

**Theorem D.3.** *Let Assumptions 1.1, 1.5, 1.6 hold for $Q = B_{2R}(x^*) = \{x \in \mathbb{R}^d \mid \|x - x^*\| \le 2R\}$, where $R \ge R_0 \stackrel{def}{=} \|x^0 - x^*\|$, and*

$$0 < \gamma \le \frac{1}{400\ell \ln \frac{4(K+1)}{\beta}}, \tag{211}$$

$$\lambda_k = \frac{\exp(-\gamma\mu(1 + k/2))R}{120\gamma \ln \frac{4(K+1)}{\beta}}, \tag{212}$$

$$m_k \ge \max\left\{ 1, \frac{27000\gamma^2(K+1)\sigma^2 \ln \frac{4(K+1)}{\beta}}{\exp(-\gamma\mu k)R^2} \right\}, \tag{213}$$

*for some $K \geq 0$ and $\beta \in (0, 1]$ such that $\ln \frac{4(K+1)}{\beta} \geq 1$. Then, after $K$ iterations the iterates produced by* clipped-SGDA *with probability at least $1 - \beta$ satisfy*

$$\|x^{K+1} - x^*\|^2 \leq 2 \exp(-\gamma\mu(K+1))R^2. \tag{214}$$

*Proof.* As in the proof of Theorem D.1, we use the following notation: $R_k = \|x^k - x^*\|^2$, $k \geq 0$. We will derive (214) by induction. In particular, for each $k = 0, \ldots, K+1$ we define probability event $E_k$ as follows: inequalities

$$R_t^2 \leq 2 \exp(-\gamma\mu t)R^2 \tag{215}$$

hold for $t = 0, 1, \ldots, k$ simultaneously. Our goal is to prove that $\mathbb{P}\{E_k\} \geq 1 - {}^{k\beta}/(K+1)$ for all $k = 0, 1, \ldots, K+1$. We use the induction to show this statement. For $k = 0$ the statement is trivial since $R_0^2 \leq 2R^2$ by definition. Next, assume that the statement holds for $k = T \leq K$, i.e., we have $\mathbb{P}\{E_T\} \geq 1 - {}^{T\beta}/(K+1)$. We need to prove that $\mathbb{P}\{E_{T+1}\} \geq 1 - {}^{(T+1)\beta}/(K+1)$. First of all, since $R_t^2 \leq 2 \exp(-\gamma\mu t)R^2 \leq 2R^2$, we have $x^t \in B_{2R}(x^*)$. Operator $F$ is $\ell$-star-cocoercive on $B_{2R}(x^*)$. Therefore, probability event $E_T$ implies

$$\|F(x^t)\| \overset{\text{(SC)}}{\leq} \ell\|x^t - x^*\| \overset{\text{(215)}}{\leq} \sqrt{2}\ell \exp(-\gamma\mu t/2)R \overset{\text{(211),(212)}}{\leq} \frac{\lambda_t}{2}. \tag{216}$$

and

$$\|\omega_t\|^2 \overset{\text{(5)}}{\leq} 2\|\widetilde{F}_{\boldsymbol{\xi}}(x^t)\|^2 + 2\|F(x^t)\|^2 \overset{\text{(216)}}{\leq} \frac{5}{2}\lambda_t^2 \overset{\text{(212)}}{\leq} \frac{\exp(-\gamma\mu t)R^2}{4\gamma^2} \tag{217}$$

for all $t = 0, 1, \ldots, T$.

Applying Lemma D.3 and $(1 - \gamma\mu)^T \leq \exp(-\gamma\mu T)$, we get that probability event $E_T$ implies

$$R_T^2 \leq \exp(-\gamma\mu T)R^2 + 2\gamma \sum_{t=0}^{T}(1 - \gamma\mu)^{T-t}\langle x^t - x^* - \gamma F(x^t), \omega_t\rangle$$

$$+\gamma^2 \sum_{t=0}^{T}(1 - \gamma\mu)^{T-t}\|\omega_t\|^2.$$

To estimate the sums in the right-hand side, we introduce new vectors:

$$\eta_t = \begin{cases} x^t - x^* - \gamma F(x^t), & \text{if } \|x^t - x^* - \gamma F(x^t)\| \leq \sqrt{2}(1 + \gamma\ell)\exp(-\gamma\mu t/2)R, \\ 0, & \text{otherwise,} \end{cases} \tag{218}$$

for $t = 0, 1, \ldots, T$. First of all, we point out that vector $\eta_t$ is bounded with probability 1, i.e., with probability 1

$$\|\eta_t\| \leq \sqrt{2}(1 + \gamma\ell)\exp(-\gamma\mu t/2)R \tag{219}$$

for all $t = 0, 1, \ldots, T$. Next, we notice that $E_T$ implies $\|F(x^t)\| \leq \sqrt{2}\ell \exp(-\gamma\mu t/2)R$ (due to (216)) for $t = 0, 1, \ldots, T$, i.e., probability event $E_T$ implies $\eta_t = x^t - x^* - \gamma F(x^t)$ for all $t = 0, 1, \ldots, T$. Therefore, $E_T$ implies

$$R_T^2 \leq \exp(-\gamma\mu T)R^2 + 2\gamma \sum_{t=0}^{T}(1 - \gamma\mu)^{T-t}\langle \eta_t, \omega_t\rangle$$

$$+\gamma^2 \sum_{t=0}^{T}(1 - \gamma\mu)^{T-t}\|\omega_t\|^2.$$

As in the monotone case, to continue the derivation, we introduce vectors $\omega_t^u, \omega_t^b$ defined as

$$\omega_t^u \overset{\text{def}}{=} \mathbb{E}_{\boldsymbol{\xi}^t}\left[\widetilde{F}_{\boldsymbol{\xi}^t}(x^t)\right] - \widetilde{F}_{\boldsymbol{\xi}^t}(x^t), \quad \omega_t^b \overset{\text{def}}{=} F(x^t) - \mathbb{E}_{\boldsymbol{\xi}^t}\left[\widetilde{F}_{\boldsymbol{\xi}^t}(x^t)\right], \tag{220}$$

for all $t = 0, \ldots, T$. By definition we have $\omega_t = \omega_t^u + \omega_t^b$ for all $t = 0, \ldots, T$. Using the introduced notation, we continue our derivation as follows: $E_T$ implies

$$R_T^2 \overset{(5)}{\leq} \exp(-\gamma\mu T)R^2 + \underbrace{2\gamma \sum_{t=0}^{T}(1 - \gamma\mu)^{T-t}\langle\eta_t, \omega_t^u\rangle}_{\textcircled{1}} + \underbrace{2\gamma \sum_{t=0}^{T}(1 - \gamma\mu)^{T-t}\langle\eta_t, \omega_t^b\rangle}_{\textcircled{2}}$$

$$+ \underbrace{2\gamma^2 \sum_{t=0}^{T}(1 - \gamma\mu)^{T-t}\mathbb{E}_{\boldsymbol{\xi}^t}\left[\|\omega_t^u\|^2\right]}_{\textcircled{3}} + \underbrace{2\gamma^2 \sum_{t=0}^{T}(1 - \gamma\mu)^{T-t}\left(\|\omega_t^u\|^2 - \mathbb{E}_{\boldsymbol{\xi}^t}\left[\|\omega_t^u\|^2\right]\right)}_{\textcircled{4}}$$

$$+ \underbrace{2\gamma^2 \sum_{t=0}^{T}(1 - \gamma\mu)^{T-t}\left(\|\omega_t^b\|^2\right)}_{\textcircled{5}}. \tag{221}$$

The rest of the proof is based on deriving good enough upper bounds for $\textcircled{1}, \textcircled{2}, \textcircled{3}, \textcircled{4}, \textcircled{5}$, i.e., we want to prove that $\textcircled{1} + \textcircled{2} + \textcircled{3} + \textcircled{4} + \textcircled{5} \leq \exp(-\gamma\mu T)R^2$ with high probability.

Before we move on, we need to derive some useful inequalities for operating with $\omega_t^u, \omega_t^b$. First of all, Lemma B.2 implies that
$$\|\omega_t^u\| \leq 2\lambda_t \tag{222}$$
for all $t = 0, 1, \ldots, T$. Next, since $\{\xi^{i,t}\}_{i=1}^{m_t}$ are independently sampled from $\mathcal{D}$, we have $\mathbb{E}_{\boldsymbol{\xi}^t}[F_{\boldsymbol{\xi}^t}(x^t)] = F(x^t)$, and

$$\mathbb{E}_{\boldsymbol{\xi}^t}\left[\|F_{\boldsymbol{\xi}^t}(x^t) - F(x^t)\|^2\right] = \frac{1}{m_t^2}\sum_{i=1}^{m_t}\mathbb{E}_{\xi^{i,t}}\left[\|F_{\xi^{i,t}}(x^t) - F(x^t)\|^2\right] \overset{(1)}{\leq} \frac{\sigma^2}{m_t},$$

for all $l = 0, 1, \ldots, T$. Moreover, as we already derived, probability event $E_T$ implies that $\|F(x^t)\| \leq \lambda_t/2$ for all $t = 0, 1, \ldots, T$ (see (216)). Therefore, in view of Lemma B.2, $E_T$ implies that

$$\|\omega_t^b\| \leq \frac{4\sigma^2}{m_t\lambda_t}, \tag{223}$$

$$\mathbb{E}_{\boldsymbol{\xi}^t}\left[\|\omega_t\|^2\right] \leq \frac{18\sigma^2}{m_t}, \tag{224}$$

$$\mathbb{E}_{\boldsymbol{\xi}^t}\left[\|\omega_t^u\|^2\right] \leq \frac{18\sigma^2}{m_t}, \tag{225}$$

for all $l = 0, 1, \ldots, T$.

**Upper bound for $\textcircled{1}$.**  Since $\mathbb{E}_{\boldsymbol{\xi}^t}[\omega_t^u] = 0$, we have

$$\mathbb{E}_{\boldsymbol{\xi}^t}\left[2\gamma(1 - \gamma\mu)^{T-t}\langle\eta_t, \omega_t^u\rangle\right] = 0.$$

Next, the summands in $\textcircled{1}$ are bounded with probability 1:

$$\begin{aligned}|2\gamma(1 - \gamma\mu)^{T-t}\langle\eta_t, \omega_t^u\rangle| &\leq 2\gamma\exp(-\gamma\mu(T-t))\|\eta_t\|\cdot\|\omega_t^u\| \\ &\overset{(219),(222)}{\leq} 4\sqrt{2}\gamma(1 + \gamma\ell)\exp(-\gamma\mu(T - t/2))R\lambda_t \\ &\overset{(211),(212)}{\leq} \frac{\exp(-\gamma\mu T)R^2}{5\ln\frac{4(K+1)}{\beta}} \overset{\text{def}}{=} c.\end{aligned} \tag{226}$$

Moreover, these summands have bounded conditional variances $\sigma_t^2 \overset{\text{def}}{=} \mathbb{E}_{\boldsymbol{\xi}^t}\left[4\gamma^2(1 - \gamma\mu)^{2T-2t}\langle\eta_t, \omega_t^u\rangle^2\right]$:

$$\begin{aligned}\sigma_t^2 &\leq \mathbb{E}_{\boldsymbol{\xi}^t}\left[4\gamma^2\exp(-\gamma\mu(2T - 2t))\|\eta_t\|^2\cdot\|\omega_t^u\|^2\right] \\ &\overset{(219)}{\leq} 8\gamma^2(1 + \gamma\ell)^2\exp(-\gamma\mu(2T - t))R^2\mathbb{E}_{\boldsymbol{\xi}^t}\left[\|\omega_t^u\|^2\right] \\ &\overset{(211)}{\leq} 10\gamma^2\exp(-\gamma\mu(2T - t))R^2\mathbb{E}_{\boldsymbol{\xi}^t}\left[\|\omega_t^u\|^2\right].\end{aligned} \tag{227}$$

That is, sequence $\{2\gamma(1-\gamma\mu)^{T-t}\langle\eta_t, \omega_t^u\rangle\}_{t\geq 0}$ is a bounded martingale difference sequence having bounded conditional variances $\{\sigma_t^2\}_{t\geq 0}$. Applying Bernstein's inequality (Lemma B.1) with $X_t = 2\gamma(1-\gamma\mu)^{T-t}\langle\eta_t, \omega_t^u\rangle$, $c$ defined in (226), $b = \frac{1}{5}\exp(-\gamma\mu T)R^2$, $G = \frac{\exp(-2\gamma\mu T)R^4}{150\ln\frac{4(K+1)}{\beta}}$, we get that

$$\mathbb{P}\left\{|①| > \frac{1}{5}\exp(-\gamma\mu T)R^2 \text{ and } \sum_{t=0}^{T}\sigma_t^2 \leq \frac{\exp(-2\gamma\mu T)R^4}{150\ln\frac{4(K+1)}{\beta}}\right\} \leq 2\exp\left(-\frac{b^2}{2G+2cb/3}\right) = \frac{\beta}{2(K+1)}.$$

In other words, $\mathbb{P}\{E_①\} \geq 1 - \frac{\beta}{2(K+1)}$, where probability event $E_①$ is defined as

$$E_① = \left\{\text{either} \quad \sum_{t=0}^{T}\sigma_t^2 > \frac{\exp(-2\gamma\mu T)R^4}{150\ln\frac{4(K+1)}{\beta}} \quad \text{or} \quad |①| \leq \frac{1}{5}\exp(-\gamma\mu T)R^2\right\}. \tag{228}$$

Moreover, we notice here that probability event $E_T$ implies that

$$\sum_{t=0}^{T}\sigma_t^2 \quad \overset{(227)}{\leq} \quad 10\gamma^2\exp(-2\gamma\mu T)R^2\sum_{t=0}^{T}\frac{\mathbb{E}_{\boldsymbol{\xi}^t}\left[\|\omega_t^u\|^2\right]}{\exp(-\gamma\mu t)}$$

$$\overset{(225),T\leq K+1}{\leq} \quad 180\gamma^2\exp(-2\gamma\mu T)R^2\sigma^2\sum_{t=0}^{K}\frac{1}{m_t\exp(-\gamma\mu t)}$$

$$\overset{(213)}{\leq} \quad \frac{\exp(-2\gamma\mu T)R^4}{150\ln\frac{6(K+1)}{\beta}}. \tag{229}$$

**Upper bound for ②.** Probability event $E_T$ implies

$$② \quad \leq \quad 2\gamma\exp(-\gamma\mu T)\sum_{t=0}^{T}\frac{\|\eta_t\|\cdot\|\omega_t^b\|}{\exp(-\gamma\mu t)}$$

$$\overset{(219),(223)}{\leq} \quad 8\sqrt{2}\gamma(1+\gamma\ell)\exp(-\gamma\mu T)R\sum_{t=0}^{T}\frac{\sigma^2}{m_t\lambda_t\exp(-\gamma\mu t/2)}$$

$$\overset{(212)}{\leq} \quad 960\sqrt{2}\gamma^2(1+\gamma\ell)\exp(-\gamma\mu(T-1))\sum_{t=0}^{T}\frac{\sigma^2\ln\frac{4(K+1)}{\beta}}{m_t\exp(-\gamma\mu t)}$$

$$\overset{(213),T\leq K+1}{\leq} \quad \frac{1}{5}\exp(-\gamma\mu T)R^2. \tag{230}$$

**Upper bound for ③.** Probability event $E_T$ implies

$$③ \quad = \quad 2\gamma^2\exp(-\gamma\mu T)\sum_{t=0}^{T}\frac{\mathbb{E}_{\boldsymbol{\xi}^t}\left[\|\omega_t^u\|^2\right]}{\exp(-\gamma\mu t)}$$

$$\overset{(225)}{\leq} \quad 36\gamma^2\exp(-\gamma\mu T)\sum_{t=0}^{T}\frac{\sigma^2}{m_t\exp(-\gamma\mu t)}$$

$$\overset{(213),T\leq K+1}{\leq} \quad \frac{1}{5}\exp(-\gamma\mu T)R^2. \tag{231}$$

**Upper bound for ④.** First of all, we have

$$2\gamma^2(1-\gamma\mu)^{T-t}\mathbb{E}_{\boldsymbol{\xi}^t}\left[\|\omega_t^u\|^2 - \mathbb{E}_{\boldsymbol{\xi}^t}\left[\|\omega_t^u\|^2\right]\right] = 0.$$

Next, the summands in ④ are bounded with probability 1:

$$2\gamma^2(1-\gamma\mu)^{T-t}\left|\|\omega_t^u\|^2 - \mathbb{E}_{\boldsymbol{\xi}^t}\left[\|\omega_t^u\|^2\right]\right| \quad \overset{(222)}{\leq} \quad \frac{16\gamma^2\exp(-\gamma\mu T)\lambda_t^2}{\exp(-\gamma\mu t)}$$

$$\overset{(212)}{\leq} \quad \frac{\exp(-\gamma\mu T)R^2}{5\ln\frac{4(K+1)}{\beta}}$$

$$\overset{\text{def}}{=} \quad c. \tag{232}$$

Moreover, these summands have bounded conditional variances $\widetilde{\sigma}_t^2 \overset{\text{def}}{=} \mathbb{E}_{\boldsymbol{\xi}^t}\left[4\gamma^4(1-\gamma\mu)^{2T-2t}\left|\|\omega_t^u\|^2 - \mathbb{E}_{\boldsymbol{\xi}^t}\left[\|\omega_t^u\|^2\right]\right|^2\right]$:

$$\widetilde{\sigma}_t^2 \overset{(232)}{\leq} \frac{2\gamma^2\exp(-2\gamma\mu T)R^2}{5\exp(-\gamma\mu t)\ln\frac{4(K+1)}{\beta}}\mathbb{E}_{\boldsymbol{\xi}^t}\left[\left|\|\omega_t^u\|^2 - \mathbb{E}_{\boldsymbol{\xi}^t}\left[\|\omega_t^u\|^2\right]\right|\right]$$

$$\leq \frac{4\gamma^2\exp(-2\gamma\mu T)R^2}{5\exp(-\gamma\mu t)\ln\frac{4(K+1)}{\beta}}\mathbb{E}_{\boldsymbol{\xi}^t}\left[\|\omega_t^u\|^2\right]. \tag{233}$$

That is, sequence $\left\{2\gamma^2(1-\gamma\mu)^{T-t}\left(\|\omega_t^u\|^2 - \mathbb{E}_{\boldsymbol{\xi}^t}\left[\|\omega_t^u\|^2\right]\right)\right\}_{t\geq 0}$ is a bounded martingale difference sequence having bounded conditional variances $\{\widetilde{\sigma}_t^2\}_{t\geq 0}$. Applying Bernstein's inequality (Lemma B.1) with $X_t = 2\gamma^2(1-\gamma\mu)^{T-t}\left(\|\omega_t^u\|^2 - \mathbb{E}_{\boldsymbol{\xi}^t}\left[\|\omega_t^u\|^2\right]\right)$, $c$ defined in (232), $b = \frac{1}{5}\exp(-\gamma\mu T)R^2$, $G = \frac{\exp(-2\gamma\mu T)R^4}{150\ln\frac{4(K+1)}{\beta}}$, we get that

$$\mathbb{P}\left\{|④| > \frac{1}{5}\exp(-\gamma\mu T)R^2 \text{ and } \sum_{t=0}^{T}\widetilde{\sigma}_t^2 \leq \frac{\exp(-2\gamma\mu T)R^4}{150\ln\frac{4(K+1)}{\beta}}\right\} \leq 2\exp\left(-\frac{b^2}{2G + \sfrac{2cb}{3}}\right) = \frac{\beta}{2(K+1)}.$$

In other words, $\mathbb{P}\{E_④\} \geq 1 - \frac{\beta}{2(K+1)}$, where probability event $E_④$ is defined as

$$E_④ = \left\{\text{either} \quad \sum_{t=0}^{T}\widetilde{\sigma}_t^2 > \frac{\exp(-2\gamma\mu T)R^4}{150\ln\frac{4(K+1)}{\beta}} \quad \text{or} \quad |④| \leq \frac{1}{5}\exp(-\gamma\mu T)R^2\right\}. \tag{234}$$

Moreover, we notice here that probability event $E_T$ implies that

$$\sum_{t=0}^{T}\widetilde{\sigma}_t^2 \overset{(233)}{\leq} \frac{4\gamma^2\exp(-2\gamma\mu T)R^2}{5\ln\frac{4(K+1)}{\beta}}\sum_{t=0}^{T}\frac{\mathbb{E}_{\boldsymbol{\xi}^t}\left[\|\omega_t^u\|^2\right]}{\exp(-\gamma\mu t)}$$

$$\overset{(225),T\leq K+1}{\leq} \frac{72\gamma^2\exp(-2\gamma\mu T)R^2\sigma^2}{5\ln\frac{4(K+1)}{\beta}}\sum_{t=0}^{K}\frac{1}{m_t\exp(-\gamma\mu t)}$$

$$\overset{(213)}{\leq} \frac{\exp(-2\gamma\mu T)R^4}{150\ln\frac{4(K+1)}{\beta}}. \tag{235}$$

**Upper bound for ⑤.** Probability event $E_T$ implies

$$⑤ \quad = \quad 2\gamma^2\sum_{t=0}^{T}\exp(-\gamma\mu(T-t))\left(\|\omega_t^b\|^2\right)$$

$$\overset{(223)}{\leq} \quad 32\gamma^2\exp(-\gamma\mu T)\sum_{t=0}^{T}\frac{\sigma^4}{m_t^2\lambda_t^2\exp(-\gamma\mu t)}$$

$$\overset{(212)}{=} \quad 460800\gamma^4\exp(-\gamma\mu(T-2))\sum_{t=0}^{T}\frac{\sigma^4\ln^2\frac{4(K+1)}{\beta}}{m_t^2R^2\exp(-2\gamma\mu t)}$$

$$\overset{(213),T\leq K+1}{\leq} \quad \frac{1}{5}\exp(-\gamma\mu T)R^2. \tag{236}$$

**Final derivation.** Putting all bounds together, we get that $E_T$ implies

$$R_T^2 \overset{(221)}{\leq} \exp(-\gamma\mu T)R^2 + ① + ② + ③ + ④ + ⑤,$$

$$② \overset{(230)}{\leq} \frac{1}{5}\exp(-\gamma\mu T)R^2,$$

$$③ \overset{(231)}{\leq} \frac{1}{5}\exp(-\gamma\mu T)R^2, \quad ⑤ \overset{(236)}{\leq} \frac{1}{5}\exp(-\gamma\mu T)R^2,$$

$$\sum_{t=0}^{T}\sigma_t^2 \overset{(229)}{\leq} \frac{\exp(-2\gamma\mu T)R^4}{150\ln\frac{4(K+1)}{\beta}}, \quad \sum_{t=0}^{T}\widetilde{\sigma}_t^2 \overset{(235)}{\leq} \frac{\exp(-2\gamma\mu T)R^4}{150\ln\frac{4(K+1)}{\beta}}.$$

Moreover, in view of (228), (234), and our induction assumption, we have

$$\mathbb{P}\{E_T\} \geq 1 - \frac{T\beta}{K+1},$$

$$\mathbb{P}\{E_{①}\} \geq 1 - \frac{\beta}{2(K+1)}, \quad \mathbb{P}\{E_{④}\} \geq 1 - \frac{\beta}{2(K+1)},$$

where probability events $E_{①}$, and $E_{④}$ are defined as

$$E_{①} = \left\{\text{either} \quad \sum_{t=0}^{T} \sigma_t^2 > \frac{\exp(-2\gamma\mu T)R^4}{150\ln\frac{4(K+1)}{\beta}} \quad \text{or} \quad |①| \leq \frac{1}{5}\exp(-\gamma\mu T)R^2\right\},$$

$$E_{④} = \left\{\text{either} \quad \sum_{t=0}^{T} \widetilde{\sigma}_t^2 > \frac{\exp(-2\gamma\mu T)R^4}{150\ln\frac{4(K+1)}{\beta}} \quad \text{or} \quad |④| \leq \frac{1}{5}\exp(-\gamma\mu T)R^2\right\}.$$

Putting all of these inequalities together, we obtain that probability event $E_T \cap E_{①} \cap E_{④}$ implies

$$R_T^2 \overset{(221)}{\leq} \exp(-\gamma\mu T)R^2 + ① + ② + ③ + ④ + ⑤$$
$$\leq 2\exp(-\gamma\mu T)R^2.$$

Moreover, union bound for the probability events implies

$$\mathbb{P}\{E_{T+1}\} \geq \mathbb{P}\{E_T \cap E_{①} \cap E_{④}\} = 1 - \mathbb{P}\{\overline{E}_T \cup \overline{E}_{①} \cup \overline{E}_{④}\} \geq 1 - \frac{(T+1)\beta}{K+1}. \tag{237}$$

This is exactly what we wanted to prove (see the paragraph after inequality (215)). In particular, with probability at least $1 - \beta$ we have

$$\|x^{K+1} - x^*\|^2 \leq 2\exp(-\gamma\mu(K+1))R^2,$$

which finishes the proof. $\qquad\square$

**Corollary D.3.** *Let the assumptions of Theorem D.3 hold. Then, the following statements hold.*

1. ***Large stepsize/large batch.*** *The choice of stepsize and batchsize*

$$\gamma = \frac{1}{400\ell\ln\frac{4(K+1)}{\beta}}, \quad m_k = \max\left\{1, \frac{27000\gamma^2(K+1)\sigma^2\ln\frac{4(K+1)}{\beta}}{\exp(-\gamma\mu k)R^2}\right\} \tag{238}$$

*satisfies conditions (211) and (213). With such choice of $\gamma, m_k$, and the choice of $\lambda_k$ as in (212), the iterates produced by* clipped-SGDA *after $K$ iterations with probability at least $1 - \beta$ satisfy*

$$\|x^{K+1} - x^*\|^2 \leq 2\exp\left(-\frac{\mu(K+1)}{400\ell\ln\frac{4(K+1)}{\beta}}\right)R^2. \tag{239}$$

*In particular, to guarantee $\|x^{K+1} - x^*\|^2 \leq \varepsilon$ with probability at least $1 - \beta$ for some $\varepsilon > 0$* clipped-SGDA *requires*

$$\mathcal{O}\left(\frac{\ell}{\mu}\ln\left(\frac{R^2}{\varepsilon}\right)\ln\left(\frac{\ell}{\mu\beta}\ln\left(\frac{R^2}{\varepsilon}\right)\right)\right) \text{ iterations,} \tag{240}$$

$$\mathcal{O}\left(\max\left\{\frac{\ell}{\mu}, \frac{\sigma^2}{\mu^2\varepsilon}\right\}\ln\left(\frac{R^2}{\varepsilon}\right)\ln\left(\frac{\ell}{\mu\beta}\ln\left(\frac{R^2}{\varepsilon}\right)\right)\right) \text{ oracle calls.} \tag{241}$$

2. ***Small stepsize/small batch.*** *The choice of stepsize and batchsize*

$$\gamma = \min\left\{\frac{1}{400\ell\ln\frac{4(K+1)}{\beta}}, \frac{\ln(B_K)}{\mu(K+1)}\right\}, \quad m_k \equiv 1 \tag{242}$$

*satisfies conditions* (211) *and* (213), *where* $B_K = \max\left\{2, \frac{(K+1)\mu^2 R^2}{27000\sigma^2 \ln\left(\frac{4(K+1)}{\beta}\right)\ln^2(B_K)}\right\} =$

$\mathcal{O}\left(\max\left\{2, \frac{(K+1)\mu^2 R^2}{27000\sigma^2 \ln\left(\frac{4(K+1)}{\beta}\right)\ln^2\left(\max\left\{2, \frac{(K+1)\mu^2 R^2}{27000\sigma^2 \ln\left(\frac{4(K+1)}{\beta}\right)}\right\}\right)}\right\}\right)$. *With such choice of*

$\gamma, m_k$, *and the choice of* $\lambda_k$ *as in* (212), *the iterates produced by* clipped-SGDA *after* $K$
*iterations with probability at least* $1 - \beta$ *satisfy*

$$\|x^{K+1}-x^*\|^2 \le \max\left\{2\exp\left(-\frac{\mu(K+1)}{400\ell\ln\frac{4(K+1)}{\beta}}\right)R^2, \frac{54000\sigma^2\ln\left(\frac{4(K+1)}{\beta}\right)\ln^2(B_K)}{\mu^2(K+1)}\right\}.$$
(243)

*In particular, to guarantee* $\|x^{K+1} - x^*\|^2 \le \varepsilon$ *with probability at least* $1 - \beta$ *for some*
$\varepsilon > 0$ clipped-SGDA *requires*

$$\mathcal{O}\left(\max\left\{\frac{\ell}{\mu}\ln\left(\frac{R^2}{\varepsilon}\right)\ln\left(\frac{\ell}{\mu\beta}\ln\left(\frac{R^2}{\varepsilon}\right)\right), \frac{\sigma^2}{\mu^2\varepsilon}\ln\left(\frac{\sigma^2}{\mu^2\varepsilon\beta}\right)\ln^2(B_\varepsilon)\right\}\right)$$
(244)

*iterations/oracle calls, where*

$$B_\varepsilon = \max\left\{2, \frac{R^2}{\varepsilon\ln\left(\frac{\sigma^2}{\mu^2\varepsilon\beta}\right)\ln^2\left(\max\left\{2, \frac{R^2}{\varepsilon\ln\left(\frac{\sigma^2}{\mu^2\varepsilon\beta}\right)}\right\}\right)}\right\}.$$

*Proof.* 1. **Large stepsize/large batch.** First of all, it is easy to see that the choice of $\gamma$ and
$m_k$ from (238) satisfies conditions (211) and (213). Therefore, applying Theorem D.3, we
derive that with probability at least $1 - \beta$

$$\|x^{K+1} - x^*\|^2 \le 2\exp(-\gamma\mu(K+1))R^2 \overset{(238)}{=} 2\exp\left(-\frac{\mu(K+1)}{400\ell\ln\frac{4(K+1)}{\beta}}\right)R^2.$$

To guarantee $\|x^{K+1} - x^*\|^2 \le \varepsilon$, we choose $K$ in such a way that the right-hand side of
the above inequality is smaller than $\varepsilon$ that gives

$$K = \mathcal{O}\left(\frac{\ell}{\mu}\ln\left(\frac{R^2}{\varepsilon}\right)\ln\left(\frac{\ell}{\mu\beta}\ln\left(\frac{R^2}{\varepsilon}\right)\right)\right).$$

The total number of oracle calls equals

$$\sum_{k=0}^{K} m_k \overset{(238)}{=} \sum_{k=0}^{K}\max\left\{1, \frac{27000\gamma^2(K+1)\sigma^2\ln\frac{4(K+1)}{\beta}}{\exp(-\gamma\mu k)R^2}\right\}$$

$$= \mathcal{O}\left(\max\left\{K, \frac{\gamma(K+1)\exp(\gamma\mu(K+1))\sigma^2\ln\frac{4(K+1)}{\beta}}{\mu R^2}\right\}\right)$$

$$= \mathcal{O}\left(\max\left\{\frac{\ell}{\mu}, \frac{\sigma^2}{\mu^2\varepsilon}\right\}\ln\left(\frac{R^2}{\varepsilon}\right)\ln\left(\frac{\ell}{\mu\beta}\ln\left(\frac{R^2}{\varepsilon}\right)\right)\right).$$

2. **Small stepsize/small batch.** First of all, we verify that the choice of $\gamma$ and $m_k$ from (242)
satisfies conditions (211) and (213): (211) trivially holds and (213) holds since for all
$k = 0, \ldots, K$

$$\frac{27000\gamma^2(K+1)\sigma^2\ln\frac{4(K+1)}{\beta}}{\exp(-\gamma\mu k)R^2} \le \frac{27000\gamma^2(K+1)\sigma^2\ln\frac{4(K+1)}{\beta}}{\exp(-\gamma\mu(K+1))R^2}$$

$$\overset{(242)}{\le} \frac{27000\ln^2(B_K)\exp(\gamma\mu(K+1))\sigma^2\ln\frac{4(K+1)}{\beta}}{\mu^2(K+1)R^2}$$

$$\overset{(242)}{\le} 1.$$

Therefore, applying Theorem D.3, we derive that with probability at least $1 - \beta$

$$
\begin{aligned}
\|x^{K+1} - x^*\|^2 \quad &\leq \quad 2\exp(-\gamma\mu(K+1))R^2 \\
&\overset{(242)}{=} \quad \max\left\{2\exp\left(-\frac{\mu(K+1)}{400\ell\ln\frac{4(K+1)}{\beta}}\right)R^2, \frac{2R^2}{B_K}\right\} \\
&= \quad \max\left\{2\exp\left(-\frac{\mu(K+1)}{400\ell\ln\frac{4(K+1)}{\beta}}\right)R^2, \frac{54000\sigma^2\ln\left(\frac{4(K+1)}{\beta}\right)\ln^2(B_K)}{\mu^2(K+1)}\right\}.
\end{aligned}
$$

To guarantee $\|x^{K+1} - x^*\|^2 \leq \varepsilon$, we choose $K$ in such a way that the right-hand side of the above inequality is smaller than $\varepsilon$ that gives $K$ of the order

$$
\mathcal{O}\left(\max\left\{\frac{\ell}{\mu}\ln\left(\frac{R^2}{\varepsilon}\right)\ln\left(\frac{\ell}{\mu\beta}\ln\left(\frac{R^2}{\varepsilon}\right)\right), \frac{\sigma^2}{\mu^2\varepsilon}\ln\left(\frac{\sigma^2}{\mu^2\varepsilon\beta}\right)\ln^2(B_\varepsilon)\right\}\right),
$$

where

$$
B_\varepsilon = \max\left\{2, \frac{R^2}{\varepsilon\ln\left(\frac{\sigma^2}{\mu^2\varepsilon\beta}\right)\ln^2\left(\max\left\{2, \frac{R^2}{\varepsilon\ln\left(\frac{\sigma^2}{\mu^2\varepsilon\beta}\right)}\right\}\right)}\right\}.
$$

The total number of oracle calls equals $\sum_{k=0}^{K} m_k = (K+1)$.

$\square$

# E Extra Experiments

In this section, we provide more details for the experiments done in § 4, as well as additional tables, figures, and image samples from some of our trained models.

## E.1 WGAN-GP

In all cases, everything in the experimental setup other than learning rates and clip values remained constant. We use the same ResNet architectures and training parameters specified in Gulrajani et al. [2017]: the gradient penalty coefficient $\lambda_{GP} = 10$, $n_{dis} = 5$ where $n_{dis}$ is the number of discriminator steps for every generator step, and a learning rate decayed linearly to 0 over 100k steps. The only exception is we double the feature map of the generator from 128 to 256 dimensions. For all stochastic extragradient (SEG) methods, we use the ExtraSGD implementation provided by Gidel et al. [2019a]. We alternate between exploration and update steps and do not treat the exploration steps as "free" – this means we only have 50k parameter updates as opposed to 100k for all SGDA methods (we decay the learning rate twice as fast such that it still reaches 0 after 50k parameter updates).

All of the hyperparameter sweeps performed for SGDA, clipped-SGDA, clipped-SEG, clipped-SGDA (coordinate), and clipped-SEG (coordinate), as well as the associated best FID score obtained within the first 35k training steps, can be found in Tables 2, 6, 5, 6, and 7 respectively. **Bold** rows denote the hyperparameters that were trained for the full 100k steps and are henceforth referred to as the *"best models"*. For each of the methods tested, additional samples for the best models trained can be found in Figures 7, 8, 9, 10, & 11. We also plot the evolution of the gradient noise histograms in Figures 12, 13, 14, 15, & 16. We emphasize that our goal is not to get the best possible FID score (e.g. are often able to obtain marginally better FIDs by training for longer), but rather to compare the systematic differences in performance between the various unclipped and clipped methods. Therefore, log-space hyperparameter sweeps are appropriate for our experiments and we do not tune further.

## E.2 StyleGAN2

We train on FFHQ downsampled to $128 \times 128$ pixels, and use the recommended StyleGAN2 hyperparameter configuration for this resolution: batch size $= 32$, $\gamma = 0.1024$, map depth $= 2$, and channel multiplier $= 16384$. For both SGDA and clipped-SGDA, we sweep over a (roughly) log-scale of learning rates and clipping values; a summary of the hyperparmaters and best FID scores obtained Table 8 and Table 9 respectively.

Based on the results in Table 9, the best hyperparameters are lr=0.35 and clip=0.0025 which we then used to train our *"best model"*. We trained for longer, and decayed the learning rate twice (by a factor of $\times 10$) when the FID plateaued or worsened. The best schedule we found was to scale the learning rate by $\times 0.1$ after 6000 kimgs (thousands of real images shown to the discriminator), by another $\times 0.1$ after 3600 kimgs, and then train until the FID begins increasing (for another 8000 kimgs) – for a total of 17600 kimgs. We did not explore different scale factors or other schedules (such as cosine annealing). Additional samples for this model can be found in Figure 18(a).

In general, we observe that every SGDA-trained model for the wide range of learning rates we tested failed to improve the FID, while models trained with clipped-SGDA (with appropriately set hyperparameters) are generally able to learn some meaningful features and improve the FID. We show this behaviour in Figure 17 – the FID scores for SGDA-trained models fluctuate around 320 and only generate noise such as the samples shown in Figure 18(b), which is in contrast to models trained with clipped-SGDA. Note that the range of the hyperparameter sweep is fairly narrow and favourable for clipped-SGDA, while being quite wide for SGDA. The purpose for these parameter ranges is not to directly compare the parameter sweeps (which would unfairly favour clipped-SGDA), but to show that in general SGDA fails, while clipped-SGDA is capable of learning.

Table 2: SGDA hyperparameter sweep, and the best FID score obtained in 35k training steps.

| G-LR | D-LR | FID |
|---|---|---|
| 6e-06 | 6e-06 | 233.3 |
| 2e-05 | 2e-05 | 177.2 |
| 2e-05 | 4e-05 | 183.4 |
| 2e-05 | 8e-05 | 187.3 |
| 0.0002 | 0.0002 | 85.6 |
| **0.0002** | **0.0004** | **82.8** |
| 0.0002 | 0.0008 | NaN |
| 0.002 | 0.002 | NaN |
| 0.02 | 0.02 | NaN |
| 0.2 | 0.2 | NaN |

Table 3: SEG hyperparameter sweep, and the best FID score obtained in 35k training steps.

| G-LR | D-LR | FID |
|---|---|---|
| 6e-06 | 6e-06 | 236.1 |
| 2e-05 | 2e-05 | 208.6 |
| 2e-05 | 4e-05 | 213.7 |
| **4e-05** | **4e-05** | **176.5** |
| 4e-05 | 0.0001 | NaN |
| 0.0002 | 0.0002 | NaN |
| 0.0002 | 0.0004 | NaN |
| 0.0002 | 0.0008 | NaN |
| 0.002 | 0.002 | NaN |
| 0.02 | 0.02 | NaN |
| 0.2 | 0.2 | NaN |
| 2 | 2 | NaN |

Table 4: clipped-SGDA (norm) hyperparameter sweep, and the best FID score obtained in 35k training steps.

| G-LR | D-LR | G-clip | D-clip | FID |
|---|---|---|---|---|
| 0.002 | 0.002 | 0.1 | 0.1 | 257.6 |
| 0.002 | 0.002 | 1 | 1 | 121.6 |
| 0.002 | 0.002 | 10 | 10 | 145.4 |
| 0.02 | 0.02 | 0.1 | 0.1 | 115.4 |
| 0.02 | 0.02 | 1 | 1 | 141.8 |
| 0.02 | 0.02 | 10 | 10 | 27.4 |
| 0.2 | 0.2 | 0.1 | 0.1 | 133.0 |
| **0.2** | **0.2** | **1** | **1** | **26.3** |
| 2 | 2 | 0.1 | 0.1 | 26.1 |

Table 5: clipped-SEG (norm) hyperparameter sweep, and the best FID score obtained in 35k training steps (17.5k parameter updates).

| G-LR | D-LR | G-clip | D-clip | FID |
|---|---|---|---|---|
| 0.002 | 0.002 | 0.1 | 0.1 | 232.5 |
| 0.002 | 0.002 | 1 | 1 | 150.5 |
| 0.002 | 0.002 | 10 | 10 | 192.7 |
| 0.02 | 0.02 | 0.1 | 0.1 | 161.0 |
| 0.02 | 0.02 | 1 | 1 | 160.3 |
| 0.02 | 0.02 | 10 | 10 | 39.3 |
| 0.2 | 0.2 | 0.1 | 0.1 | 160.0 |
| **0.2** | **0.2** | **1** | **1** | **36.3** |
| 2 | 2 | 0.1 | 0.1 | 37.7 |

Table 6: clipped-SGDA (coordinate) hyperparameter sweep, and the best FID score obtained in 35k training steps.

| G-LR | D-LR | G-clip | D-clip | FID |
|---|---|---|---|---|
| 0.0002 | 0.0002 | 0.001 | 0.001 | 292.2 |
| 0.0002 | 0.0002 | 0.01 | 0.01 | 108.6 |
| 0.0002 | 0.0002 | 0.1 | 0.1 | 91.5 |
| 0.002 | 0.002 | 0.001 | 0.001 | 76.5 |
| 0.002 | 0.002 | 0.01 | 0.01 | 43.5 |
| 0.002 | 0.002 | 0.1 | 0.1 | 45.1 |
| 0.02 | 0.02 | 0.001 | 0.001 | 37.3 |
| **0.02** | **0.02** | **0.01** | **0.01** | **26.7** |
| 0.02 | 0.02 | 0.1 | 0.1 | 34.7 |

Table 7: clipped-SEG (coordinate) hyperparameter sweep, and the best FID score obtained in 35k training steps (17.5k parameter updates).

| G-LR | D-LR | G-clip | D-clip | FID |
|---|---|---|---|---|
| 0.0002 | 0.0002 | 0.001 | 0.001 | 298.7 |
| 0.0002 | 0.0002 | 0.01 | 0.01 | 146.5 |
| 0.0002 | 0.0002 | 0.1 | 0.1 | 158.4 |
| 0.002 | 0.002 | 0.001 | 0.001 | 112.8 |
| 0.002 | 0.002 | 0.01 | 0.01 | 52.7 |
| 0.002 | 0.002 | 0.1 | 0.1 | 66.5 |
| 0.02 | 0.02 | 0.001 | 0.001 | 43.5 |
| **0.02** | **0.02** | **0.01** | **0.01** | **36.2** |
| 0.02 | 0.02 | 0.1 | 0.1 | 75.3 |

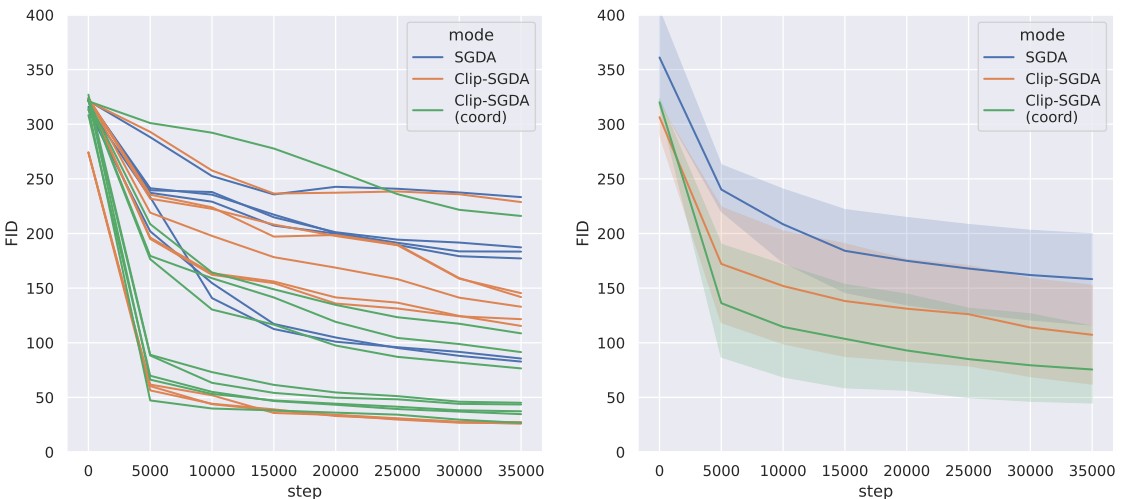

Figure 5: FID curves when training WGAN-GP for 35k steps with SGDA, clipped-SGDA (norm), and clipped-SGDA (coordinate), corresponding to the hyperparameters in Tables 2, 4 & 6 respectively. The left figure is the individual runs for each choice of hyperparameters, and the right is the mean and 95% confidence interval of these runs. Note that 4 of 10 runs diverged (NaN loss) for SGDA, which is not reflected in the mean FID for the right figure beyond the first step.

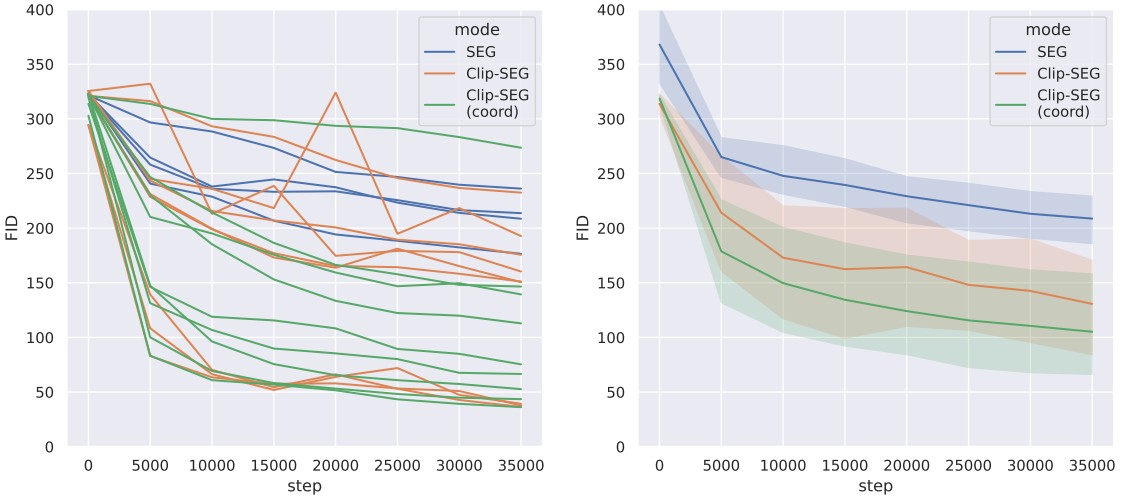

Figure 6: FID curves when training WGAN-GP for 35k steps with SEG, clipped-SEG (norm), and clipped-SEG (coordinate), corresponding to the hyperparameters in Tables 3, 5 & 7 respectively. The left figure is the individual runs for each choice of hyperparameters, and the right is the mean and 95% confidence interval of these runs. Note that 8 of 12 runs diverged (NaN loss) for SEG, which is not reflected in the mean FID for the right figure beyond the first step.

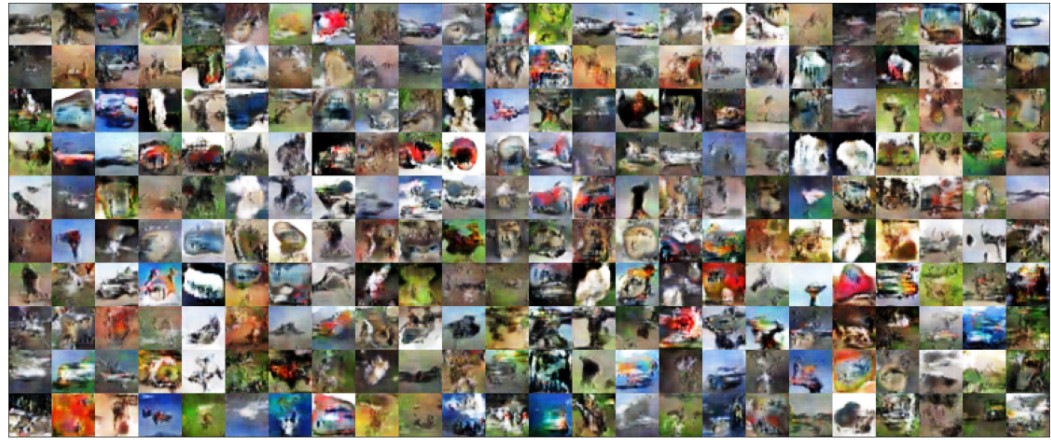

Figure 7: Samples generated from the best WGAN-GP model trained with SGDA.

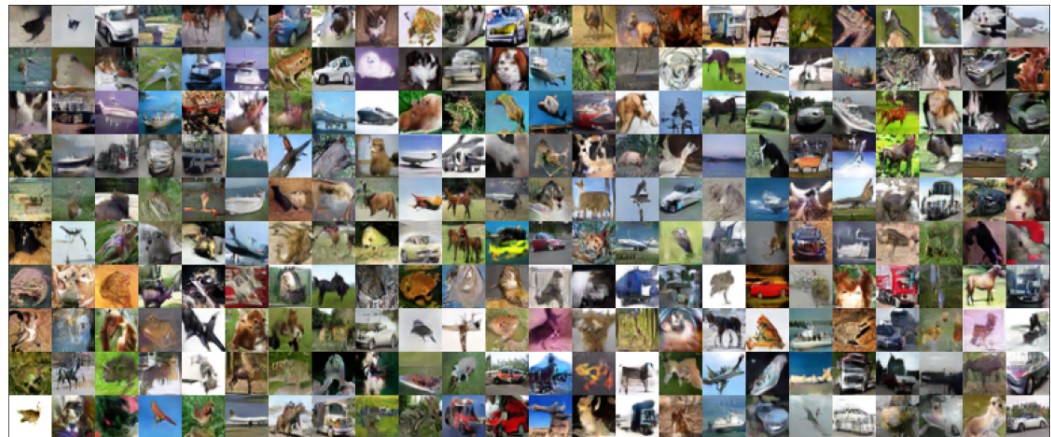

Figure 8: Samples generated from the best WGAN-GP model trained with clipped-SGDA.

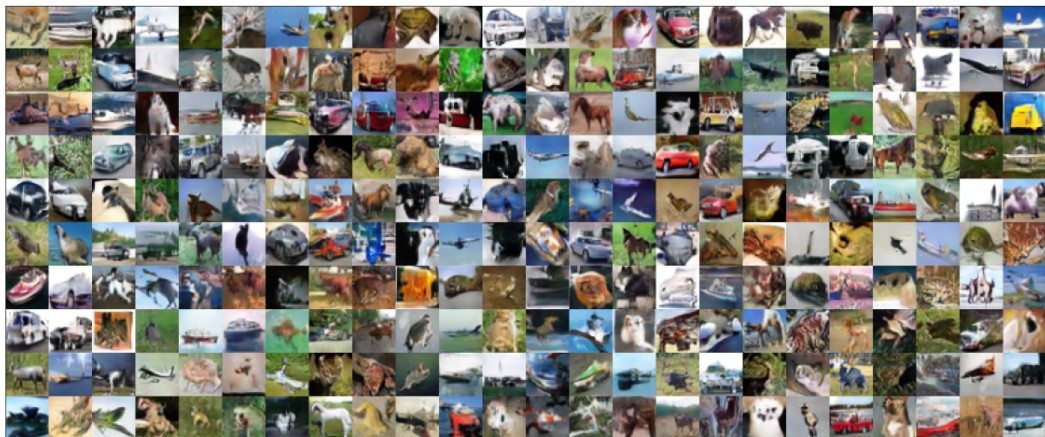

Figure 9: Samples generated from the best WGAN-GP model trained with clipped-SEG.

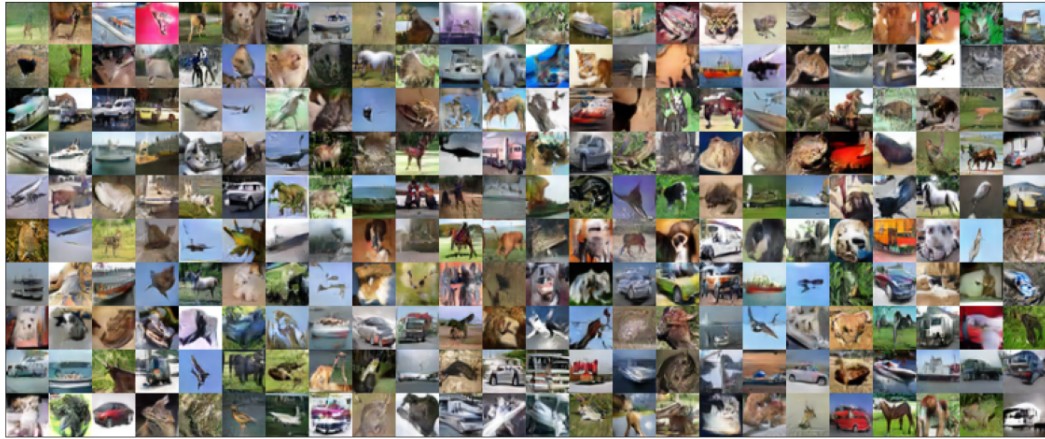

Figure 10: Samples generated from the best WGAN-GP model trained with clipped-SGDA (coordinate clipping).

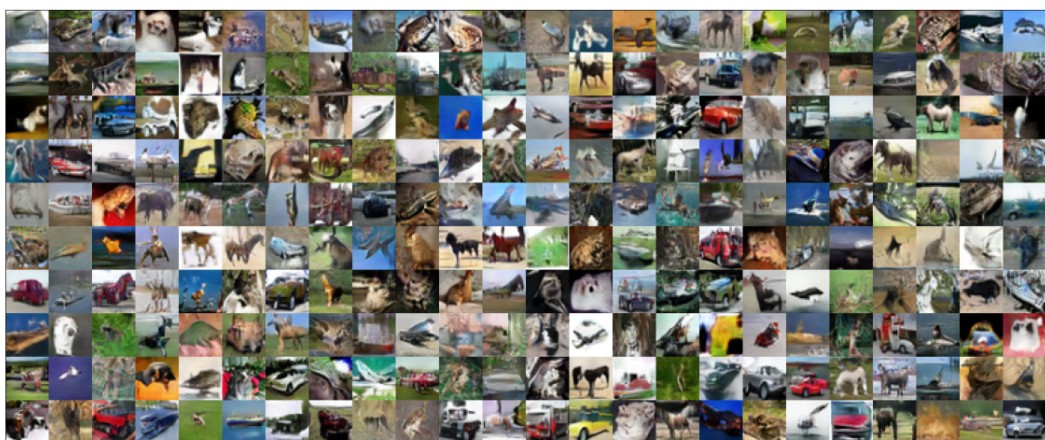

Figure 11: Samples generated from the best WGAN-GP model trained with clipped-SEG (coordinate clipping).

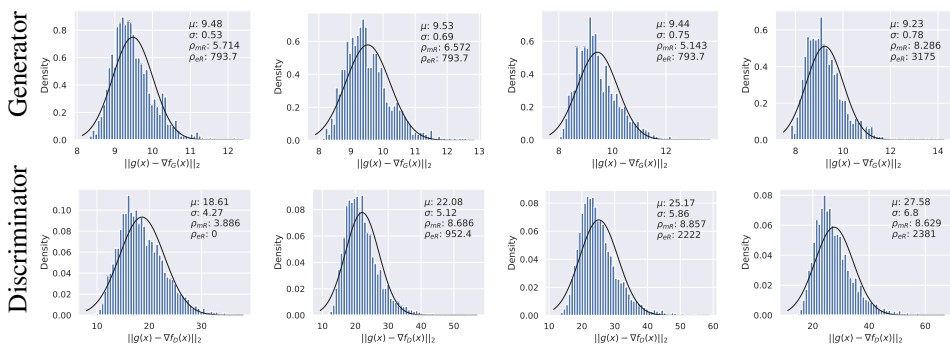

Figure 12: Evolution of gradient noise histograms for the best WGAN-GP model trained with SGDA.

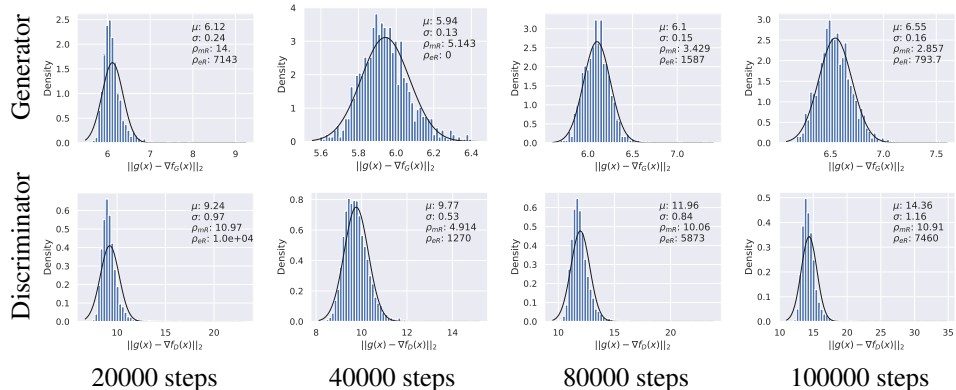

Figure 13: Evolution of gradient noise histograms for the best WGAN-GP model trained with clipped-SGDA.

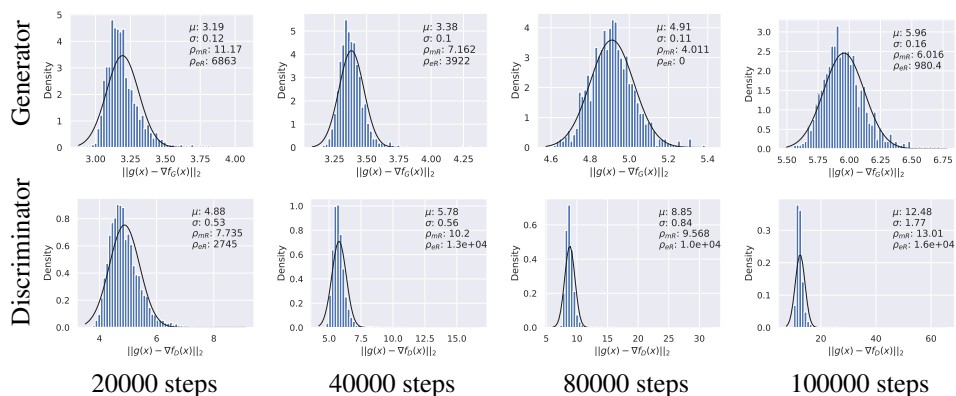

Figure 14: Evolution of gradient noise histograms for the best WGAN-GP model trained with clipped-SEG.

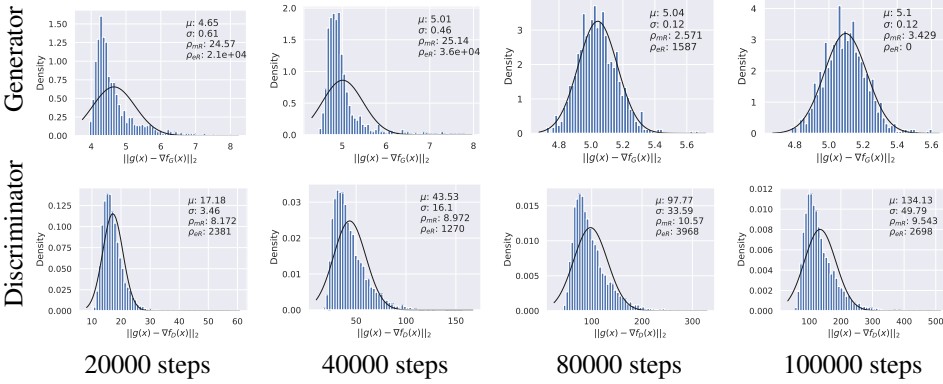

Figure 15: Evolution of gradient noise histograms for the best WGAN-GP model trained with clipped-SGDA (cordinate clipping).

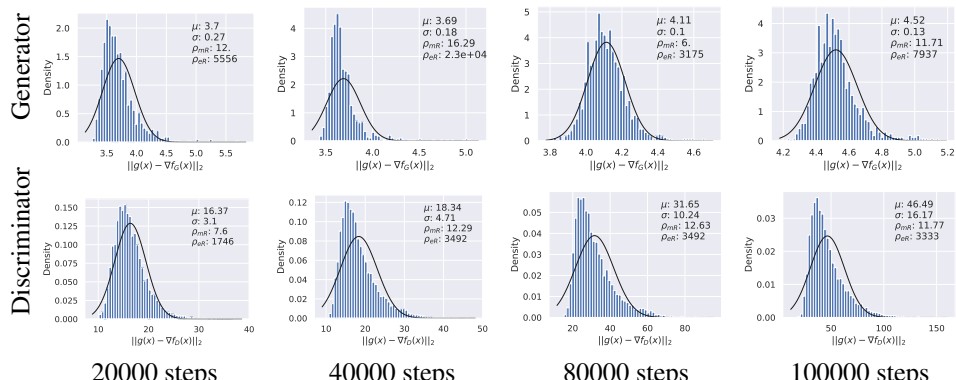

Figure 16: Evolution of gradient noise histograms for the best WGAN-GP model trained with clipped-SEG (cordinate clipping).

Table 8: StyleGAN2 SGDA hyperparameter sweep, and the best FID score obtained in 2600 kimgs.

| G-LR | D-LR | FID |
|---|---|---|
| 0.003 | 0.003 | 319.7 |
| 0.0075 | 0.0075 | 318.5 |
| 0.01 | 0.01 | 317.7 |
| 0.035 | 0.035 | 301.9 |
| 0.05 | 0.05 | 300.3 |
| 0.075 | 0.075 | 299.6 |
| 0.1 | 0.1 | 308.5 |
| 0.35 | 0.35 | 342.6 |
| 0.5 | 0.5 | 346.6 |

Table 9: StyleGAN2 clipped-SGDA (coordinate) hyperparameter sweep, and the best FID score obtained in 2600 kimgs. Bold row denotes the best run which was trained to convergence.

| G-LR | D-LR | G-clip | D-clip | FID |
|---|---|---|---|---|
| 0.2 | 0.2 | 0.001 | 0.001 | 243.5 |
| 0.3 | 0.3 | 0.001 | 0.001 | 169.5 |
| 0.35 | 0.35 | 0.0005 | 0.0005 | 192.9 |
| 0.35 | 0.35 | 0.001 | 0.001 | 148.6 |
| **0.35** | **0.35** | **0.0025** | **0.0025** | **104.9** |
| 0.35 | 0.35 | 0.005 | 0.005 | 149.1 |
| 0.35 | 0.5 | 0.01 | 0.01 | 170.6 |
| 0.4 | 0.4 | 0.001 | 0.001 | 155.8 |
| 0.5 | 0.5 | 0.0001 | 0.0001 | 289.8 |
| 0.5 | 0.5 | 0.001 | 0.001 | 136.1 |

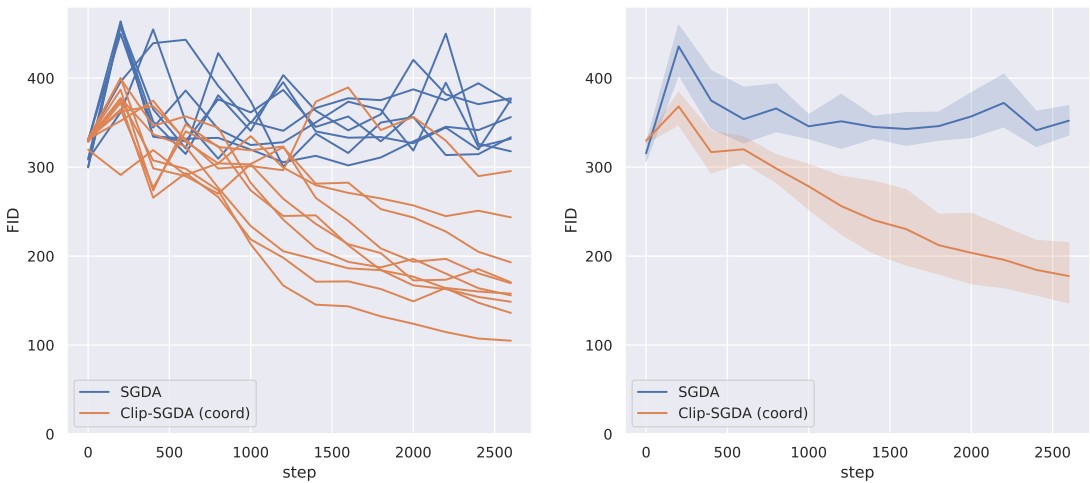

Figure 17: FID curves when training StyleGAN2 for 2600 kimgs (thousands of images seen by the discriminator) with SGDA and clipped-SGDA (coordinate), corresponding to the hyperparameters in Tables 8 & 9 respectively. The left figure is the individual runs for each choice of hyperparameters, and the right is the mean and 95% confidence interval of these runs. Every SGDA-trained model for the wide range of learning rates we tried failed to improve the FID, while models trained with clipped-SGDA (with appropriately set hyperparameters) are generally able to learn some meaningful features and improve the FID.

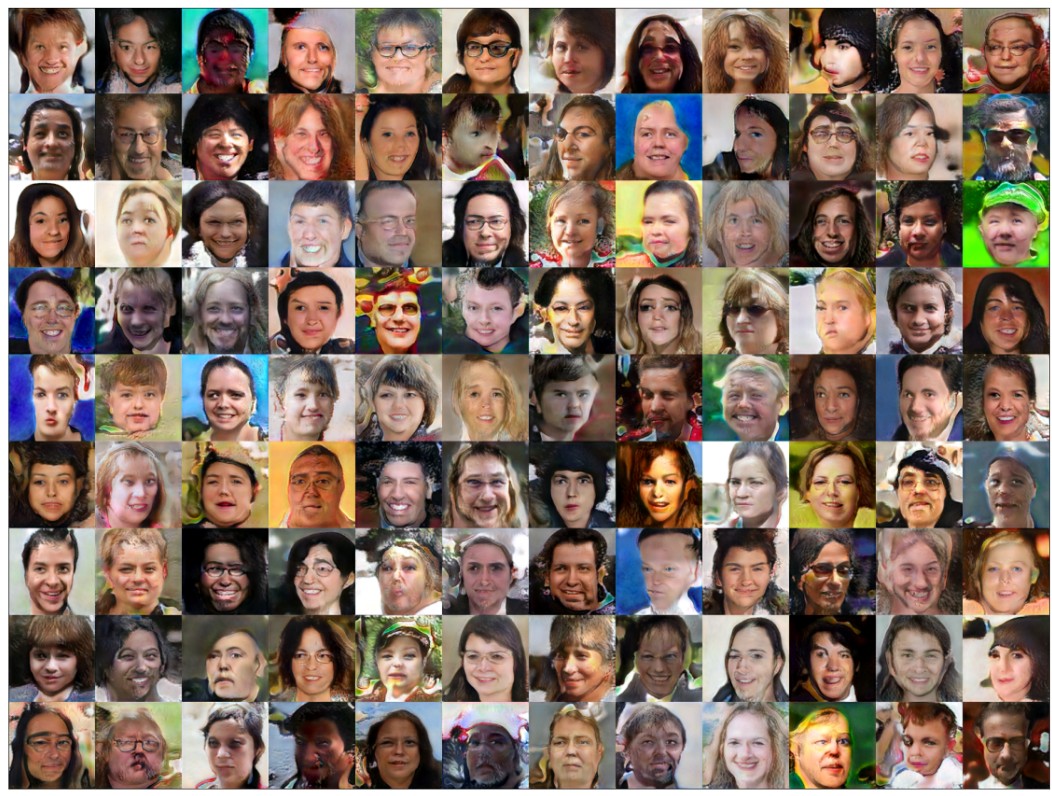

(a) Additional samples generated from our best model trained with clipped-SGDA (lr=0.35, clip=0.0025).

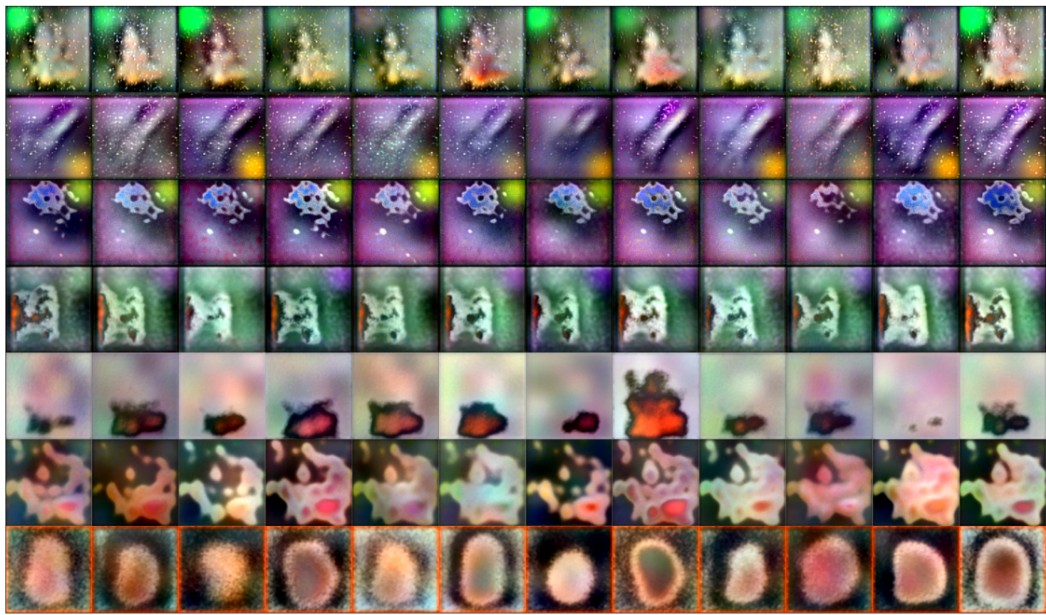

(b) Additional samples generated from several different SGDA trained models, all of which failed to generate meaningful features. Each row corresponds to a model trained with different learning rates.

Figure 18: More StyleGAN2 samples.