# OpenReview forum: "Clipped Stochastic Methods for Variational Inequalities with Heavy-Tailed Noise"
_NeurIPS.cc/2022/Conference — NeurIPS 2022 Accept_

### Official Review · Reviewer_nza4 · 2022-07-12

**Rating:** 7
**Confidence:** 3
**Soundness:** 4 excellent
**Presentation:** 3 good
**Contribution:** 3 good

**Summary:**

This paper studies variational inequality problem (VIP) with heavy-tailed noise and unbounded domain. It proposes a stochastic extragradient (SEG) method and a stochastic gradient descent-ascent (SGDA) method with gradient clipping techinique to address the problem. To handle the heavy-tailed noise, the paper considers clipping value for stochastic gradients, which controls the impact from heavy-tailedness but also introduces bias. The bias derived is addressed by different step-sizes and batch-sizes. To handle the unbounded domain, the paper only requires an upper bound of the distance between the initialization and the optimal solution. The convergence analysis on both stochastic methods and their proofs are discussed respectively in monotonic, star-monotonic and quasi-strong monotonic cases according to the respective property of the operator in the VIP.

**Questions:**

In the extra experiments of the paper, we can find the FID curves on training StyleGAN2 with the original SGDA method and the clipping-value SGDA method. Can you provide the similar results on training WGAN-GP with the original SEG method and the clipping-value SEG method?

**Limitations:**

In its experiments, the paper does not compare the results of its stochastic methods with those of Adam.

**Strengths And Weaknesses:**

Originality: Although stochastic extragradient method, stochastic gradient descent-ascent method and gradient clipping techinique are well-studied in the literature, this paper successfully combines them and applies to the VIP with heavy-tailed noise, inidcating authors' original effort.

Quality: This paper gives the convergence results for monotonic, star-monotonic and quasi-strong monotonic operators in the VIP, which is comprehensive and readable for practitioners.

Clarity: The structure of this paper is well designed. The table of results in the introduction is straightforward.

Significance: This paper addresses VIP with heavy-tailed noise, which is common in industries and business.

---

> ### Author Response · Authors · 2022-08-01
> **Authors' response to Official Review of Paper9065 by Reviewer nza4**
>
> We thank the reviewer for such a positive evaluation of our work once again.
>
> ---
>
> > In the extra experiments of the paper, we can find the FID curves on training StyleGAN2 with the original SGDA method and the clipping-value SGDA method. Can you provide the similar results on training WGAN-GP with the original SEG method and the clipping-value SEG method?
>
> We have added the requested figures for WGAN-GP to the appendix. In addition to comparing the FID curves for SEG and clipped-SEG (coordinate), we also show the FID curves for every scenario we have tested for our hyperparameter sweep experiments (i.e. SEG, clipped-SEG (coordinate/value), SGDA, and clipped-SGDA (coordinate/value). These can be found in figures 5 and 6 in the appendix.
>
> ---
>
> > In its experiments, the paper does not compare the results of its stochastic methods with those of Adam.
>
> The goal of our experiments is to complement our theory with experiments, rather than to provide a new state of the art technique (which would require the use of many heuristics).We had two hypotheses that we needed to verify experimentally. 1) Is the gradient noise heavy tailed for practical GANS? 2) if yes, does clipping improve the performance of SEG/SGDA? Our experiments provided a positive answer to these two questions in the context of GANs with real-world data. There exists many heuristics that could be used to improve these results (see for instance [Sauer et al. 2022] for the most up-to-date heuristics on StyleGAN). We consider the use of Adam, a method that can provably diverge [Reddi et al. 2019], to be a heuristic that is out of the scope of this paper.
>
> ---
>
> ### References:
>
> Reddi et al. 2019. On the Convergence of Adam and Beyond
>
> Sauer et al. 2022. StyleGAN-XL: Scaling StyleGAN to Large Diverse Datasets

---

### Official Review · Reviewer_q39a · 2022-07-15

**Rating:** 6
**Confidence:** 3
**Soundness:** 4 excellent
**Presentation:** 4 excellent
**Contribution:** 3 good

**Summary:**

This paper studies the stochastic Variational Inequality Problem (VIP) with a potentially unbounded domain and the gradient might not satisfy the light tail assumption. The authors consider Stochastic ExtraGradient (SEG) and Stochastic Gradient Descent Ascent (SGDA) methods using clipped mini-batch stochastic gradients. Under various assumptions such as they proved that both methods converge with a high probability.




**Questions:**

1. Why SNC, SM, QSM and SC are interesting properties? Can the authors provide some real problems (for example some GAN models) that are proven to satisfy these conditions? In literature, these properties are mostly studied for theoretical purposes with only some artificially constructed examples.

2. Can the authors provide some justification for using  GapR(x) as the convergence criterion? For monotone F, we have GapR(x)<= \|F(x)\| R but not the other direction. Why a small GapR(x) indicates a good solution from a perspective of stationarity / variational analysis?

3. The studied methods require mini-batches of big enough sizes. Moreover, these specific sizes depend on some unknown parameters. This is a limitation compared to other methods like Juditsky et al. [2011a] where the mini-batch is just optional and even one sample per iteration still works. Can the authors reduce the mini-batch size to one? If not, what is the technical challenge?

4. How to estimate and select R, which affects many parameters in the theorems and the algorithms including the batch sizes.

**Limitations:**

I have listed the limitations in the questions section above. In addition, the authors do not include some state-of-the-art methods for training GAN in their numerical comparison such as the ones using Adam. What if the state-of-the-art method has even better performances?

**Strengths And Weaknesses:**

Strength: This is the first paper that provides high-probability convergence guarantee for stochastic gradient methods for solving some VIPs without assuming  sub-Gaussian noise and unbounded domains.

Weakness: The authors do not compare their methods with some state-of-the-art methods for training GAN. Also, I have some concerns about using these methods in practice. Please see my questions below.

---

> ### Author Response · Authors · 2022-08-01
> **Authors' response to Official Review of Paper9065 by Reviewer q39a [Part 2/2]**
>
> > The studied methods require mini-batches of big enough sizes. Moreover, these specific sizes depend on some unknown parameters. This is a limitation compared to other methods like Juditsky et al. [2011a] where the mini-batch is just optional and even one sample per iteration still works. Can the authors reduce the mini-batch size to one? If not, what is the technical challenge?
>
> We do not require large batch sizes for 5 out of 6 considered problem classes. In lines 213-216 and 242-243, we discuss this aspect and refer to Corollaries C.1, C.3, D.1, D.2, D.3, where we show that the results hold even for $m = 1$. The only case where our analysis significantly relies on the large batch sizes is the case of star-negatively comonotone problems (Case 2 in Theorem 2.1). This case is not considered by [Judistky et al., 2011a]. Moreover, as we notice in lines 216-219, even known in-expectation complexity results from [Diakonikolas et al., 2021, Lee and Kim, 2021] also rely on large $\mathcal{O}(K)$ batch sizes in this case.
>
> ---
>
> > How to estimate and select R, which affects many parameters in the theorems and the algorithms, including the batch sizes.
>
> In the theoretical results, several parameters do depend on the choice of $R$. However, in practice, parameters are usually tuned because other problem-dependent constants like $L$ are also unknown. Batch sizes are usually chosen according to the memory/computation limits of the machine(s) on which the training is performed. Clipping level usually does not require a thorough tuning [Pascanu et al., 2013]. Stepsize/learning rate is the only parameter requiring thorough tuning, but it is common for many stochastic methods.

---

> ### Author Response · Authors · 2022-08-01
> **Authors' response to Official Review of Paper9065 by Reviewer q39a [Part 1/2]**
>
> We thank the reviewer for their time and their insightful questions.
>
> === Edit on August 3rd (8:00PM UTC): we slightly edited our answer regarding negative co-monotonicity that contained an inaccuracy ===
>
> ---
>
> > The authors do not compare their methods with some state-of-the-art methods for training GAN. […] In addition, the authors do not include some state-of-the-art methods for training GAN in their numerical comparison such as the ones using Adam. What if the state-of-the-art method has even better performances?
>
> The goal of our experiments is to complement our theory with experiments rather than to provide a new state-of-the-art technique (which would require the use of many heuristics). We had two hypotheses that we needed to verify experimentally. 1) Is the gradient noise heavy-tailed for practical GANS? 2) if yes, does clipping improve the performance of SEG/SGDA? Our experiments provided a positive answer to these two questions in the context of GANs with real-world data. Many heuristics could be used to improve these results (see, for instance [Sauer et al. 2022] for the most up-to-date heuristics on StyleGAN). We consider the use of Adam, a method that can provably diverge [Reddi et al. 2019], to be a heuristic that is out of the scope of this paper.
>
> References:
>
> Reddi et al. 2019. On the Convergence of Adam and Beyond
>
> Sauer et al. 2022. StyleGAN-XL: Scaling StyleGAN to Large Diverse Datasets
>
> ---
>
> > Why SNC, SM, QSM and SC are interesting properties? Can the authors provide some real problems (for example some GAN models) that are proven to satisfy these conditions? In literature, these properties are mostly studied for theoretical purposes with only some artificially constructed examples.
>
> The eigenvalues of the Jacobian around the equilibrium of GAN games have been theoretically studied by Mescheder et al. [2018] and Nagaran and Kolter [2017] and practically by Berard et al. [2018]. Certain bounds on the eigenvalues imply some local monotonicity properties. For instance, if $\Re(\lambda)>\mu$ for all $\lambda$ an eigenvalue of the Jacobian around the equilibrium then the operator $F$ is locally strongly monotone around the equilibrium (see, for instance see Azizian et al. [2019]).  One high-level conclusion that can be drawn from Mescheder et al. [2018] and Nagaran and Kolter [2017], and Berard et al. [2018] is that for some GAN formulations, some of the SNC, SM, QSM, and SC hold (at least locally).
>
> That being said, we would like to emphasize the generality of the SNC assumption (Assumption 1.4). This assumption is very mild. For instance, when the operator $F$ is $L$-Lipschitz, it is automatically star-negative strongly-monotone with $\mu=-L$, i.e.,
> $<F(x^*),x-x^*> \geq - L ||x-x^*||^2,$
> and $1/L$-comonotonicity is a slightly stronger condition (if $||F(x^*)|| \approx L||x-x^*||$ then we recovers $<F(x^*),x-x^*> \gtrsim - 1/L ||F(x^*)||^2$). So, it holds for many $L$-Lipschitz operators automatically. Our assumption is similar to the works using the weakest notion of structured non-monotonicity initially proposed for the analysis of EG-type methods by Diakonikolas et al. ([2021]).
>
>
> References:
>
> Mescheder et al. Which Training Methods for GANs do actually Converge?
>
> Nagarajan and Kolter Gradient descent GAN optimization is locally stable
>
> Azizan et al. Accelerating Smooth Games by Manipulating Spectral Shapes
>
> ---
>
> > Can the authors provide some justification for using GapR(x) as the convergence criterion? For monotone F, we have GapR(x)<= |F(x)| R but not the other direction. Why a small GapR(x) indicates a good solution from a perspective of stationarity / variational analysis?
>
> This is the classical convergence criterion for monotone variational inequalities [Nesterov, 2007]. $\text{Gap}_R(x)$ is a valid convergence metric due to the following reason. First of all, (VIP) is equivalent to finding $x^\ast$ such that $\langle F(x^\ast), x - x^\ast \rangle \geq 0$ for all $x \in B_R(x^\ast)$ (this version is usually called *strong variational inequality*). One can show that for continuous and monotone $F$ this problem is equivalent to finding $x^\ast$ such that $\langle F(x), x^{\ast} - x \rangle \leq 0$ for all $x \in B_R(x^\ast)$ (this version is usually called *weak variational inequality*). By definition $\text{Gap}_R (\hat{x}) = \max \langle F(x), \hat{x} - x \rangle$ (maximum is taken over $x \in B_R (x^{\ast})$) shows how $\hat x$ is close to the solution in terms of solving weak variational inequality (note also that $\text{Gap}_R(x^{\ast}) = 0$). See also Lemma 1 from “An Adaptive Mirror-Prox Algorithm for Variational Inequalities with Singular Operators” by Kimon Antonakopoulos, E. Veronica Belmega, Panayotis Mertikopoulos (NeurIPS 2019).

---

### Official Review · Reviewer_NZw2 · 2022-07-15

**Rating:** 6
**Confidence:** 3
**Soundness:** 3 good
**Presentation:** 3 good
**Contribution:** 3 good

**Summary:**

The paper proves the first high-probability complexity results with logarithmic dependence on the confidence level for stochastic methods for solving monotone and structured non-monotone variational inequality problems (VIP) with heavy-tailed noise and unbounded domains.

**Questions:**

1. The paper focuses on monotone variational inequalities, is it possible to get similar results for non-monotone variational inequalities?
2. It would be better to see comparison with SEG in experiments.

**Limitations:**

See above.

**Strengths And Weaknesses:**

Strengths:
The paper is sound, well-organized and well-written.

Weaknesses:
Contribution is not significant enough. The results in the paper still need many assumptions. The paper only focuses on monotone variational inequalities.

---

> ### Author Response · Authors · 2022-08-01
> **Authors' response to Official Review of Paper9065 by Reviewer NZw2 [1/2]**
>
> We thank the reviewer for their review. It seems that there are some misunderstandings that we think we clarified. We hope the reviewer will engage in the reviewer-author discussion if they have any additional questions.
>
> === Edit on August 3rd (7:55PM UTC): we slightly edited our answer regarding negative co-monotonicity that contained an inaccuracy ===
>
> ---
>
> > Contribution is not significant enough.
>
> We politely disagree with this statement. We consider six different problem classes (see Table 1) and provide the first high-probability complexity results for 5 of them. That is, for the five classes of problems we focus on, there were no high-probability results (with logarithmic dependence on the confidence level) *even under the light-tailed noise assumption*. Next, for monotone and Lipschitz variational inequalities, we derive the first high-probability complexity results under the heavy-tailed noise assumption. Moreover, we achieve several other important improvements over the previous result from [Juditsky et al., 2011a]. In particular, we get rid of the bounded domain assumption and relax all the assumptions in such a way that we require the monotonicity, Lipschitzness, and bounded variance assumptions to hold only on some ball around the solution with a radius proportional to the initial distance to the solution. The same is true for other problem classes.
>
> Considering the importance of high-probability convergence analysis of stochastic methods, we believe achieving such generality and tightness makes our contributions highly significant.
>
> ---
>
> > The results in the paper still need many assumptions.
>
> We politely disagree with this claim. As shown in Table 1, in each of the six considered cases, we assume that the variance of the stochastic estimator is bounded on some ball around the solution and 1-2 additional assumptions. Typically, one assumption is about monotonicity/structured non-monotonicity, and the second one is about Lipschitzness (for clipped-SEG) or star-cocoercivity (for clipped-SGDA). As we explain in the next response, monotonicity/structured non-monotonicity is inevitable to guarantee the existence of a solution. Lipschitzness for Extragradient(EG)-type of methods is also a central property for its convergence since EG is an approximation of the Proximal Point method, and the approximation argument is valid due to Lipschitzness of the operator (see Theorem 1 from [Mishchenko et al., 2020]). Next, it is known that Gradient Descent-Ascent is not necessarily converging for general monotone and Lipschitz variational inequalities. Therefore, assuming a sort of cocoercivity is inevitable in the analysis of clipped-SGDA. Bounded variance assumption is also reasonable and relatively popular [Ghadimi and Lan, 2012, 2013, Juditsky et al., 2011b, Nemirovski et al., 2009], since one has to assume something about the noise to guarantee the convergence. Finally, the assumptions we use in the analysis are the mildest known ones in the literature on high-probability convergence of stochastic methods for variational inequalities [Judistky et al., 2011a].
>
> ---
>
> > The paper focuses on monotone variational inequalities, is it possible to get similar results for non-monotone variational inequalities?
>
> It is known that assuming some kind of structured non-monotonicity is necessary to provide positive convergence results since counter–examples can be constructed otherwise [Letcher et al 2021, Hsieh et al. 2021].
>
> Moreover, Daskalakis et al. (2021) show, that the computation of approximate first-order locally optimal solutions is intractable for general variational inequalities. Therefore, it makes sense to consider non-monotone problems with a certain structure. We point out that only one problem class (out of 6 considered in the paper) relies on monotonicity. All other cases do cover some special non-monotone operators. Indeed, Assumptions 1.5 and 1.6 do not imply monotonicity (see lines 114-115 and 122-125).
>
> Moreover, we would like to emphasize the generality of the SNC assumption (Assumption 1.4). This assumption is very mild. For instance, when the operator $F$ is $L$-Lipschitz, it is automatically star-negative strongly-monotone with $\mu=-L$, i.e.,
> $<F(x^*),x-x^*> \geq - L ||x-x^*||^2,$
> and $1/L$-comonotonicity is a slightly stronger condition (if $||F(x^*)|| \approx L||x-x^*||$ then we recovers $<F(x^*),x-x^*> \gtrsim - 1/L ||F(x^*)||^2$). So, it holds for many $L$-Lipschitz operators automatically. Our assumption is similar to the works using the weakest notion of structured non-monotonicity initially proposed for the analysis of EG-type methods by Diakonikolas et al. ([2021]).
>
> That is, our paper focuses not only on monotone variational inequalities.
>
> Hsieh et al. The Limits of Min-Max Optimization Algorithms: Convergence to Spurious Non-Critical Sets
>
> Letcher. On the Impossibility of Global Convergence in Multi-Loss Optimization

---

> > ### Author Response · Authors · 2022-08-02
> > **Authors' response to Official Review of Paper9065 by Reviewer NZw2 [2/2]**
> >
> >
> > ---
> >
> > > It would be better to see comparison with SEG in experiments.
> >
> > We have added the requested experiments for WGAN-GP, and the results can be found in figure 2(f) in the main paper, as well as table 3 and figure 6 in the appendix. These results are consistent with the rest of our experimental findings: the clipped methods consistently outperform the non-clipped methods by a significant margin. The best FID for SEG obtained over the hyperparameter sweep is 176.5, vs 36.3 for Clipped-SEG (norm) and 36.2 for Clipped-SEG (coordinate).

---

> > > ### Comment · Reviewer_NZw2 · 2022-08-09
> > > **Response to the authors' feedback**
> > >
> > > Thank you for your replies! I raised my score since the authors give good explanations for my concerns and add requested experiments.

---

> > > > ### Author Response · Authors · 2022-08-09
> > > > **Thank you!**
> > > >
> > > > We thank the reviewer for the feedback and for raising the score!

---

### Author Response · Authors · 2022-08-01
**General response to the reviewers**

We thank the reviewers for their positive feedback. The general evaluation is mostly positive. In summary, the reviewers identified the following strengths of our work:
- Quality and Clarity: The paper is “sound, well-organized and well-written”. “The table of results in the introduction is straightforward”.(reviewers NZw2 and nza4)
- Originality: This is the first paper that provides a high-probability convergence guarantee for stochastic gradient methods for solving some VIPs without assuming sub-Gaussian noise and unbounded domains. (reviewers q39a and nza4)
- Significance: This paper addresses VIP with heavy-tailed noise, which is common in industries and business (reviewer nza4)

Two limitations commonly emerged from the reviews:

> The relevance of our assumptions

We believe we have among the mildest assumptions one could find in the literature for such analysis. For instance for clipped-SEG, we rely on:

1. Bounded variance only in a ball around the optimum (which is weaker that the standard assumption that ask for uniformly bounded variance)

2. Lipchitz continuity of the operator (a very standard assumption)

3. A notion of monotonicity or structured non-monotonicity (either assumption 1.3, 1.4 or 1.5)
Importantly, to our knowledge, Assumption 1.4 is the weakest structured non-monotonicity that has been used to provide convergence rates for VIPs.

> The experimental comparison with Adam

The goal of our experiments is to complement our theory with experiments, rather than to provide a new state of the art technique (which would require the use of many heuristics).We had two hypotheses that we needed to verify experimentally. 1) Is the gradient noise heavy tailed for practical GANS? 2) if yes, does clipping improve the performance of SEG/SGDA? Our experiments provided a positive answer to these two questions in the context of GANs with real-world data. There exists many heuristics that could be used to improve these results (see for instance [Sauer et al. 2022] for the most up-to-date heuristics on StyleGAN). We consider the use of Adam, a method that can provably diverge [Reddi et al. 2019], to be a heuristic that is out of the scope of this paper.

References:

Reddi et al. 2019. On the Convergence of Adam and Beyond

Sauer et al. 2022. StyleGAN-XL: Scaling StyleGAN to Large Diverse Datasets

---

### Meta-Review · Area_Chair_89Sz · 2022-08-26

**Recommendation:** Accept
**Confidence:** Certain

**Metareview:**

This is the first paper that provides high-probability convergence guarantee for stochastic gradient methods for solving some VIPs without assuming sub-Gaussian noise and unbounded domains. Although the numerical experiments are not fully convincing (e.g., no comparison with Adam for training GAN), the overall theoretical contribution is significant enough. I would recommend accepting this paper.

**Award:**

No

---

### Decision · Program_Chairs · 2022-09-14

Accept